# A decision-space model explains context-specific decision-making

Dirk W. Beck[1], Cory N. Heaton [2], Luis D. Davila[1,3,11], Lara I. Rakocevic[1,3,11], Sabrina M. Drammis[4,11], Danil Tyulmankov[5,11], Atanu Giri[1], Shreeya Umashankar Beck[2], Qingyang Zhang[6], Michael Pokojovy [7], Kenichiro Negishi [8], Alexis A. Salcido [2], Neftali F. Reyes[2], Andrea Y. Macias[2], Serina A. Batson[2], Paulina Vara[2], Raquel J. Ibáñez Alcalá [2], Safa B. Hossain[2], Graham L. Waller[2], Laura E. O'Dell[9], Travis M. Moschak[2], Ki A. Goosens [3,10] ✉ & Alexander Friedman [1,2] ✉

Optimal decision-making requires consideration of internal and external contexts. Biased decision-making is a transdiagnostic symptom of neuropsychiatric disorders. We created a computational model demonstrating how the striosome compartment of the striatum constructs a context-dependent mathematical space for decision-making computations, and how the matrix compartment uses this space to define action value. The model explains multiple experimental results and unifies other theories like reward prediction error, roles of the direct versus indirect pathways, and roles of the striosome versus matrix, under one framework. We also found, through new analyses, that striosome and matrix neurons increase their synchrony during difficult tasks, caused by a necessary increase in dimensionality of the space. The model makes testable predictions about individual differences in disorder susceptibility, decision-making symptoms shared among neuropsychiatric disorders, and differences in neuropsychiatric disorder symptom presentation. The model provides evidence for the central role that striosomes play in neuroeconomic and disorder-affected decision-making.

Decision-making is altered in neuropsychiatric disorders affecting the basal ganglia[1]. A range of experimental evidence links balances between the striatal compartments and connected brain regions to decision-making function and dysfunction[2-6]. Understanding these intricate interactions will be crucial for designing next-generation treatments.

Striatal neurons can be categorized into groups via neurochemistry and connectivity, including the striosome and matrix compartments[4,7]. Striosomal spiny projection neurons (sSPNs) make up ~10–15% of the striatum and matrix spiny projection neurons (mSPNs) the remaining ~85–90%[8,9] (for acronyms, see Table 1). New technologies, including recording, targeting methods, and genetically engineered mice, have

[1]Computational Science Program, University of Texas at El Paso, El Paso, TX, USA. [2]Department of Biological Sciences, University of Texas at El Paso, El Paso, TX, USA. [3]Department of Psychiatry, Department of Pharmacological Sciences, Icahn School of Medicine at Mount Sinai, New York, NY, USA. [4]Artificial Intelligence Laboratory, Department of Computer Science, Massachusetts Institute of Technology, Cambridge, MA, USA. [5]Ming Hsieh Department of Electrical and Computer Engineering, Viterbi School of Engineering, University of Southern California, Los Angeles, CA, USA. [6]Department of Biomedical Informatics, Harvard Medical School, Cambridge, MA, USA. [7]Department of Mathematics and Statistics, Old Dominion University, Norfolk, VA, USA. [8]National Institute on Drug Abuse, Baltimore, MD, USA. [9]Department of Psychology, University of Texas at El Paso, El Paso, TX, USA. [10]Center for Translational Medicine and Pharmacology, Icahn School of Medicine at Mount Sinai, New York, NY, USA. [11]These authors contributed equally: Luis D. Davila, Lara I. Rakocevic, Sabrina M. Drammis, Danil Tyulmankov. ✉e-mail: ki.goosens@mssm.edu; afriedman@utep.edu

enabled important new discoveries about differential roles for strio-somes and matrix in decision-making[3,10–19], including in disorders[2,4]. Further, both sSPNs and mSPNs belong to either the direct pathway (dsSPNs and dmSPNs), identified by D1 receptor expression, or the indirect pathway (isSPNs and imSPNs) identified by D2 receptor expression[4,20]. Notably, dsSPNs, to a greater extent than dmSPNs, project to regions which influence midbrain dopamine release via a sub-circuit conserved across species[8,16,21–24] (Fig. 1, Supplementary Table 1). Thus, dsSPNs, isSPNs, dmSPNs, and imSPNs appear to have different

circuit roles[16,20], raising the possibility they each play a distinct functional role during decision-making (see Supplementary Note 1).

Models of cortico-striatal-basal ganglia-thalamic regions often examine the direct, indirect, and, more recently, the hyperdirect pathways[25,26] to elucidate how they interact during different cognitive processes like decision-making, action selection, and learning. These models explore potential circuit functionality in a variety of ways including: exploring how cortico-striatal neurons may utilize general features to facilitate adaptable/flexible decision-making[27], examining

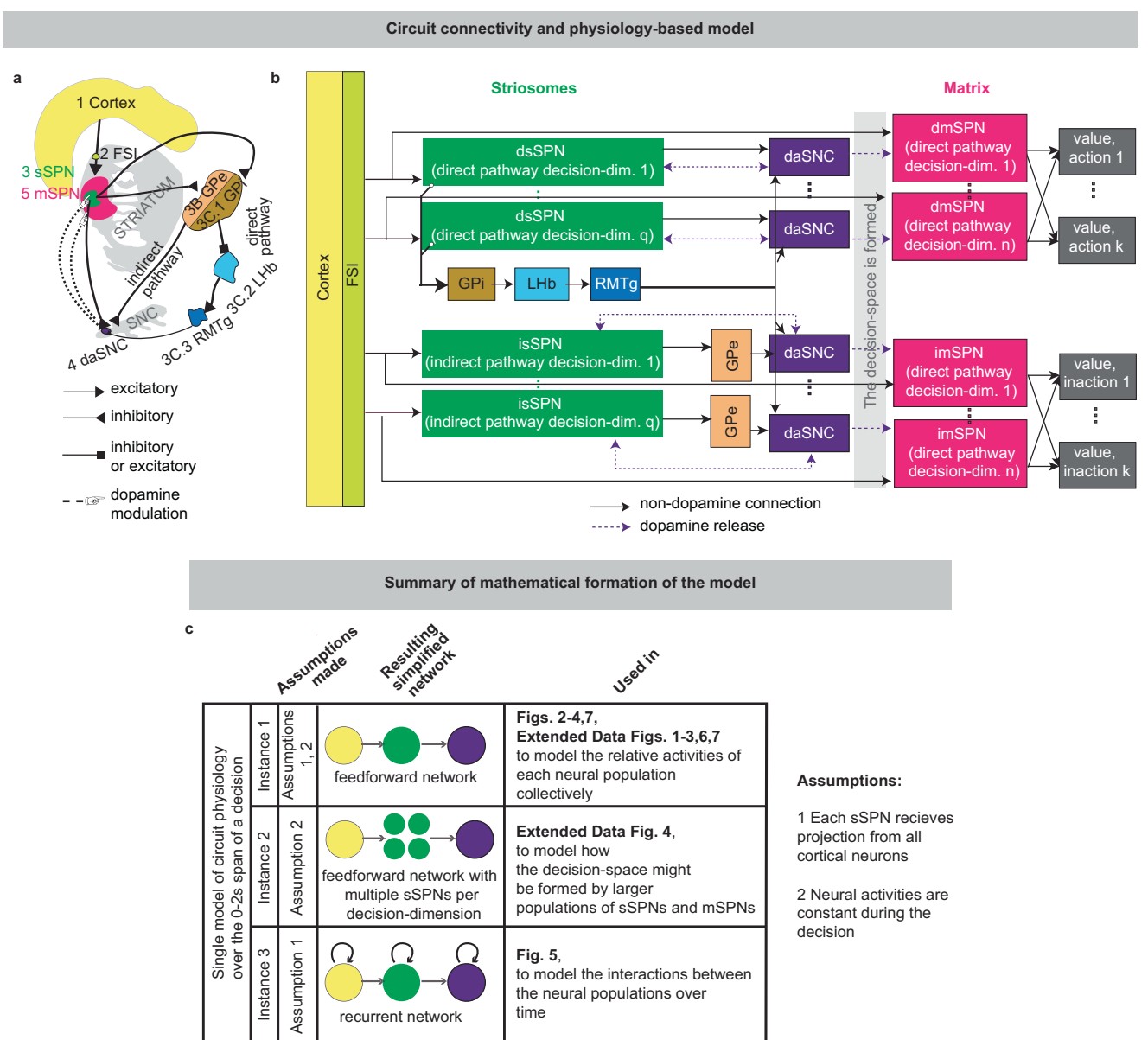

**Fig. 1 | A striosome-centered circuit model regulating dopamine release to the striatum. a** The circuit of the cortex, striosomes (sSPN), globus pallidus externus (GPe), globus pallidus internus (GPi), lateral habenula (LHb), rostromedial tegmental nucleus (RMTg), dopaminergic neurons of the substantia nigra compacta (daSNC), and matrix neurons (mSPN). Numbers show the order of connection for the three subcircuits (1 → 2 → 3 → 4, 1 → 2 → 3 → 3B → 4 → 5, 1 → 2 → 3 C.1 → 3.C.2 → 3.C.3 → 4 → 5). **b** Circuit architecture. Cortical activity is processed by several sSPN and mSPN subpopulations corresponding to "decision-dimensions." The sSPN subpopulations, together with GPi, LHB, and RMTg (in the direct pathway) and GPe (in the indirect pathway), control the activity of subpopulations of daSNC, each also corresponding to a decision-dimension. When these subpopulations release dopamine to the corresponding mSPN subpopulations, a "decision-space" is formed (gray arrows). Based on the decision-space, mSPNs determine action values

(dark gray rectangles). **c** The circuit is modeled over the period of time of a decision. In each of three model instances, an assumption is made that leads to a simple mathematical formulation (**Analyzed Instances of the Model, Methods**). After these assumptions, Instance 1 simplifies to a feedforward network with either one element per decision-dimension per neuron type (for cortex, sSPN, daSNC, and mSPN) or one element per neuron type (for GPi, LHb, RMTg, GPe). Instance 2 simplifies to a feedforward network with multiple sSPNs and mSPNs corresponding to each decision dimension. Instance 3 simplifies to a recurrent network with an architecture similar to Instance 1. Thus, Instance 1 is used to model the basic relationships between circuit elements, Instance 2 to model the behavior of populations of sSPNs and mSPNs, and Instance 3 to model interactions between circuit elements over time.

**Table 1 | Terminology**

| Term | Definition |
|---|---|
| GPi | Globus pallidus internus |
| GPe | Globus pallidus externus |
| LHb | Lateral habenula |
| RMTg | Rostromedial tegmental nucleus |
| daSNC | Dopaminergic neurons of the substantia nigra compacta |
| FSI | Striatal fast-spiking Interneuron |
| SPN | Striatal projection neuron |
| sSPN | Striosomal striatal projection neuron |
| mSPN | Matrix striatal projection neuron |
| dsSPN | Direct pathway striosomal striatal projection neuron |
| isSPN | Indirect pathway striosomal striatal projection neuron |
| dmSPN | Direct pathway matrix striatal projection neuron |
| imSPN | Indirect pathway matrix striatal projection neuron |
| BG | Basal ganglia |
| Decision-dimension | An axis of the coordinate system with which SPNs (dsSPNs, isPSNs, dmSPNs, imSPNs) process cortical activity. Subpopulations of SPNs (for each dsSPNs, isSPNs, dmSPNs, imSPNs) encode data along different decision-dimensions. Cortical activity is linearly mapped to this basis of decision-dimensions such that the activity of a single cortical neuron is no longer encoded by a single SPN. Certain decision-dimensions might correspond more predominantly, for instance, to reward level, cost level, or novelty level, as encoded across multiple cortical neurons. Decision-dimensions are modeled as the principal components of cortical activity. For additional details, see Decision-dimensions and decision-space, Methods. |
| Decision-space | The subspace produced by the decision-dimensions which are selected by the circuit to be used during a decision. Decision-space is formed when dopamine releases to matrix striatal projection neurons (dmSPNs and imSPNs), signaling that certain decision-dimensions are important and others are unimportant, and therefore can be excluded from the subspace. |
| Action value | Value assigned by the circuit to an action |
| Inaction value | Value assigned by the circuit for refraining from an action |
| Prediction error | The difference between expected and observed information along a data axis (for instance, reward prediction error, punishment prediction error, or novelty prediction error) |
| Circuit activity | The set of average activities of each circuit element during a decision |
| Advantage | The degree to which a circuit activity is preferred. Used in our analysis of the change in circuit activity between trials. |

how this circuit shifts reward-seeking based on internal states[28], explaining how basal ganglia structures exercise nuanced control over action selection despite subsequent regions containing fewer neurons[29], and exploring the role of direct and indirect pathway competition during action selection[30,31]. Another approach taken when modeling the basal ganglia is to explore how dopamine and basal ganglia corticostriatal loops influence basal ganglia functions like decision-making, learning, and action selection. One example is where corticostriatal loops modulate dopaminergic activity/plasticity where dopamine signals vector-based error signals associated with different task-related factors[32]. Many of these models, while exploring important ideas for cortico-basal ganglia-thalamic interactions/functions during decision-making, learning, and action selection, do not account for differences between the striatal matrix and striosomal compartments. Without the striosome versus matrix distinction, they do not capture the interplay between dopamine and the striatum in a precise way. The dopaminergic interplay is especially important for the study of neuropsychiatric disorders, because disorders that differentially affect the direct versus indirect pathways[33] have been found to also affect striosomes versus matrix differently[2], suggesting that attention to all four dsSPN/isSPN/dmSPN/imSPN compartments is necessary for an accurate understanding of decision-making in normal and disordered states.

To close this gap, we formed a model that accounts for striosome versus matrix subdivisions, including the selective modulation of midbrain dopamine by striosomes (Fig. 1, Supplementary Fig. 1). From our physiological model arises the concept of a "decision-dimension", our term for an axis along which the modeled circuit encodes decision-related information (Fig. 2, Supplementary Fig. 2; for terminology, see Table 1). During a decision, decision-dimensions are prioritized in a context-dependent manner, forming a mathematical "decision-space."

We present evidence, via our analysis of neural recordings and our models of findings from experimental literature, for the core tenets of our model: that subpopulations of SPNs encode information along decision-dimensions, that a decision-space is formed, and that the decision-space changes across contexts (Figs. 3,4, Supplementary Figs. 3–6). Then we demonstrate the power of the model to explain a range of physiological and behavioral phenomena, including the roles of the indirect/direct pathway and reward prediction error (RPE) (Fig. 5, Supplementary Fig. 7). Finally, we speculate how the model might explain behavioral phenomena observed in psychiatry (Figs. 6, 7, Supplementary Figs. 8, 9) and suggest future experiments (Supplementary Fig. 10).

## Results
### Model description
Our model describes how physiological interactions between elements of a striosome-centered circuit inform decision-making (Fig. 1a, b, for extended reasoning behind our choice of circuit elements, see Supplementary Note 2; for extended documentation of the model, see **Analyzed instances of the model, Methods**). Depending on the purpose of our analysis, we either consider the dynamics of the circuit over the span of the decision or assume, for simplicity, that the circuit elements express constant activities across the short decision period (Fig. 1c, Supplementary Fig. 1).

In the model, **striatum-projecting cortical neurons** that encode mixed information serve as the input to the modeled circuit. We assume for the purpose of our model that the information passed from the cortex to each striatal compartment is roughly similar (in actuality, there is much overlap with some differences, see Supplementary Table 1 and Supplementary Note 3). The cortical neurons synapse on

subpopulations of proximate SPNs which have been found to each encode distinct information[34]. During this process, fast spiking inter-neurons (**FSIs**) perform a normalization operation (Supplementary Fig. 2a). Mathematically, we represent this as a matrix $\mathbf{W}_P$ dictating cortex-SPN connection weights for each pathway $P$ (direct or indirect) mapping cortical activity $\mathbf{x}_P$ to the coordinate space of sSPNs, where it undergoes divisive normalization by FSI activity $c_P$ and a shift in activity $b_{\text{sSPN}}$ to form sSPN activity $\mathbf{s}_{\text{sSPN},P}$:

$$\mathbf{s}_{\text{sSPN},P} = \frac{1}{c_P}\mathbf{W}_P^{\mathbf{T}}\mathbf{x}_P + b_{\text{sSPN}} \qquad (1)$$

Information from sSPNs is then passed to dopaminergic neurons of the substantia nigra compacta (**daSNC**). daSNC neurons, like sSPNs, have been shown to be organized topologically into subpopulations that each encode distinct information[35]. Experimental evidence suggests that signals are passed from sSPN subpopulations to daSNC neurons in three ways (see Supplementary Table 1): **A (dsSPN→daSNC)**. Subpopulations of dsSPNs inhibit daSNC subpopulations directly via dendrite bouquets[36]. Thus, in our model, each dsSPN subpopulation inhibits a corresponding daSNC subpopulation. **B (isSPN → GPe→daSNC)**. isSPNs send signals to daSNC neurons via GPe[16], which inhibit daSNC subpopulations. Thus, in our model, each isSPN subpopulation disinhibits a corresponding daSNC subpopulation. **C (dsSPN → GPi → LHb → RMTg→daSNC)**. GPi integrates signals from many sSPN subpopulations through synapses that release both GABA and glutamate[37], and the LHb, when activated by GPi, powerfully inhibits multiple of the dopaminergic subpopulations via RMTg[38,39]. So, in our model, shifts in this pathway lead to a shift across all daSNC subpopulations. Mathematically, we represent the three circuits as daSNC combining activity from sSPNs $\mathbf{S}_{\text{sSPN},P}$ (with connection weights $w_{\text{sSPN}\rightarrow\text{daSNC},i,P}$ corresponding to each decision-dimension $i$ and pathway $P$) with RMTg activity, GPe activity $z_{\text{GPe},P}$ (only for the indirect pathway), and an additive shift $z_{\text{daSNC},i,P}$:

$$\text{daSNC}_{i,P} = \frac{1}{1 + \exp(w_{\text{sSPN}\rightarrow\text{daSNC},i,P} \cdot (s_{\text{sSPN},i,P} + z_{\text{GPe},P}) + \text{RMTg} - z_{\text{daSNC},i,P})} \qquad (2)$$

In addition to sSPNs, there are subpopulations of **mSPNs**, termed matrisomes[4], that densely surround sSPN subpopulations. In our model, we hypothesize that these sSPN and mSPN subpopulations communicate with one another via dopamine release from the corresponding daSNC subpopulation (there are other sSPN→mSPN connections that we do not model that play more local roles, see Supplementary Note 4). There are multiple groups of these functionally connected sSPNs, daSNCs, and mSPNs. In the model, when a daSNC subpopulation is active, dopamine is released to the corresponding sSPN and mSPN subpopulations, resulting in enhanced or inhibited mSPN reception of cortical signal among the subpopulations, as shown in experimental work[40]. Mathematically, mSPN activity is defined similarly to sSPN activity, but for a diagonal matrix $\mathbf{S}_P$ corresponding to the dopamine release that probabilistically defines the decision-space, with $P(\mathbf{S}_{P,\text{ii}} = 1) = \text{daSNC}_{i,P}$:

$$\mathbf{s}_{\text{mSPN},P} = \tfrac{1}{c_P}\mathbf{S}_P\mathbf{W}_P^{\mathbf{T}}\mathbf{x}_P \qquad (3)$$

mSPNs have been found to be primarily involved in motor functions, projecting to regions including the GPi, SNr, and then to brainstem motor programs[41]. SPN activity (which is predominantly mSPN) has been shown to contribute to action selection and initiation[42]. The direct pathway has generally been implicated in promoting actions and the indirect pathway in preventing actions[43]. So, in our model, the

output of the mSPN circuit is the definition of action values (the value of performing various actions, encoded by the direct pathway) and inaction values (the value of refraining from those actions, encoded by the indirect pathway) by mSPN signals on route to downstream regions. Mathematically, values $v_{j,P}$ (for each action/inaction $j$, pathway $P$) are defined based on mSPN activity, internal coefficients $\boldsymbol{\beta}_{j,P}$, and priors $\alpha_{j,P}$ (which are set to arbitrary values that are constant across analyses, see **Common parameters, Rationale behind parameter choices, Methods**):

$$v_{j,P} = \frac{1}{1 + \exp(-\boldsymbol{\beta}_{j,P}\,\mathbf{s}_{\text{mSPN},P} - \alpha_{j,P})}, \qquad (4)$$

Based on these values, actions are either performed or refrained from over time. We model this using a Merton process model[44] where the first process to hit a threshold is enacted (direct pathway) or refrained from (indirect pathway). This model functions similarly to a drift-diffusion model[45], which has been used to model decisions including in the basal ganglia[46], but is scaled to the case where more than two actions can be taken. See **Defining choice, Methods**.

Thus, our model is constructed based on the anatomy and physiology of the striosome-centered circuit. The physiological description also produces a simple and convenient geometric interpretation (Fig. 2a, b). If we let each SPN and daSNC subpopulation encode the principal components of cortical activity, such as could be learned via a modified Oja's rule[47], then the columns of $\mathbf{w}_P$ become orthogonal. So, each SPN subpopulation can be thought of as encoding information along an axis of Euclidean space. We term these axes "decision-dimensions" and have evidence that they correspond to constructs such as reward, cost, or novelty (discussed in more detail below). Because, in the model, dsSPNs and isSPNs form subpopulations separately, there is one set of decision-dimensions corresponding to the direct pathway and another set corresponding to the indirect pathway. We suggest that when dopamine is released to SPNs, selectively enhancing the reception of the cortical signal, decision-dimensions are effectively prioritized. Therefore, cortex, striosomes, and dopamine work together to form a "decision-space", only focusing on decision-dimensions that are necessary to solve the task at hand. In this light, sSPN and daSNC, via pairs of connected subpopulations that process information in parallel, serve the functional role of selecting precisely which decision-dimensions should receive high priority. In particular, dsSPNs select which direct pathway decision-dimensions to use, and isSPNs select which indirect pathway decision-dimensions to use (Fig. 2c, d, discussed in detail in **Functional roles of dsSPNs, isSPNs, dmSPNs, and imSPNs**). On the other hand, GPi, LHb, and RMTg, by prioritizing or deprioritizing all daSNC subpopulations together, determine precisely how many decision-dimensions should be used (Fig. 2e–g, Supplementary Fig. 2b–i).

Importantly, the advantage of this formation is not only conceptual, but practical for linking physiology to decision-making (Fig. 2h, Supplementary Fig. 2j–p). Distinct sSPN and/or daSNC subpopulations have been found to encode, for instance, reward[12,14,15,35], cost[12,14,15,35], or novelty[48]. Thus, we might imagine that decision-dimensions could correspond loosely to reward level, cost level, or novelty level. If this is the case, a logical prediction of the model arises: we would expect a low-dimensional decision-space to be formed during a simple choice (e.g. between two rewards) and a high-dimensional decision-space to be formed during a more difficult choice (e.g. between offers which each have benefits and costs that must be weighed in order to solve the problem). This hypothesis, if proven, would allow us to infer the decision-spaces of behaving rodents or humans simply by regressing sSPN activity on experimental parameters (for example, temperature or music volume), as we demonstrate using synthetic data (Supplementary Fig. 2q, r). For example, a significant correlation between sSPN activity and novelty

level would indicate the existence of a decision-dimension that corresponds roughly to novelty (**Inferring decision-space from SPN activity and choice, Methods**). We sought to determine if this hypothesis is supported by experimental physiological data collected during decision-making.

### Support for the model: context-dependent sSPN physiology matches model predictions

We began by asking whether the results of the physiological sSPN literature support a link between the decision-space (which our model postulates is driven by sSPN activity) and task difficulty (inferred from experimental inputs, for instance, a simple task with reward only versus a difficult task with conflicting rewards and costs). We began with the experimental literature on sSPNs. In one experiment[11], sSPNs were optogenetically stimulated (or inhibited) during a rodent conflict decision-making task. Per our model, this should cause inhibition (or disinhibition) of daSNC neurons, leading to reduced (or enhanced) dopamine release to SPNs, producing a lower- (or higher-) dimensional decision-space. Indeed, the stimulation led to choices indicative of decision-making using few informational dimensions, while inhibition led to choices indicative of decision-making using multiple informational dimensions (Fig. 3a, b, Supplementary Fig. 3a, **Effect of sSPN activity on decision-space** and **Effect of decision-space on choice, Methods**). A second experiment[11] tested, reversely, the activity of striosomes during simple versus difficult tasks. As the model expects, striosomal activity during simple tasks resembled the levels observed during optogenetic excitation, whereas striosomal activity during difficult tasks resembled the levels observed during optogenetic inhibition (Fig. 3c, Supplementary Fig. 3b–d). In another study, striosome activity was lower among rodents that learned a difficult reversal learning task than among rodents that did not[10]. Thus, there appears to be a relationship between sSPN physiology and task difficulty in the direction expected by our model.

### Support for the model: context-dependent sSPN-mSPN synchrony matches model predictions

Importantly, the result above does not distinguish our model from the alternative, simpler explanation that sSPNs might collectively encode task difficulty. We could term such a model a "conflict model", where sSPN activity tracks the overall conflict present in a task (see Supplementary Table 2). To determine whether it was changes to sSPN subpopulations driving the overall change in sSPN activity (i.e. decision-space), rather than a general effect, we analyzed the paired activities of sSPNs during simple versus difficult tasks using the Corticostriosomal Circuit Stress Experimental database[3]. We hypothesized that a greater number of sSPNs and mSPNs would be functionally connected during difficult tasks. This prediction arises from the circuit connectivity in our model, where sSPNs are functionally connected to mSPNs via daSNC, and each additional prioritized subpopulation causes more sSPN→daSNC→mSPN modulation (Fig. 3d). This could be observed, for instance, as an increase in striosome and matrix correlation as decision-space increases. This hypothesis is also inspired by the observation that striosome and matrix activity has been found to roughly track one another over time in a difficult task[13].

To test this, we analyzed the synchrony between striosome neurons and matrix neurons during simple decisions (that forced choice between either two rewards or two costs) and during difficult decisions (that forced choice between offers that contained both rewards and costs together; see Supplementary Note 5). We measured synchrony as the cross-correlation between sSPN activity and mSPN activity over the period of the task. To control for possible physiological differences across the phases of the rats' movements, we also developed a custom tool that uses Granger

causality to measure how neurons interact over the span of the task (see **Connected SPNs through Granger causality, Methods**). Per both metrics, synchrony was significantly higher in the difficult task. Further, synchrony scaled with the difficulty with which the rats treated the difficult task, as measured based on deliberation time; that is, given different deliberation times for individual rats, given the same difficult decisions, longer deliberation times correlated with greater synchrony (Fig. 3e, Supplementary Fig. 3e–n, **Connected SPNs through cross-correlation, Methods**). Thus, rather than sSPNs encoding conflict in their general level of activity, there appears to be an important relationship between sSPN and mSPN subpopulations during high-conflict tasks. Our analysis does not confirm that sSPNs have a causal effect on mSPNs. However, if this is assumed, the evidence suggests enhanced modulation of mSPN subpopulations by sSPN subpopulations during the difficult tasks, that is, a formation of a higher-dimensional decision-space.

### Support for the model: Dimensionality reduction from the cortex to SPNs

Notably, the synchrony analysis above demonstrates the use of subpopulations during context-dependent decision-making, but it does not test whether those subpopulations correspond to decision-dimensions. There is, however, evidence that SPNs encode what we term decision-dimensions. Experimental work has demonstrated that distinct SPN subpopulations encode different information, and that these subpopulations persist across days[34]. Further, our analysis suggests that dimensionality reduction occurs from the cortex to SPNs, as it does in our model during the mapping of information to a basis of decision-dimensions. Cortical neurons had the most coordinated activities over time (measured as effective correlation[49]), then FSIs, followed by sSPNs and mSPNs (Fig. 3f, **Analyzing neural dimensionality reduction, Methods**). This would suggest a higher-dimensional representation in cortex, where neurons encode similar information over time, than in the downstream regions, where information is compactly encoded in neurons that behave differently over time. Thus, it seems that cortex→SPN dimensionality reduction occurred during the tasks, like in our model the mapping from high-dimensional cortical information to a basis of SPN decision-dimensions.

### Support for the model: Strong alignment to the experimental literature on SPNs compared to alternative models

Up until this point, the analysis is based on a selection of experimental literature relating sSPN activity to mSPN activity and choice. We wished to test the model more broadly using a range of experimental results. To this end, we devised five tests that link sSPN activity to choice, each verifiable with experimental literature, that might support or reject the decision-space model (Supplementary Fig. 3o–x, Supplementary Table 2). As benchmarks, we also constructed, from roles commonly assigned to sSPNs, four alternative models in which sSPNs only encode 1) conflict, 2) subjective value, 3) prediction error, or 4) actions. We found that while the alternative models each can be used to interpret a subset of the experimental evidence, only the decision-space model aligned to the breadth of it (Supplementary Table 3). A selection of experimental studies on GPi, LHb, and daSNC also align with the decision-space model (see Supplementary Tables 4–7), thus offering a new lens through which to interpret their functions.

### Inability to form a high-dimensional decision-space in disorders

We next applied our model to the experimental literature on neuropsychiatric disorders, wondering if it could offer insight into disorders associated with sSPN changes. Interestingly, in an experimental study on chronic stress[3], sSPNs were hyperactive compared to controls during the most difficult task but less affected during the simpler tasks (Fig. 4a). Meanwhile, the rodents were less adherent to reward level

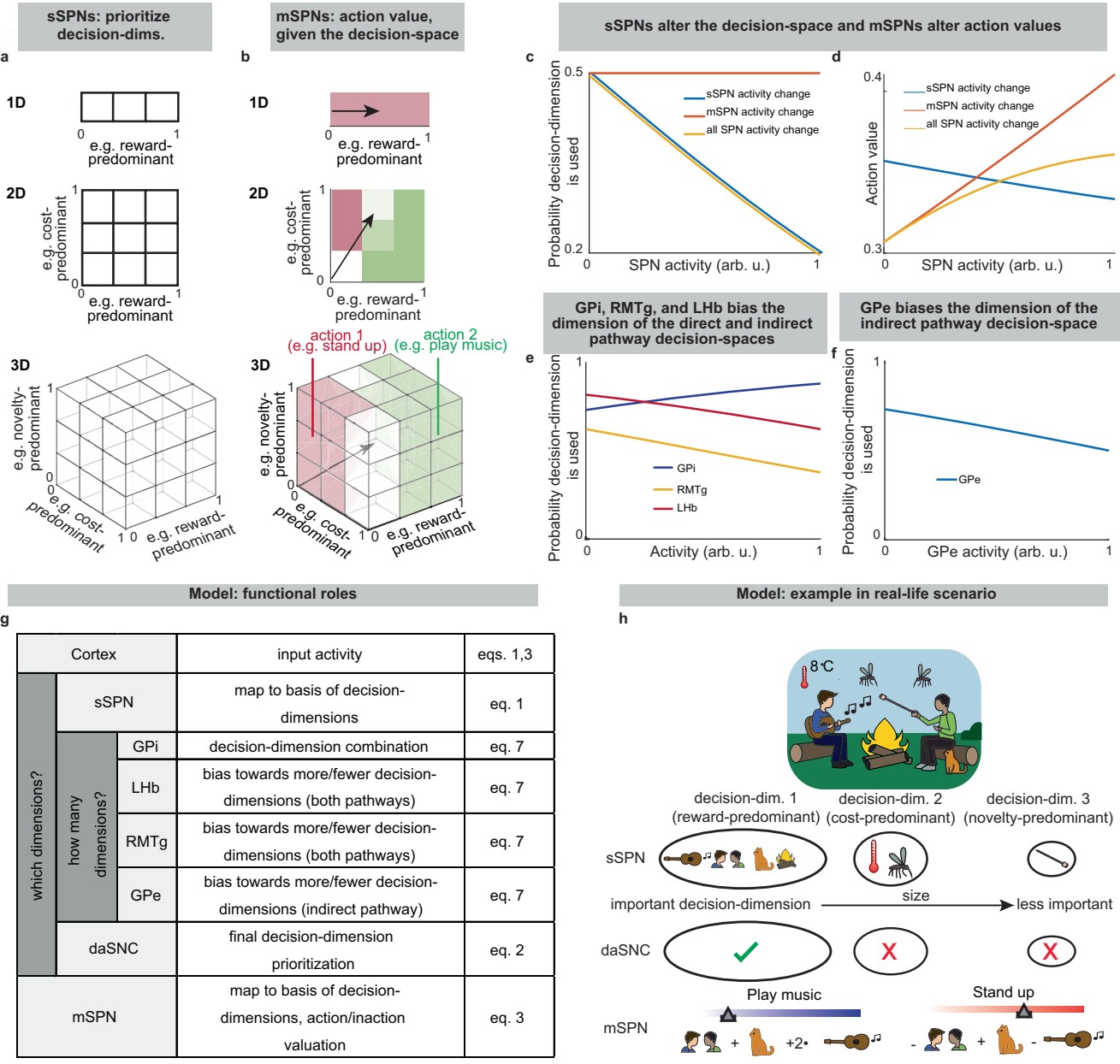

**Fig. 2 | A conceptually convenient "decision-space" framework arises from the physiological model. a** Striosome (sSPN) subpopulations encode information along orthogonal decision-dimensions, so sSPNs can be thought of as encoding information in Euclidean space, with each axis a decision-dimension. During the decision, important decision-dimensions are retained and non-important decision-dimensions are deprioritized, forming a decision-space. One decision-space is formed in the direct pathway and another in the direct pathway. **b** After the decision-space is formed, matrix neurons (mSPNs) define action values (colors) based on cortical input (vectors) and the decision-space (grids). In a 1D decision-space, action values (in the direct pathway) or inaction values (in the indirect pathway) are assigned purely based on the activity of mSPN along a single decision-dimension. In multi-dimensional spaces, action/inaction values depend on the activity along multiple decision-dimensions. **c** sSPN, but not mSPN, activity affects the dimension of the decision-space in each pathway. **d** mSPN activity directly affects action values. sSPN activity also affects action values through its alteration of the decision-space. **e** Globus pallidus internus (GPi), lateral habenula (LHb), rostromedial tegmental nucleus (RMTg), and dopaminergic neurons of the

substantia nigra compacta (daSNC) activities affect the dimension of the decision-space in the direct pathway. **f** Globus pallidus externus (GPe) activity affects the dimension of the decision-space in the indirect pathway. **g** Conceptual circuit roles. Cortex encodes environmental/internal data. sSPN subpopulations encode data along decision-dimensions. The GPi→LHb→RMTg pathway biases the circuit's use of decision-dimensions. daSNC subpopulations, corresponding to decision-dimensions, activate when their dimension is important, releasing dopamine to select mSPN subpopulations. This forms a decision-space from important decision-dimensions. mSPNs then define action values within this space. **h** Example of how the circuit might process a decision. Decision-dimensions (e.g., "reward-predominant" for music/cat; "cost-predominant" for temperature/mosquitoes; "novelty-predominant" for marshmallows) capture information. The most important decision-dimensions (e.g., only reward-predominant) are retained. mSPN forms action values using rules for these retained dimensions, creating the decision-space. Actions (e.g., "play music," "stand up") are assigned values, and decisions are made based on these values.

only in the difficult task, suggesting dysfunction in processing reward and cost together (Fig. 4b, c, **Effect of decision-space on choice, Methods**). Thus, we wondered if post-stress sSPN hyperactivity could cause altered choices due to a reduction in the dimensionality of the

decision-space, similar to the model in Fig. 3b where optogenetic excitation reduced the decision-space in controls.

Our analysis supports this. Animals made decisions in the cost-benefit conflict task more quickly after stress, as if the task were less

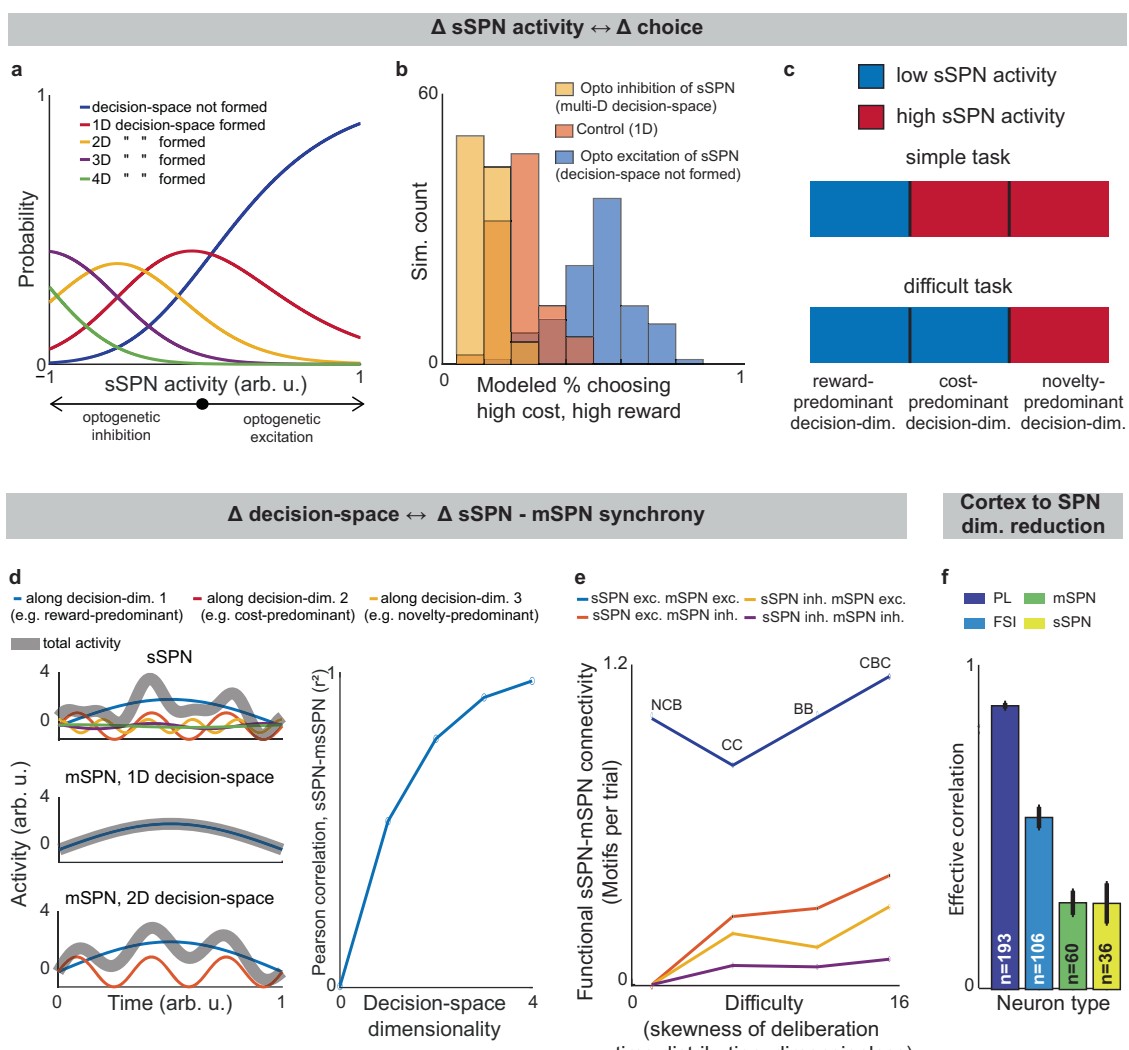

**Fig. 3 | Evidence of decision-space formation in sSPNs and mSPNs. a** Changes to striosome (sSPN) activity, for instance during optogenetic manipulation, cause changes in the modeled number of decision-dimensions used to form decision-space. In the analysis, $b_{sSPN}$ in eq. 1 is incremented from −1 (low activity) to 1 (high activity). **b** More consistent choices are made at lower sSPN activity. 100 choices are simulated for each of the three decision-space scenarios. Modeled choices are made between a high-cost, high-reward option and a low-cost, low-reward option. The multi-D decision-space applies rules for both reward and cost, the 1D decision-space rules about only reward, and the "decision-space not formed" case applies neither. Then action values and choice are derived. **c** An sSPN subpopulation has reduced activity when it is used to form decision-space. Therefore, mean sSPN activity is different in simple tasks requiring low-dimensional decision-space versus difficult tasks requiring high-dimensional decision-space. **d** When a one-dimensional decision-space is formed, sSPN (top-left panel) and matrix (mSPN; middle-left) activities have low correlation over time. As more decision-dimensions are used to form the decision-space, sSPN and mSPN activities increasingly correlate (bottom-left). Pearson's correlation between sSPN and mSPN activities (right

panel). **e** Experimental data showing that task difficulty (measured through skewness of the deliberation time distribution, (dimensionless units) increases with functional sSPN-mSPN connectivity (measured through Granger causality) during a decision. Tasks: NCB = non-conflict cost-benefit (14 sSPNs, 260 mSPNs), CC = cost-cost (46 sSPNs, 400 mSPNs), BB = benefit-benefit (83 sSPNs, 1246 mSPNs), CBC = cost-benefit conflict (84 sSPNs, 717 mSPNs). The CBC task has significantly more motif counts per trial (p < 0.003, significance assessed via a permutation test comparing actual motif counts to those derived from repeatedly shuffling the trial data). Source data are provided as a Source Data file. **f** Experimental data show that simultaneously recorded striatal projection neurons (SPNs) have less correlation (measured as effective correlation) than fast-spiking interneurons (FSIs) or pre-limbic cortical neurons. This indicates dimensionality reduction from cortical neurons to SPNs. Significances of difference from cortical neurons: FSI p < 10⁻¹⁸, mSPN p < 10⁻⁴⁵, sSPN p < 10⁻⁶³ (two-sided two-sample t-test for each comparison). 'n' indicates the number of sessions. Error bars show the mean squared error between the sessions. Source data are provided as a Source Data file.

difficult (Supplementary Fig. 4a–d, **Defining decision difficulty by task, Methods**). Meanwhile, after stress, choices involving both reward and cost no longer had more functionally connected sSPNs and mSPNs than the simple tasks, suggesting a change to the decision-space as well as a general shift in activity. In fact, synchrony was similar across tasks and to the simple tasks for controls (Supplementary Fig. 4e–i). Thus, after stress, the rodents both showed both neural signatures aligned with a low-dimensional decision-space and choices aligned with processing of information in a low-dimensional way.

The inability to form a high-dimensional decision-space can also explain a counterintuitive finding that stress causes rodents to prefer a reward-cost combination over a reward presented without cost (Supplementary Fig. 4j, **Changes to choice after adding cost to a reward offer, Methods**). In classic economic theory, the addition of cost to a reward typically makes a commodity less attractive[50] and thus our observation cannot be readily explained. In contrast, the decision-space model offers a simple explanation: cost can increase offer attractiveness in instances where cost level causes a transition from a default low-dimensional decision-space to a higher-dimensional

decision-space, as we hypothesized is the case after stress in Fig. 4b, c. In these cases, the rules encoded by mSPNs can assign a higher value to accepting versus avoiding an offer when reward and cost are considered rather than only reward (Fig. 4d, e). In other cases, decisions are predicted to resemble those predicted by classic theory, for example in cases where either a one-dimensional (as in Fig. 4d) or two-dimensional (Fig. 4e) decision-space is used across cost levels.

Cortex→FSI connectivity is impacted by chronic stress[3] (Supplementary Fig. 4k–r, **Analyzed cortex-FSI connectivity** and **Modeled cortex-FSI connectivity after stress, Methods**), leading to hyperactive sSPNs. Our model suggests that this causes the formation of a lower-dimensional decision-space (Fig. 4f, g), which would lead to lower variance and higher mean activity of SPN subpopulations (Supplementary Fig. 5). Notably, the cortex→FSI connection is also impacted in Huntington's disease and aged rodents[10], raising the possibility that an inability to form a high-dimensional decision-space during difficult decisions is a feature of multiple health conditions. Supporting this hypothesis, a model where Huntington's disease and aged subjects use a lower-dimensional decision-space produces action values that follow the trend of experimental choice (Fig. 4h, **Effect of decision-space on choice, Methods**).

Interestingly, our model expects a low dimensional decision-space to be beneficial in disorder conditions. A feature of chronic stress[3] and schizophrenia[51] is reduced cortical signal-to-noise ratio (SNR) and disrupted cortical signaling. In these conditions, a low dimensional decision-space is theoretically optimal because only the highest-priority decision-dimensions carry enough signal to outweigh the drawback of noise (Supplementary Fig. 6, **Effect of cortical SNR on choice, Methods**). Our analysis raises the possibility that FSIs help steer the circuit towards a helpful decision-space in disorders.

## Functional roles of dsSPNs, isSPNs, dmSPNs, and imSPNs

sSPNs and mSPNs have been found to be distributed between the direct and indirect pathways of the striatum[24,52,53], with dopamine release differently affecting sSPNs versus mSPNs and also dSPNs versus iSPNs[20,40,54]. Various neuropsychiatric disorders are associated with disturbed sSPN versus mSPN[2] and direct versus indirect pathway balances[33]. Thus, the compartments likely play distinct functional roles in decision-making, including in disorders.

Because dsSPNs connect to dmSPNs through daSNC in our model, and isSPNs to imSPNs through GPe and daSNC, two

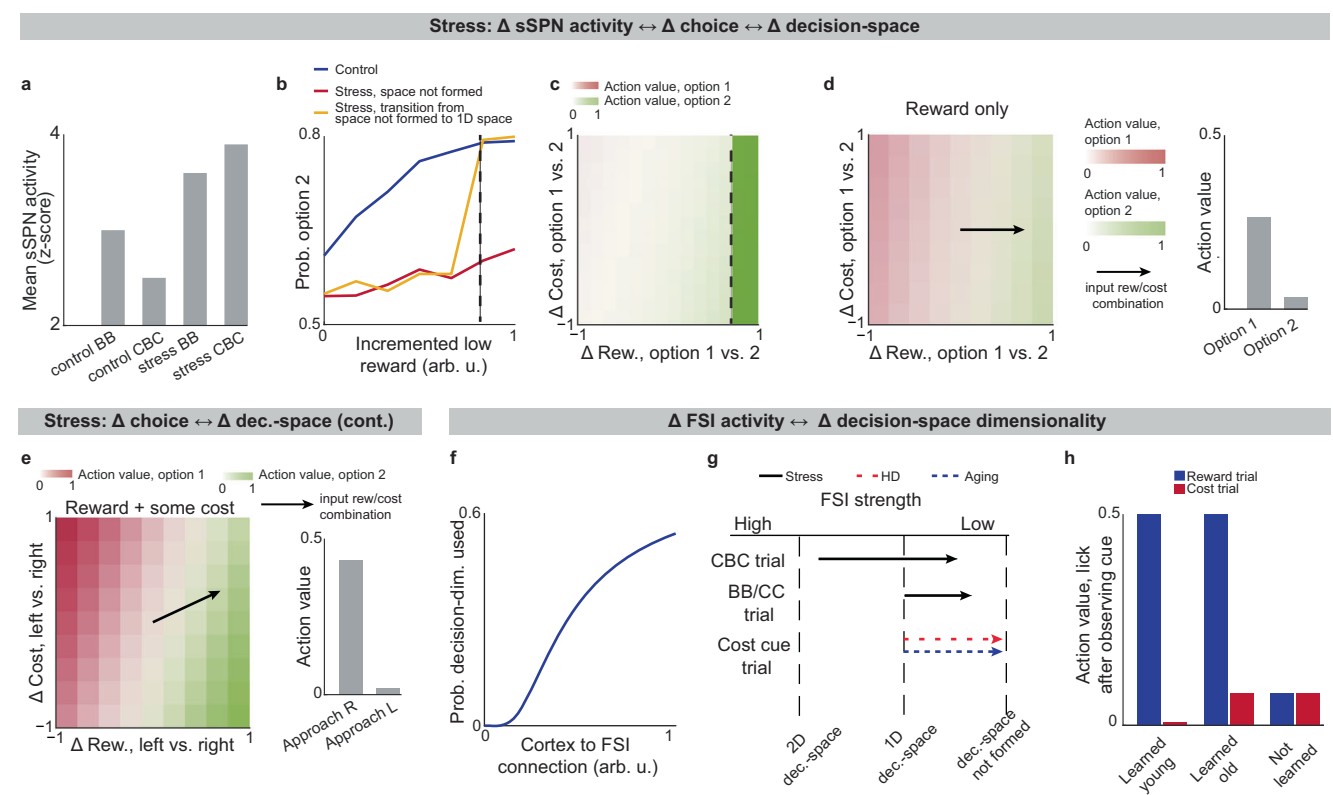

**Fig. 4 | Lower-dimensional decision-spaces are produced in stress, aging, and Huntington's disease, affecting decisions.** Data was taken from experiments where rodents performed the cost-benefit conflict (CBC) task, in which rodents had to select between a high cost-high reward option and a lower cost-lower reward option. Two behavioral control tasks were also used: the benefit-benefit (BB) task, in which rodents selected between a high reward option and a low reward option (equal and minimal cost for both), and the cost-cost (CC) task, in which rodents selected between a high cost option and a low cost option (equal and minimal reward for both). **a** Summary of the experimental finding that striosome (sSPN) activity is increased after stress, especially during the more difficult CBC task. **b, c** Modeled psychometric functions (**b**) and action values across experimental conditions (**c**) for a rodent in a T-maze task after chronic stress. The circuit forms a 1D decision-space only after reward exceeds a critical concentration (dashed line). At this point, the stress-group rodents switch from choosing the options roughly evenly to most often turning right towards the lower-cost option. Psychometric functions resemble experimental decision-making data. **d, e** After stress, a low-dimensional decision-space may be used by default (**d**) but a higher-dimensional decision-space may sometimes form during difficult tasks, for example those with both reward and cost (**e**). This leads to the counterintuitive result that adding cost to an offer (vectors on colormaps) can increase its action value (bar plots). **f** Cortex→FSI (fast-spiking interneuron) connection strength affects decision-space dimensionality by altering the propensity of a decision-dimension to form decision-space. **g** Cortex→FSI connection strength is reduced in stress, Huntington's disease, and aging. This leads to lower-dimensional decision-space. **h** Model of an operant conditioning task in Friedman et al (2020). Modeled action values for licking versus not licking in an operant conditioning task. There are two tasks: 1) responding to a reward cue by forming a decision-space from a reward-predominant decision-dimension, and 2) likewise for cost. "Learned young" succeeds at tasks 1 and 2, "Learned old" succeeds at task 1 but not task 2, and "Not learned" at neither task. Resembles experimental licking rates.

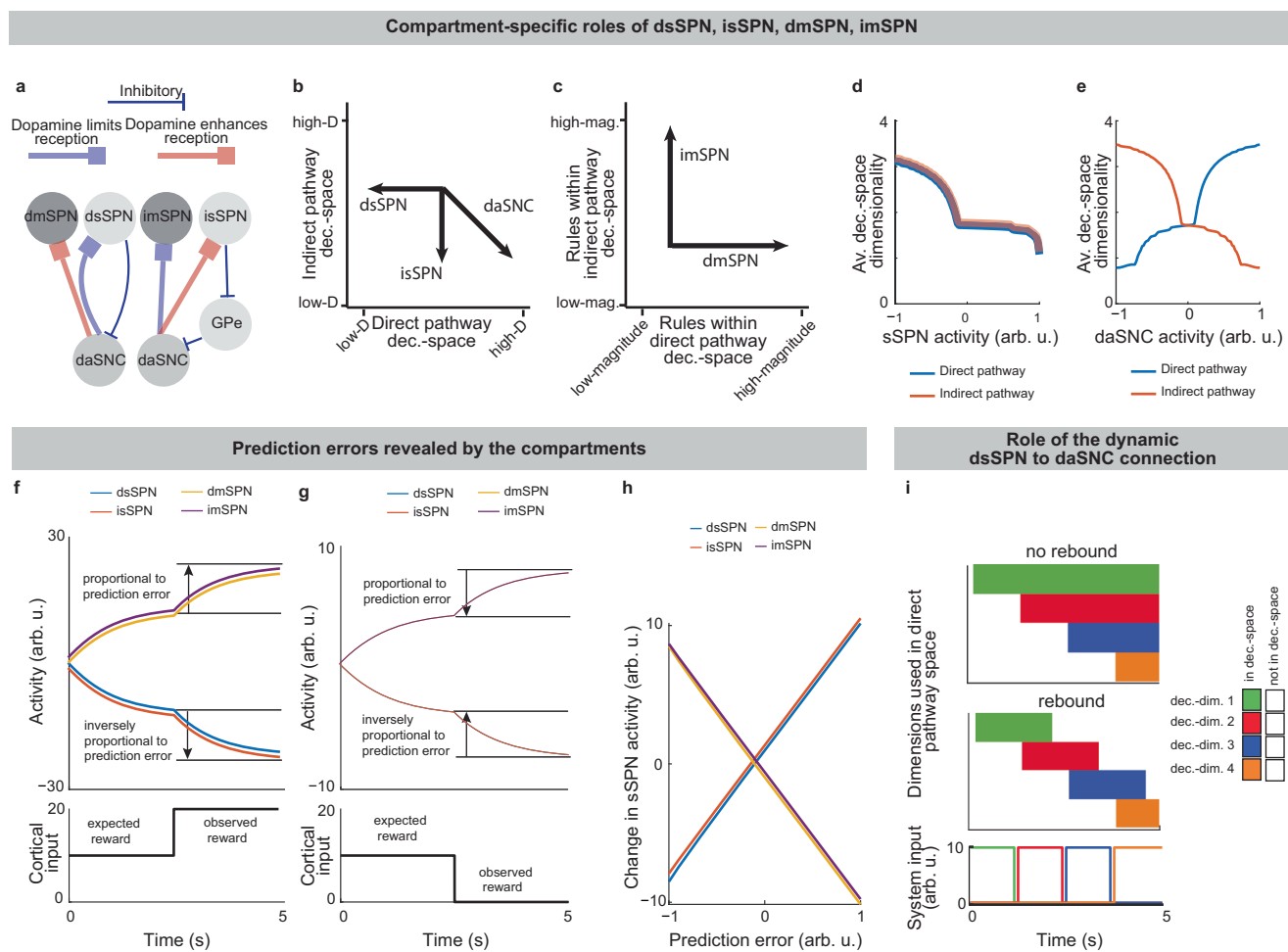

**Fig. 5 | dsSPNs, isSPNs, dmSPNs, and imSPNs serve unique roles in constructing and using the decision-space. a** Direct-pathway striosomes (dsSPNs) inhibit dopaminergic neurons of the substantia nigra compacta (daSNC) subpopulations. Conversely, indirect-pathway striosomes (isSPNs) disinhibit daSNC via globus pallidus externus (GPe). Dopamine lengthens direct-pathway matrix (dmSPN) upstates, enhancing cortical signal reception, and shortens indirect-pathway matrix (imSPN) upstates, limiting it. For connectivity details, see Supplementary Table 1. **b** This connectivity creates two parallel decision-spaces: one direct, one indirect. Active circuit elements bias the dimensionality of one or both decision-spaces. **c** dmSPN activity amplifies information along direct-pathway decision-dimensions, and imSPN activity does so for indirect-pathway dimensions. **d** Simulations show that as overall sSPN activity changes, the dimensionalities of direct and indirect decision-spaces change in the same direction. For instance, low sSPN activity might lead to actions based on many decision-dimensions, while high sSPN activity might base actions on only reward or cost. **e** Conversely, when overall daSNC activity changes, the dimensionalities of direct and indirect decision-spaces change in opposite directions. High daSNC activity might lead to actions based on

many direct-pathway dimensions or refraining based on only cost (potential impulsivity). Low daSNC activity might result in actions based on only reward or refraining based on many indirect-pathway dimensions (potential low motivation). **f, g** SPNs signal prediction errors. Cortical input (e.g., a bell) predicts a reward, and the actual reward (e.g., chocolate milk) at 2.5 s is higher (**f**, positive error) or lower (**g**, negative error) than predicted. Specific dsSPN, isSPN, dmSPN, and imSPN subpopulations related to reward reflect this error. **h** Simulations varying expected versus observed reward (−1 to +1) show these SPN populations change activity roughly proportionally to the prediction error. **i** 1.25 s pulses of input from the cortex (bottom panel) are applied along the four decision-dimensions in succession, testing the circuit's ability to rapidly prioritize or deprioritize decision-dimensions. A dynamic dsSPN→daSNC connection allows rapid de-prioritization of unneeded dimensions via a "rebound" effect, where daSNC activity surges above baseline after dsSPN inhibition. This rebound is absent in a "no rebound" scenario. The mechanism involves the weakening with prolonged sSPN inhibition. Then, when sSPN finally signals a dimension's unimportance, daSNC's reception of this signal is enhanced.

decision-spaces are formed in parallel, one related to each pathway (Fig. 5a, Supplementary Note 6). Thus, based on the functional roles we assign to the pathways, the circuit uses a direct pathway decision-space to determine whether to perform an action and an indirect pathway decision-space to determine whether to refrain from it. dsSPNs influence the direct pathway decision-space while isSPNs influence the indirect pathway decision-space, and dmSPNs promote actions while imSPNs discourage actions (Fig. 5b–e, **Modeling time-variant input, Methods**). The circuit uses these compartment-specific mechanisms to calculate which actions should be performed with one set of decision-dimensions and calculate which actions should not be performed with another.

The answers to these questions might overlap. For instance, the direct-pathway and indirect-pathway space should provide divergent answers to the value of consuming cocaine based on the dimensions they prioritized; the former, focusing on reward, might assign it great value while the latter, focusing on cost, might assign great value to not consuming it. When balances between the direct versus indirect pathways and striosome versus matrix changes, this calculus changes. For example, it has been found that dopamine release is enhanced to sSPNs versus mSPNs after cocaine administration[55] and simultaneously dSPNs are enhanced in the short-term and iSPNs over longer-term scales of time[56]. This might lead to a high-dimensional direct-pathway

decision-space but pruning of ordinarily important decision-dimensions from the indirect pathway decision-space, producing a reduction of nuance in determining when to avoid actions and heightened impulsivity.

Indeed, the model's interpretation of the two parallel decision-spaces offers an intuitive explanation for a range of experimental observations on the direct versus indirect pathway. For example, our model replicates the experimental observations that increased dopamine leads to riskier and quicker decisions and preference for nearby offers, in time or physical proximity, compared to distant offers (Supplementary Fig. 7, Supplementary Table 7, **Effect of dopamine on action/inaction values** and **Effect of decision-dimensions on choice, Methods**).

### Prediction error encoding is an emergent property of the model

A range of experimental studies have shown that SPN activities track prediction errors[12,57]. This observation has led to hypotheses that SPNs encode a function related to prediction errors in a reinforcement learning framework[12]. The decision-space model offers a different explanation. In our model, the weights from cortical neurons to SPNs naturally separate cortical information by their associations. For instance, if a bell tends to sound when a subject drinks chocolate milk, both stimuli, even if they arrive from different cortical sources, will likely be mapped to the same reward-related decision-dimension as synapses adjust per Oja's rule. Less reliable cues are expected to develop mappings with smaller weights. Therefore, the activities of reward-related SPNs may rise when cues predicting rewards appear and fall when cues predicting less reward appear. A sudden change to reward information, for instance a predictive cue, should thus lead to a sudden change in the activity of an SPN subpopulation related to reward, and this change in activity should resemble a prediction error (Fig. 5f–h). Thus, in contrast with the more traditional interpretation where SPNs internally encode a temporal difference value function, the decision-space model suggests that the mapping of cortical activity to the basis of striatal decision-dimensions is sufficient to track prediction error in many cases, without additional computational work performed by the SPNs.

Both interpretations can be used to explain much of the experimental evidence, although the interpretation of the decision-space model may more closely align to recent experimental data (Supplementary Tables 2, 3). Our model also may provide a functional rational for the observation that separate SPNs encode data along different informational axes[12,14]. In fact, our model expects more of these axes to be uncovered by future work. We might expect, for instance, a cue predictive of the novelty of an object to produce an immediate change in activity of an SPN subpopulation related to novelty.

How might the circuit respond in cases where new information diverges sharply from expectations? Our model predicts that in these cases, sSPNs will signal to daSNC that a decision-dimension should be reprioritized, effectively adding or removing the dimension from the decision-space. Interestingly, the circuit has an inherent physiological mechanism to quickly transition away from a decision-space that is no longer optimal. Experimental evidence has identified rebounds in daSNC activity[21] and striatal dopamine release[18] after sSPN optogenetic stimulation. Our model suggests that these observations are part of a system by which the circuit can rapidly de-prioritize a decision-dimension after a negative prediction error (e.g. less reward than expected). Thus, the circuit is able to quickly shift to a more helpful decision-space (Fig. 5i). During learning, these shifts between decision-spaces will occur when the subject is surprised to encounter a decision-dimension that does not align with their experience. For instance, during a reinforcement learning T-maze task, the decision-space will shift when the two options are suddenly reversed (Supplementary Fig. 7j–l). These shifts occur during the trials where there are the

largest RPE. So, shifts in the decision-space might not only align with RPEs in magnitude, but might be coordinated with RPEs over time.

## Discussion

We found evidence, through experimental literature and our analysis of neural recordings, to support our hypothesis that modeled physiological patterns in SPN activity (the decision-space) can be used to predict patterns in decision-making, and vice versa. This supports our model of the roles of striosomes and matrix neurons of the direct and indirect pathways in context-dependent decision-making.

Due to the circuit's important role in decision-making processes, including in neuropsychiatric disorders, our model provides a framework with which to study decision-making phenomena commonly observed in psychiatry. An important prediction of our model is variance in context-dependent decisions, between individuals and over time (Fig. 6a–d, Supplementary Fig. 8a–e, **Effects of sSPN, LHb, and daSNC activity on choice profiles, Methods**). Individual differences in decision-making as a function of disorders, as seen in the experimental literature[58-60], could arise in cases where there are slight differences in activity of the circuit we model, leading to similar decision-making phenotypes only when a similar decision-space is formed. Daily variance in decision-making, a common observation in pscyhology[61], could arise from daily variance in circuit activity, causing day-to-day variability in the decision-spaces formed most often. Further, differences in circuit activity may explain the established inter-individual differences in the severity of psychiatric disorder symptoms observed during decision-making[62,63]. Individual differences in disorder susceptibility could arise from reliance upon or avoidance of a decision-space that leads to extreme decision-making tendencies (e.g. extremely action-heavy, extremely risk-averse) when combined with abnormal action value rules in mSPNs (Fig. 6e and Supplementary Fig. 8f).

Our model serves as a framework for forming hypotheses about changes to the circuit across days and weeks, including during neuropsychiatric disorder progression. Our model expects the circuit to adapt between trials as it adjusts to more frequently form a preferred decision-space (Fig. 7a–f, **Effect of initial circuit activity on future trials, Methods**). So, vulnerability or resilience to disorders can be framed as an adaptation that is favorable (e.g., to adeptly form a high-dimensional decision-space) or maladaptive (e.g., to only form a low-dimensional decision-space, regardless of decision context). Depending on its initial activity, a modeled circuit can adapt to reach very different activity, leading to disposition to either a high- or low-dimensional decision-space (Fig. 7g and Supplementary Fig. 9a). Thus, differences in the circuit before exposure to a traumatic event, for instance, may explain why two subjects that encounter the same traumatic event do not always develop the aberrant decision-making symptoms of post-traumatic stress disorder (PTSD)[64]. It may also shed light on the neural processes underlying incubation of fear[65] and incubation of craving[66], where disorders progress over the span of weeks or months, even when the traumatic event or addictive substance does not reappear (Supplementary Fig. 9b–d, **Effect of altered advantage score on future trials, Methods**). By representing the role of SPNs in a compartment-specific way, our model facilitates understanding of disorders that affect striosomes and matrix differentially.

Our model carries several limitations, including its limited focus on a dorsomedial striosomal circuit and certain physiological assumptions (Supplementary Notes 1-4). We limit our focus to a specific circuit that has been implicated in decision-making, rather than attempting a unifying theory of basal ganglia function or decision-making encoding across the brain (Supplementary Note 2). While we demonstrate alignment to the existing experimental evidence in Figs. 2 and 3 and Supplementary Tables 2–7, future experiments (outlined in Supplementary Fig. 10) will be required to confirm the

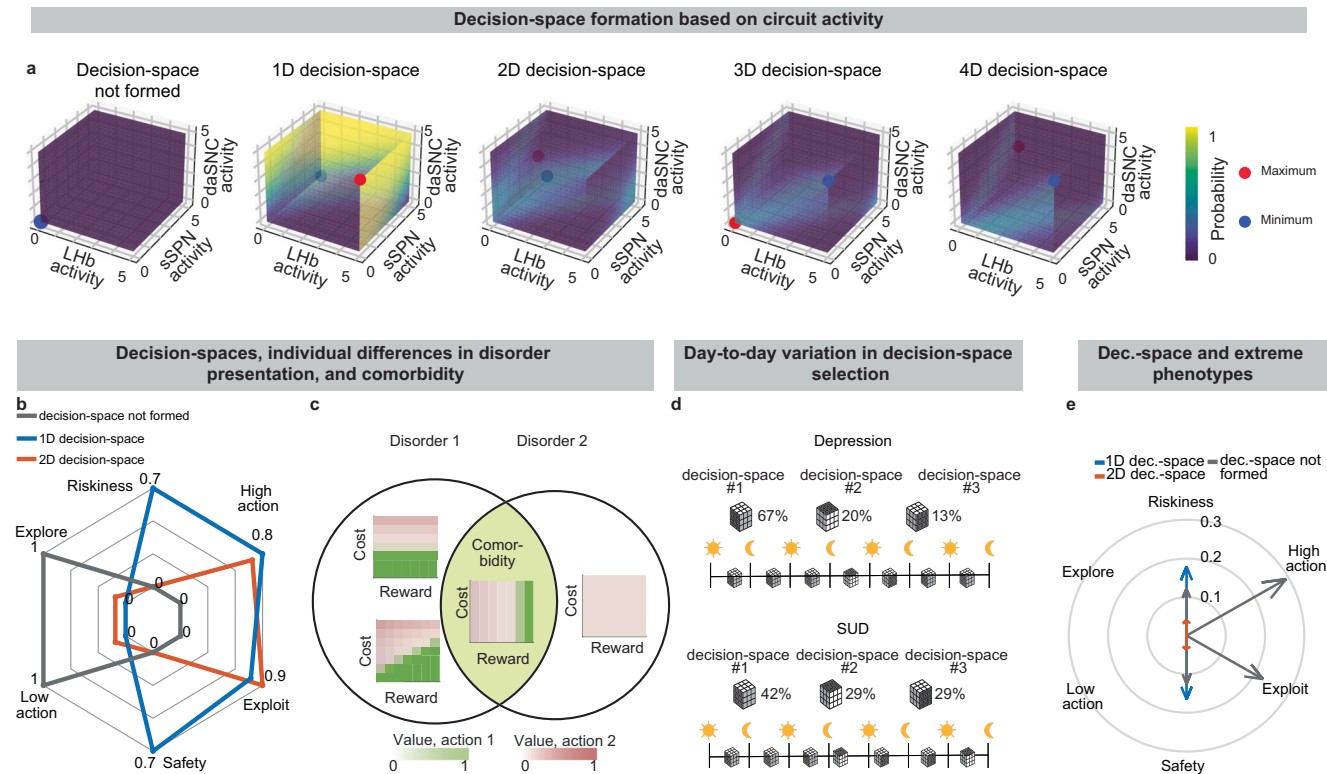

**Fig. 6 | Differences in the decision space could explain comorbidity, individual differences, and daily variations. a** A decision-space is simulated for various striosome (sSPN), dopaminergic neurons of the substantia nigra compacta (daSNC), and lateral habenula (LHb) activities. The circuit forms a bias towards certain decision-spaces over others, but different decision-spaces can form at the same circuit activity. **b** Summaries of overall decision-making profiles across trials in a T-maze task, scored in six ways, showing that transitions between decision-spaces can lead to very different decisions. A disorder could produce a bias, for instance, towards low-dimensional decision-spaces, which would in turn alter the decision-making profile. Scores are formed by quantifying the trend of action values across reward and cost levels. Riskiness/safety: treatment of high-reward, high-cost (or low-cost, low-reward) levels; high/low action: tendency towards high

(or low) action values; exploit/explore: tendency to focus on one action versus many (see Supplementary Fig. 8c). **c, d** Differences in circuit activity between individuals could lead to decision strategies observed at different rates, as is the case in individuals with disorder comorbidity (**c**). Further, day-to-day shifts in circuit activity shifts could cause stark differences in decision strategies between days (**d**). Cartoon shows the hypothetical use of different decision-spaces for depression and substance use disorders. **e** Certain decision-spaces more often lead to action values that are extreme (ratio formed over 1000 simulations, as scored using the metrics in **c**), a feature of disorders. Vector length corresponds to outlier rate (proportion of scores for each group that fall within the top 10% of observations across all groups).

assumptions we make. Despite these limitations, our model has demonstrated success in relating neural activity to decision-making across a range of behavioral tasks and has the power to explain a range of phenomena, from neural processes to psychiatric observations. Additionally, we expect it to serve as a foundation for future work on other brain regions like substantia nigra and other pathways such as are formed by arkypallidal neurons of the GPe (Supplementary Note 7). Finally, it adds precision to other models (Supplementary Note 8), including reinforcement learning models of the basal ganglia (Supplementary Note 9).

## Methods
### Outline
- **Decision-dimensions and decision-space.** Explanation of the foundational concept of the model.
- **Analyzed instances of the model.** We conduct our analysis using three instances of the conceptual model. In the following sections, we formally define the model in each instance, then detail the methods behind our related analyses.

  **Instance 1: full connectivity and feedforward.** Related to Figs. 2–4, 7, Supplementary Figs. 2–4, 7, 8. Used to link neural activity, the decision-space, and choice.

  **Modeled circuit manipulation using Instance 1.**
  **Instance 2: sparse connectivity and feedforward.** Related to Supplementary Fig. 5. Used to demonstrate how a large network might encode the decision-space.
  **Modeling SPN encoding of data, using Instance 2.**
  **Instance 3: full connectivity and dynamics.** Related to Fig. 5. Used to demonstrate how the decision-space might form over time.
  **Modeling time-variant input, using Instance 3.**
- **Shifts in the decision-space in a reinforcement learning task.** Reinforcement learning simulations to analyze the differences between the decision-space model and the classical reinforcement learning model of the basal ganglia. Related to Supplementary Fig. 6j–l.
- **Movement of circuit activity across multiple trials.** An extension of the model to view possible changes of the circuit between trials in the context of decision-space. Related to Fig. 7, Supplementary Fig. 9.
- **Reasoning behind the FSI model.** Mathematical choices made in the model of fast-spiking interneurons (FSIs).
- **Inferring decision-space from SPN activity and choice.** A method we designed in which decision-space can be inferred from experimental data. Related to Supplementary Fig. 2q, r.

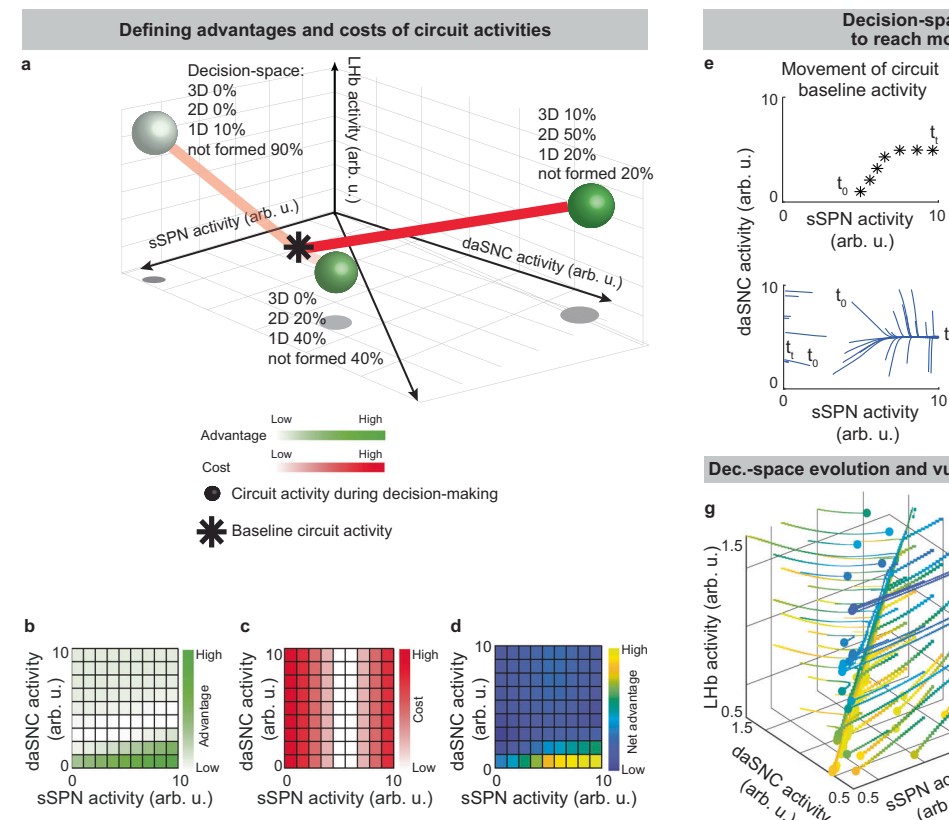

**Fig. 7 | Circuit adapts between trials to form preferred decision-spaces, leading to disorder progression. a** Cartoon illustrating how the modeled circuit adjusts to facilitate construction of preferred decision-spaces ("advantage") while limiting large changes in circuit activity during decision-making ("cost"). Possible advantages and costs (whose difference is "net advantage") are shown at three circuit activities (balls). The decision-space is formed differently at each circuit activity (annotations), leading to differences in advantage. Between trials, the circuit adjusts its baseline activity (i.e. resting activity outside decision-making) in the direction of highest net advantage. **b–d** Simulated advantages (**b**), costs (**c**), and net advantages (**d**) shown across combinations of striosome (sSPN) and dopaminergic neurons of the substantia nigra compacta (daSNC) activity. **e** sSPN and daSNC activity adjusts between trials to best form preferred decision-spaces. Due to this,

the circuit adjusts its activity from an initial baseline activity ($t_0$) to other activities associated with the required decision-spaces ($t_t$). **f** Similar to **d** but for sSPN, lateral habenula (LHb), and daSNC activities, showing a simulated circuit where it is most advantageous to have high sSPN activity during simple choices. The trend in the continuous 3D decision-space is visualized using isosurfaces. **g** Similar to **e** but for sSPN, LHb, and daSNC activities. Dots show ending circuit activities (i.e. $t_t$) and beginnings of lines possible initial circuit activities (i.e. $t_0$). Depending on initial activity, the circuit may increasingly use or disregard a decision-dimension when forming decision-space. Right panel shows trajectories from three starting circuit activities, two of which lead to the decision-dimension commonly being used to form decision-space (e.g. resilient subjects) and one which leads to it commonly being disregarded (e.g. vulnerable subjects).

---

- **Testing the model through analysis of neural data.** Analysis of neural data which supports our model. Related to Fig. 3e, f, Supplementary Figs. 3a–n, 4a–p.

## Decision-dimensions and decision-space

The physiologies of the circuit elements produce two abstractions which we use, for convenience, throughout our work:

- A decision-dimension is an axis of the coordinate system with which SPNs (dsSPNs, isSPNs, dmSPNs, imSPNs; see Table 1 for anatomical definitions) encode data projected from the cortex. A decision-dimension is equivalent to a principal component of cortical activity. In our analysis, separate groups of SPNs encode data along the first, second, third, and fourth principal components. (Arbitrarily, we do not consider principal components beyond the first four). Each of the dsSPN/isSPN/dmSPN/imSPN subgroups have neurons corresponding to each of the four principal components.
- The decision-space is the mathematical space formed by mSPNs (both dmSPNs and imSPNs) after dopamine signaling from daSNC. Modeled dopamine signaling determines whether to include or exclude neurons encoding each decision-dimension

during a decision. Thus, from the mathematical space formed by all decision-dimensions, a mathematical subspace (i.e. the "decision-space") is formed with which to define action values.

We use the prefix "decision" because the circuit uses decision-space, formed from a basis of decision-dimensions, to define action values during decision-making.

In our analysis, we give "reward," "cost", "novelty", and "location" as examples of informational axes that might roughly correspond to decision-dimensions. These examples follow from the intuition that: 1) these axes seem to be roughly orthogonal, as are decision-dimensions; 2) these axes are continuous, making representation in decision-dimensions intuitive. Additionally, the experimental evidence suggests that some striosomes or daSNC selectively encode reward[12,14,15,35], cost[12,14,15,35], or novelty[48]. Finally, a location-predominant decision-dimension could explain the experimental observations modeled in Supplementary Fig. 7i.

In reality, we would not expect any informational axes to align perfectly with any decision-dimension, although some axes should logically align with certain decision-dimensions better than others. (For instance, there must exist a decision-dimension that aligns with

reward level the most out of any decision-dimension.) Also, note that while we refer to, for instance, a "reward-predominant" decision-dimension, this does not mean that it is expected to track only reward. Instead, many other axes of data might roughly align with it, including reward level. For example, a decision-dimension that roughly corresponds to reward might also be found to track information about the color green, or the pleasantness of a smell, or the sound of a bell. We can imagine that a variety of other information might also map to decision-dimensions. For example, information about social dominance, the desire to replicate, exploration, and morality might be found along different decision-dimensions.

In our equations (e.g. eq. 2), we use the index $i$ to refer to the sSPN, daSNC, or mSPN population corresponding to a certain decision-dimension (out of $q$ total decision-dimensions in each of the direct and indirect pathways). Meanwhile, we use the index $j$ to refer to a certain action (out of $k$ total actions).

## Analyzed instances of the model

The general case of the model (although not formally used for analysis) is a dynamic network of cortical neurons, FSIs, dsSPNs, isSPNs, GPi, LHb, daSNCs, dmSPNs, imSPNs, and mSPN-projecting neurons which encode action values.

We conduct our analyses using three instances of this general case, which are each equivalent to the general case under the specific conditions we outline. The three instances, tailored to our various analyses, each allow for a different mathematical simplification. This allows us to conceptually and formally define the instances individually in a way that is intuitive and relates directly to our analyses.

1) Instance 1 has full cortex→FSI→SPN connectivity and constant activity in each circuit element throughout the decision. In this instance, the model can be defined equivalently using a smaller set of network elements and a feedforward network. See **Instance 1: full connectivity and feedforward**.
2) Instance 2 has constant activity in each circuit element throughout the decision. In this instance, the model can be defined equivalently using a feedforward network. See **Instance 2: sparse connectivity and feedforward**.
3) Instance 3 has full cortex→FSI→SPN connectivity. In this instance, the model can be defined equivalently using a smaller set of network elements. See **Instance 3: full connectivity and dynamics**.

## Instance 1: full connectivity and feedforward

In this section, we describe the instance of the model where each cortical neuron projects to each FSI, each FSI projects to each SPN (for dsSPN, isSPN, dmSPN, imSPN), and each cortical neuron projects to each SPN. Additionally, cortical input to the system does not change over time, and the activities of other circuit elements do not decay over time.

This instance of the model leads to a convenient formation of the model as a circuit of fewer elements (one FSI, one dsSPN, isSPN, dmSPN, and imSPN per the four decision-dimensions), and no time component. In this section, we frame this instance mathematically and then describe our related analysis. See Fig. 1c for a circuit diagram.

**Input: cortical activity**. During a decision, a vector of cortical input $\mathbf{x}_P \in \mathbb{R}^{p \times 1}$ enters each pathway $P$ in the network ($\mathbf{x}_{\text{direct}}$ to direct pathway SPNs and $\mathbf{x}_{\text{indirect}}$ to indirect pathway SPNs). The elements of $\mathbf{x}_P$ are the activities of $p$ cortical neurons. Each neuron encodes a different sensory input.

**Outputs**. We use this instance of the model to examine: 1) the activities of the circuit elements depending on the activities of other circuit elements (Fig. 2, Supplementary Fig. 2a–i); 2) the modulation of mSPNs by dopamine (i.e. decision-space, Fig. 2a, b); 3) action values given

decision-space (Supplementary Fig. 2j–l), and 4) choice given action values (Supplementary Fig. 2m–p).

1) The activities of circuit elements during a decision are related to each other based on anatomically realistic connections (eqs. 1–3, 5, 7).
2) A decision-space is formed probabilistically. The probability of a given decision-dimension $i$ being used during a decision is equivalent to the activity of daSNC (see eq. 2), which ranges from 0 to 1. Probabilities are realized in the connection from daSNC to mSPN (see eq. 3), when each decision-dimension is probabilistically assigned a weight (in most analyses, either 0 or 1). Decision-space is defined as the space formed from the basis of decision-dimensions that were not assigned a weight of 0.
3) Action value is derived based on mSPN activity during a decision.
4) Choice is derived from action values. Action values are treated as Merton processes[44] using eq. 12. Several possible actions are assigned action values and the corresponding process that hits the threshold first is enacted.

**Defining FSI activity**. FSI activity, $c_P$, is set to the magnitude of $\mathbf{x}_P$ for each pathway, multiplied by a weight of cortex→FSI connection $a_{FSI}$, plus an additive shift $b_{FSI}$:

$$c_P = a_{FSI} \cdot ||\mathbf{x}_P|| + b_{FSI} \tag{5}$$

where:
- $c_P$ is relative activities of FSIs that project to SPNs of pathway $P$ (activity arb. u.)
- $a_{FSI}$ is the weight of cortex→FSI connection. Similar for both $P$. (dimensionless)
- $\mathbf{x}_P$ is the activities of cortical neurons that project to SPNs of pathway $P$. (activity arb. u.)
- $b_{FSI}$ affects the relative activity of all sSPN neurons. Similar for both $P$. (activity arb. u.)

In the current instance of the model, there are 2 FSIs, one that receives input from $\mathbf{x}_{\text{direct}}$ and projects to dsSPNs and dmSPNs, and the other that receives input from $\mathbf{x}_{\text{indirect}}$ and projects to isSPNs and imSPNs.

For use in our analysis, see https://github.com/dirkbeck/DM_space_model/blob/main/algorithmic_model.m.

**Defining sSPN activity**. To get the activities of sSPNs in each pathway, $\mathbf{x}_P$ is normalized via division by $c_P$ and multiplied by $\mathbf{W}_P \in \mathbb{R}^{p \times q}$, which linearly transforms and reduces cortical input from the $p$-dimensional coordinate space of cortex to the smaller $q$-dimensional coordinate space of sSPN. In the sSPN coordinate space, each coordinate is a principal component of a training set of historical cortical input across $n$ time steps $\mathbf{X}_P \in \mathbb{R}^{n \times p}$ (uncorrelated, for simplicity). For each pathway, $\mathbf{W}_P$ contains the truncated first $q$ columns (corresponding to the first $q$ principal components) of $\mathbf{W}_{\text{full}, P} \in \mathbb{R}^{p \times p}$ after the decomposition $\mathbf{X}_P \mathbf{X}_P^T = \mathbf{W}_{\text{full}, P} \mathbf{\Lambda} \mathbf{W}_{\text{full}, P}^T$ is made to obtain the full principal component matrix. There is dimensionality reduction on the order of ~100 times from cortex to SPNs (Supplementary Note 3), so $q \ll p$. Note that in our analysis using the current instance of our model, $\mathbf{X}_P$ is not explicitly generated because we specify the inputs to the system in terms of the coordinate space of decision-dimensions.

For each pathway, the components of an sSPN activity vector $\mathbf{s}_{\text{sSPN}, P} \in \mathbb{R}^{q \times 1}$ each correspond to the activity of an sSPN circuit element. A constant $b_{\text{sSPN}}$, used in analyses where modeled sSPN activity is stimulated or inhibited, adjusts overall sSPN activity:

$$\mathbf{s}_{\text{sSPN}, P} = \frac{1}{c_P} \mathbf{W}_P^T \mathbf{x}_P + b_{\text{sSPN}} \tag{6}$$

where:

- $c_P$ is the relative activity of the FSI projecting to SPNs of pathway $P$ (activity arb. u.)
- $\mathbf{W}_P$ is a matrix of weights from cortical neurons to SPNs of pathway $P$. Each column is equivalent to a principal component of cortical activity. (dimensionless)
- $\mathbf{x}_P$ is the activities of cortical neurons that project to SPNs of pathway $P$. (activity arb. u.)
- $b_{\text{sSPN}}$ affects the relative activity of all sSPN neurons (activity arb. u.)

In the current instance of the model, activities are defined based on a feedforward network, so the simplification is made that sSPN activities are not affected by daSNC activities.

In the current instance, there is one sSPN per decision-dimension per pathway. So, there are $q$ dsSPNs and $q$ isSPNs. The dsSPNs receive input from $\mathbf{x}_{\text{direct}}$ and $c_{\text{direct}}$. The isSPNs receive input from $\mathbf{x}_{\text{indirect}}$ and $c_{\text{indirect}}$.

For use in our analysis, see https://github.com/dirkbeck/DM_space_model/blob/main/algorithmic_model.m.

**Defining GPi, LHb, and RMTg activities.** The GPi→LHb→RMTg→daSNC pathway performs a series of operations which influence RMTg activity RMTg, which is an input to daSNC activity in eq. 2. Weights $\mathbf{w}_{\text{GPi},P} \in \mathbb{R}^{2q \times 1}$, not necessarily positive, are combined with the activities of the $q$ dsSPNs and $q$ isSPNs, forming scalar representations of dsSPN or isSPN activity. $z_{\text{GPi}}$, $z_{\text{LHb}}$, and $z_{\text{RMTg}}$ terms reflecting the activities of those circuit elements are combined with this scalar representation:

$$\text{RMTg} = z_{\text{RMTg}} + z_{\text{LHb}} + z_{\text{GPi}} \; \cdot \; \mathbf{W}_{\text{GPi}} \; \cdot \; \begin{bmatrix} \mathbf{s}_{\text{sSPN, direct}} \\ \mathbf{s}_{\text{sSPN, indirect}} \end{bmatrix} \qquad (7)$$

where:

- $z_{\text{RMTg}}$ is an additive shift that affects relative RMTg activity (activity arb. u.)
- $z_{\text{LHb}}$ is an additive shift that affects relative LHb activity (activity arb. u.)
- $z_{\text{GPi}}$ is a coefficient that affects relative GPi activity (activity arb. u.)
- $\mathbf{w}_{\text{GPi}}$ is the weights of connection from sSPNs of pathway $P$ to the GPi neuron (dimensionless)
- $\mathbf{s}_{\text{sSPN},P}$ is the activity of sSPNs corresponding to decision-dimension $i$ and pathway $P$ (activity arb. u.)

This pathway contains one GPi element, one LHb element, and one RMTg element, as visualized in Fig. 1c. All sSPN elements project to GPi. GPi activity is an input to LHb, which after the $z_{\text{LHb}}$ addition, is an input to RMTg, which itself has a $z_{\text{RMTg}}$ addition. For simplicity, these series of operations are presented together in eq. 7.

For use in our analysis, see https://github.com/dirkbeck/DM_space_model/blob/main/algorithmic_model.m.

**Defining daSNC activity.** daSNC neurons incorporate the output of the GPi→LHb→RMTg→daSNC pathway with direct inputs from sSPN elements. There are $q$ sSPN elements of each pathway and $q$ daSNC elements corresponding to each pathway. For each pathway, the $i$th sSPN element connects to the $i$th daSNC element, but not to other daSNC elements (see Fig. 1c). These connections have weights $w_{\text{sSPN} \rightarrow \text{daSNC}, i, P}$ for $i = 1, 2, \ldots, q$. RMTg, on the other hand, connects to each daSNC element. Only for the indirect pathway, sSPNs connect to daSNC through GPe. (That is, $z_{\text{GPe, direct}} = 0$). The output of the $i$th daSNC element, constrained to between 0 and 1 via a logistic function,

captures the importance of a single decision-dimension:

$$\text{daSNC}_{i,P} = \frac{1}{1 + \exp(w_{\text{sSPN} \rightarrow \text{daSNC}, i, P} \cdot (s_{\text{sSPN}, i, P} + z_{\text{GPe}, P}) + \text{RMTg} - z_{\text{daSNC}, i, P})} \qquad (8)$$

where:

- $w_{\text{sSPN} \rightarrow \text{daSNC}, i, P}$ is the weight of connection from the sSPN corresponding to decision-dimension $i$ and pathway $P$ to the daSNC corresponding to decision-dimension $i$ and pathway $P$. The weight is fixed in this instance of the model. (dimensionless)
- $s_{\text{sSPN},i,P}$ is the activity of sSPNs corresponding to decision-dimension $i$ and pathway $P$ (activity arb. u.)
- $z_{\text{GPe},P}$ is the activity of GPe (activity arb. u.)
- $z_{\text{daSNC}, i, P}$ is an additive shift applied to the daSNC neuron corresponding to decision-dimension $i$ and pathway $P$ (activity arb. u.)
- RMTg is RMTg activity, as defined in eq. 7. (activity arb. u.)

This pathway is modeled using one neuron per decision-dimension per pathway. So, there are $q$ daSNC neurons that each receive projection from a dsSPN, and $q$ daSNC neurons that each receive projection from an isSPN. daSNC elements also receive input from RMTg.

For use in our analysis, see https://github.com/dirkbeck/DM_space_model/blob/main/algorithmic_model.m.

**Defining mSPN activity and the decision-space.** In each pathway, the decision-space is formed probabilistically. The conversion from daSNC activity to realization of the decision-space occurs in the connections from daSNC to mSPN. There are $q$ daSNC elements corresponding to each pathway and $q$ mSPN elements, and, in each pathway, the $i$th daSNC element connects to the $i$th mSPN element, but not to other mSPN elements (see Fig. 1c).

Like sSPNs, mSPNs encodes the cortical input normalized by an FSI and is transformed to a coordinate space of the first $q$ principal components. The difference is that for each of dmSPNs and imSPNs, a diagonal matrix $\mathbf{S}_P \in \mathbb{R}^{q \times q}$ is multiplied by the cortical input after transformation:

$$\mathbf{s}_{\text{mSPN}, P} = \frac{1}{c_P} \mathbf{S}_P \mathbf{W}_P^{\mathbf{T}} \mathbf{x}_P \qquad (9)$$

where:

- $c_p$ is the relative activity of the FSI projecting to SPNs of pathway $P$ (activity arb. u.)
- $\mathbf{S}_P$ is a diagonal matrix that applies dopamine release (via daSNC activity) to mSPN activity in pathway $P$. (dimensionless)
- $\mathbf{W}_P$ is a matrix of weights from cortical neurons to SPNs of pathway $P$. Each column is equivalent to a principal component of cortical activity. (dimensionless)
- $\mathbf{x}_P$ is the activities of cortical neurons that project to SPNs of pathway $P$. (activity arb. u.)

In the current instance, there is one mSPN per decision-dimension per pathway. So, there are $q$ dmSPNs and $q$ imSPNs. The dmSPNs receive input from $\mathbf{x}_{\text{direct}}$ and $c_{\text{direct}}$. The imSPNs receive input from $\mathbf{x}_{\text{indirect}}$ and $c_{\text{indirect}}$.

The diagonal elements of $\mathbf{S}_P$ are set probabilistically to either 1 (dimension in decision-space) or 0 (dimension not in decision-space) such that $P(\mathbf{S}_{P, \text{ii}} = 1) = \text{daSNC}_{i, P}$.

Thus, in the portions of our analysis where we set the activities of the $q$ daSNC elements to be equal, the decision-dimensions each have the same probability of being included in decision-space, i.e. $\text{daSNC}_1 = \text{daSNC}_2 = \ldots = \text{daSPN}_q = d$. In this case, we treat the probability of a certain decision-space dimensionality forming as a binomial

distribution:

$$P(m \text{ decision-dimensions used to form decision-space}) = \binom{q}{m} d^m (1-d)^{q-m} \quad \text{for } m = 0, 1, \ldots, q.$$

(10)

where:

- $q$ is the number of possible decision-dimensions
- $d$ is the (equal) probability that each decision-dimension is used to form decision-space

**Defining action value.** Action value (or, in the indirect pathway, inaction value) $v_{j,P}$ for each of $k$ potential actions is defined based on the activities of dmSPNs (or imSPNs). During this process, elements of a coefficient matrix $\boldsymbol{\beta}_P \in \mathbb{R}^{k \times q}$ are applied to mSPN activities for each decision-dimension, action, and pathway. Bias $\alpha_{j,P}$ is subtracted. Below, $\boldsymbol{\beta}_{j,P}$ is used to indicate row $j$ of $\boldsymbol{\beta}_P$.

$$v_{j,P} = \frac{1}{1 + \exp(-\boldsymbol{\beta}_{j,P}\, \mathbf{s}_{\mathbf{mSPN},P} - \alpha_{j,P})},$$

(11)

where:

- $\boldsymbol{\beta}_{j,P}$ is a matrix of weights from dmSPNs to downstream action value encoding neurons for the direct pathway, or imSPNs to downstream inaction value encoding neurons for the indirect pathway. (dimensionless)
- $\mathbf{s}_{\mathbf{mSPN},P}$ is the activity of sSPNs corresponding to decision-dimension $i$ and pathway $P$ (activity arb. u.)
- $\alpha_{j,P}$ is an additive shift corresponding to the neuron encoding action $j$ for the direct pathway or inaction $j$ for the indirect pathway. (activity arb. u.)

There is one neuron encoding each $v_{j,P}$. So, there are $k$ neurons encoding action values and $k$ neurons encoding inaction values. Each of these neurons receives projections from all mSPNs of the corresponding pathway.

**Defining choice.** $k$ Merton processes[44] are run to determine whether each action should be taken, and another $k$ to determine whether each action should be refrained from. Progress to choice for each action (or inaction), $Y_{j,P}$, is related to its corresponding action (or inaction) value $v_{j,P}$ and an uncorrelated Brownian component $dW_{j,P}$ scaled by a coefficient $\sigma$.

$$dY_{j,P} = v_{j,P}\, dt + \sigma\, dW_{j,P}, Y_{j,P}(0) = 0, \quad \text{where } W_{j,P} \text{ is a standard Wiener process}.$$

(12)

where:

- $Y_{j,P}$ is the progress to enaction of action $j$ in the direct pathway, and progress to refraining from action $j$ in the indirect pathway.
- $v_{j,P}$ is the action (or inaction) value corresponding to action $j$ and pathway $P$. (activity arb. u.)
- $\sigma$ is the coefficient of noise.

The time it would take to enact action $j$, $t_{\text{action},j}$, is defined as the first hit time of a threshold $h$ for process $j$ of the direct pathway:

$$t_{\text{action},j} = \min_t \{ t \mid Y_{j,\text{direct}} \geq h \}$$

(13)

The time it takes to exclude action $j$ from consideration, $t_{\text{inaction},j}$, is calculated similarly using the indirect pathway:

$$t_{\text{inaction},j} = \min_t \{ t \mid Y_{j,\text{indirect}} \geq h \}$$

(14)

The enacted action is the first to reach $h$, given that the corresponding inaction process has not first reached $h$:

$$\text{action} = \arg\min_{j \in J} (Y_j(t_{\text{action},j})), \text{ where } J \text{ is the subset of actions s.t. } t_{\text{action},j} < t_{\text{inaction},j}$$

(15)

where:

- $Y_{j,P}$ is the progress to enaction of action $j$ in the direct pathway, and progress to refraining from action $j$ in the indirect pathway.
- $t$ is time (s)
- $h$ is a threshold at which an action is considered taken (progress to decision arb. u.)

In our analysis, we run simulations using a constant time step discretization of eq. 12.

For code, see https://github.com/dirkbeck/DM_space_model/blob/main/weiner_process_model.m.

**Modeled circuit manipulation using Instance 1**

To get a sense of the functional role of the circuit elements, we conducted sensitivity analyses by changing parameters in the model individually and determining their effect on the activities of other circuit elements, the decision-space and/or choice.

**Common parameters.** The values specified here, arbitrarily chosen, are used in the analyses in **Instance 1** unless otherwise indicated:

- throughout, $k = 4$
- in eq. 1: $b_{\text{sSPN}} = 0$
- in eq. 2: $w_{\text{sSPN} \rightarrow \text{daSNC}, i, P} = 1$ for all $i$ and $P$
- in eq. 2: $z_{\text{daSNC}, i, P} = 1$ for all $i$ and $P$
- in eq. 4:, $\boldsymbol{\beta}_{\text{direct}} = \begin{pmatrix} 1 & -1 & 0 & 0 \\ -1 & 1 & 0 & 0 \\ 0 & 0 & 0 & 0 \\ 0 & 0 & 0 & 0 \end{pmatrix}$
- in eq. 4: $\alpha_{j,P} = -3$ for all $j$ and $P$
- in eq. 5: $a_{\text{FSI}} = 1$
- in eq. 5: $b_{\text{FSI}} = 0.5$
- in eq. 7: $z_{\text{LHb}} = 0.5$
- in eq. 7: $z_{\text{RMTg}} = 0.5$
- in eq. 7: $z_{\text{GPi}} = 1$
- in the inputs to eq. 10, $q = 4$
- in eq. 12: $\sigma = 1$
- in eq. 13, 14: $h = 2$

whose rows correspond to, for example: turning left, turning right, turning around, wandering; and whose columns correspond to, for example: a reward-predominant decision-dimension 1, a cost-predominant decision-dimension 2, a novelty-predominant decision-dimension 3, and a location-predominant decision-dimension 4. The coefficients model a T-maze where a choice is made to turn right or left based on relative values of cost and reward.

**Rationale behind parameter choices.** The parameters chosen above are set to these default values in order to probe the selective effects of different circuit elements on the decision-space, action values, and choice, whose units are often arbitrary. Thus, we make our parameter choices arbitrarily, but with an emphasis on computational simplicity.

We can think of the free parameters as falling into three groups. This is because the circuit operates in a three-step process: first the decision-space is formed, then action values are defined, then choices are made by downstream neural circuits. So, parameter choices in a Group 1 affect the decision-space, parameter choices in a Group 2 affect action values, and parameter choices in a Group 3 affect choice. More precisely,

1) Group 1 affects the decision-space, and action values and choice only via the decision-space. The parameters in eqs. 1, 2, 5, 7, fall in Group 1.
2) Group 2 affects action values and choice via action values but not the decision-space. The parameters in eq. 4 fall in Group 2.
3) Group 3 affects choice, but not action values nor the decision-space. The parameters in eqs. 12, 13, and 14 fall in Group 3.

We chose the common free parameters in Group 1 with the goal in mind of probing the relative effects of circuit elements on the decision-space. That is, we use the model to gain intuition about how each circuit element will affect daSNC activity, given their physiological relationships. For instance, $z_{LHb}$ and $z_{RMTg}$ in eq. 7 are mathematically redundant, but are included because the effect of the brain regions individually is important to our analysis (see Supplementary Table 5). In all cases in our fitting to data involving Group 1 free parameters, we analyze relative rather than absolute effects of changes to neural activity. For instance, we examine differences between high versus medium versus low striosome activity in Figs. 3 and 4 and Supplementary Table 3. Note that the activities are given in arbitrary units, so other combinations of parameter values with the same net effect would reproduce our results exactly (for instance, $z_{LHb} = 1$, $z_{RMTg} = 0$). The current choices of common parameters were arbitrary, but selected so that the eq. 2 simplifies to a logistic function with a bias term $f(x) = \frac{1}{1+e^{x+b}}$ when the activities of all circuit elements are set to their defaults. So, absent any change to the defaults among the circuit elements, and absent any inputs from cortex above or below baseline, each decision-dimension has 50% probability of being used to form the decision-space.

In Group 2, $\beta_{direct}$ and $\alpha_{j,P}$ are constructed as an example of a scenario often used during our analysis, where there are two decision-dimensions that convey opposite information (e.g., the first reward-predominant, the second cost-predominant) that affect whether to take two actions (e.g. turning right or left). Thus, the signs of the numbers in $\beta_{direct}$ are modeled from the task, but it could be scaled by any coefficient greater than 0 and achieve similar results to our analysis. We chose a negative value for $\alpha_{j,P}$ to reflect its functional role as a prior. So, a given action is assigned low value unless there is significant evidence that it is valuable. $\alpha_{j,P}$ is assigned arbitrarily, although very low $\alpha_{j,P}$ would lead to action values near 0 and very high $\alpha_{j,P}$ would lead to action values near 1.

In Group 3, we chose parameters that roughly reproduced experimental deliberation time distributions (see Supplementary Fig. 3b, c).

**Effect of reward/costs on LHb/RMTg/daSNC activity.** In Supplementary Fig. 2e, f, we modeled the effect of incrementing reward or cost on the activities of LHb, RMTg, and daSNC.

The inputs enter the model circuit in two ways: 1) reward and cost are mapped to decision-dimensions; and 2) cost level leads to changes in LHb and RMTg activities, similar to what has been demonstrated in the experimental literature (described in Supplementary Note 12). The modeled LHb and RMTg responses to cost are proportional to cost level with an arbitrary coefficient (set to 1 for LHb and 0.9 for RMTg for the purposes of plotting).

The modeled results show that the mean activity of a daSNC sub-population encoding reward-predominant data responds positively to increases in reward and negatively to decreases in reward, similar to what has been demonstrated in the experimental literature (described in Supplementary Note 12). LHb and RMTg respond negatively linearly to reward level and positive linearly to cost level, similar to trends in the experimental evidence (see Supplementary Information). Sudden changes in reward or cost level, therefore, lead to shifts in activities that track changes to expectations of future reward or cost value, including reward or cost currently received, i.e. reward or cost prediction error.

For code, see https://github.com/dirkbeck/DM_space_model/blob/main/model_overview/GPi_LHb_RMTg_DA_model.m.

**Effect of LHb/RMTg/daSNC activity on the decision-space.** In Supplementary Fig. 2g-i, we modeled the effect of incrementing GPi, LHb, RMTg, or daSNC activity on the type of decision-space formed during a decision.

In the plotted analysis, we altered $z_{GPi}$ in eq. 7, $z_{LHb}$ in eq. 7, $z_{RMTg}$ in eq. 7, and $z_{daSNC, i, P}$ (uniform change for all $i$, a single pathway is considered) in eq. 7 such that they took 10 values incremented from 0 to 1. Parameters not altered took default values (see **Common parameters**).

We also examined the role of each component in decision-space formation through the perspective of a series of steps, each carried out by a different circuit element. For this analysis, we substituted eq. 7 into eq. 2 and altered each parameter in turn. The plots illustrate the value of daSNC$_{i,P}$ if the other parameters were set to 1 ($z_{GPi}$) or 0 ($z_{RMTg}$, $z_{daSNC, i, P}$). $b_{LHb}$ is set to 0.5 (control), −5 (lesioned LHb), or 5 (stimulated LHb).

See Supplementary Tables 5,6 for alignment to the experimental literature.

For code, see https://github.com/dirkbeck/DM_space_model/blob/main/model_overview/GPi_LHb_RMTg_DA_model.m.

**Effect of sSPN and mSPN activity on the decision-space and action values.** In the analysis plotted in Fig. 2c, we incremented $b_{sSPN}$ in eq. 1 from 0 to 1 arbitrary units and recorded the value of daSNC$_i$ in eq. 2. Similarly, we incremented an addition to mSPN activity in eq. 3 from 0 to 1 arbitrary units, and note that this does not affect daSNC$_i$.

In the analysis plotted in Fig. 2d, we again incremented $b_{sSPN}$ and a similar constant for an mSPN activity addition from 0 to 1 arbitrary units. Here, we computed action values as a multiple of mSPN activity, passed through an activation function, per eqs. 3, 4. The unspecified inputs to the equation ($\beta_{direct}$, cortical input, $\alpha_{j,P}$) are randomly assigned.

For code, see https://github.com/dirkbeck/DM_space_model/blob/main/model_overview/sSPN_versus_mSPN_effect_on_decision_space_and_action_values.m

**Effect of GPi, LHb, RMTg, and GPe activity on the decision-space.** In the analyses plotted in Fig. 2e, f, the activities of the circuit elements are incremented from 0 to 1 arbitrary units in eqs. 2 and 7 and daSNC$_i$ in eq. 2 is recorded.

For code, see https://github.com/dirkbeck/DM_space_model/blob/main/model_overview/GPi_LHb_RMTg_DA_model.m.

**Effect of sSPN activity on the decision-space.** In the analysis plotted in Fig. 3a, we incremented $b_{sSPN}$ in eq. 1 and, for each increment, recorded daSNC$_i$ in eq. 2. Then, using the approach in eq. 10, we converted the probability that one decision-dimension is used in the formation of decision-space to the probability that a decision-spaces of a certain dimensionality is formed.

For code, see https://github.com/dirkbeck/DM_space_model/blob/main/model_tests/friedman2015optogeneticmanipulation.m.

**Effect of the decision-space on choice.** In the analysis plotted in Fig. 3b, we changed which decision-space was formed by mSPNs and measured choice.

The excitation group was modeled using a zero-dimensional decision-space (dopamine→mSPN weights of 0 reward-predominant decision-dimension, 0 cost-predominant dimension). The control group was modeled using a 1D direct pathway decision-space (dopamine→mSPN weights of 0.5 reward-predominant dimension, 0 cost-predominant dimension). The inhibition group was modeled using a 2D direct pathway decision-space (dopamine→mSPN weights of 1 reward-predominant dimension, 1 cost-predominant dimension). The

modeled T-maze task was a choice between reward=2, cost=1 (high reward, high cost) and reward=1, cost=0.5 (low reward, low cost). 20 simulations were run per modeled subject for 100 subjects. Other parameters for forming the decision-space and calculating action value are set to their defaults (see **Common parameters**). For simplicity, the indirect pathway is not modeled in this analysis.

For code, see https://github.com/dirkbeck/DM_space_model/blob/main/model_tests/friedman2015optogeneticmanipulation.m.

In the analysis in Fig. 4b,c, we modeled changes to decision-space and choice after stress.

Here, modeled control rodents made decisions using a 2D direct pathway decision-space formed from reward-predominant and cost-predominant decision-dimensions. This corresponds mathematically to a truncation of $\boldsymbol{\beta}_{\text{direct}}$ (see eq. 4 and **Common parameters**) to two columns. The first subset of modeled stress-group rodents made decisions without forming direct pathway decision-space. This corresponds to an elimination of $\boldsymbol{\beta}_{\text{direct}}$ such that action value is defined purely based on priors ($\alpha_{j,\text{direct}}$ in eq. 4). The second subset made decisions without forming direct pathway decision-space until they reached a critical threshold, beyond which they formed a 1D direct pathway decision-space with a reward-predominant dimension. Action values are derived for the three groups across multiple reward and cost combinations (Fig. 4c) via eqs. 3 and 4. Then choices are modeled using eqs. 12, 13, and 15 across 2000 simulations per group for each reward concentration (each incremented from 0 to 1 arbitrary units, 7 increments). Cost concentration is set to 0.5 arbitrary units (set at this level to resemble the steepness of increase in the experimental psychometric function). Default parameters are used for action value formation and the Merton process model. For simplicity, the indirect pathway is not modeled in this analysis. Figure 4b plots the averages of the simulations.

For code, see https://github.com/dirkbeck/DM_space_model/blob/main/disorder_hypotheses/Friedman2017_lowD_space.m.

In the analysis plotted in Fig. 4d,e, we modeled the effect on choice of shifts in decision-space after a small cost is added to a reward (experimental data is plotted in Supplementary Fig. 4j).

Rodents in the only-reward task were modeled as forming a lower-dimensional direct pathway decision-space (decision-dimension 1 weight = 0.5, decision-dimension 2 weight = 0.2) while animals in the reward-and-cost task formed a higher-dimensional direct pathway decision-space (decision-dimension 1 weight = 1, decision-dimension 2 weight = 0.5).

To do this, we truncated $\boldsymbol{\beta}_{\text{direct}}$ (see eq. 4 and **Common parameters**) to two columns or derived action value purely based on priors ($\alpha_{j,\text{direct}}$ in eq. 4). A cortical input of reward = 0.7, cost = 0.3 is shown in the plots. For simplicity, the indirect pathway is not modeled in this analysis.

For code, see https://github.com/dirkbeck/DM_space_model/blob/main/disorder_hypotheses/altered_choice_after_space_transition.m.

In the analysis plotted in Fig. 4h, we modeled changes to choice after aging in young and old groups.

Here, we truncated $\boldsymbol{\beta}_{\text{direct}}$ (see eq. 4 and **Common parameters**) to two decision-dimensions, the first corresponding to a reward-predominant decision-dimension and the second corresponding to a cost-predominant decision-dimension. In the current analysis, the first row of $\boldsymbol{\beta}_{\text{direct}}$ corresponded to licking, while the second row corresponds to performing a different action, e.g. movement. The licking action was assigned a larger prior, $\alpha_{1,\text{direct}} = 0$, $\alpha_{2,\text{direct}} = -3$, due to the strong association developed in the rodents between the experimental apparatus and licking. For the modeled "learned, young" group, no decision-space is formed during the reward-cue task and a decision-space using only a cost-predominant decision-dimension is formed during the cost-cue task (i.e.

$\mathbf{s}_{\text{mSPN, reward, direct}} = \begin{bmatrix} 0 \\ 0 \end{bmatrix}$, $\mathbf{s}_{\text{mSPN, cost, direct}} = \begin{bmatrix} 0 \\ 1 \end{bmatrix}$ in eq. 3). For the

modeled "learned, old" group, no decision-space is formed during the reward-cue task and a decision-space involving a cost-predominant decision-dimension is partially formed during the cost task $\left( \mathbf{s}_{\text{mSPN, reward, direct}} = \begin{bmatrix} 0 \\ 0 \end{bmatrix}, \mathbf{s}_{\text{mSPN, cost, direct}} = \begin{bmatrix} 0 \\ 0.5 \end{bmatrix} \right)$. For the "not learned" group, a decision-space involving a cost-predominant decision-dimension is partially formed during both tasks $\left( \mathbf{s}_{\text{mSPN, reward, direct}} = \begin{bmatrix} 0 \\ 0.5 \end{bmatrix}, \mathbf{s}_{\text{mSPN, cost, direct}} = \begin{bmatrix} 0 \\ 0.5 \end{bmatrix} \right)$. For simplicity, the indirect pathway is not modeled in this analysis.

For code, see https://github.com/dirkbeck/DM_space_model/blob/main/disorder_hypotheses/Friedman2020_lowD_space.m.

**Effect of the decision-space on sSPN-mSPN correlation.** In the analysis plotted in Fig. 3d, sSPN-mSPN correlation is compared across decision-spaces with different dimensionality.

It is assumed in the plotted examples that a 1D decision-space is only formed from the first decision-dimension, a 2D decision-space is only formed from the first and the second, and a 3D decision-space is only formed from the first, second, and third. The analysis assumes a comparison of SPNs of the same pathway (that is, either dsSPN-dmSPN or isSPN-imSPN). For this analysis, eigenvalues of cortical activity are set to 2, 1, 0.5, 0.2, and 0.1, respectively. Weighted averages of example signals (left panel) and correlation for different decision-spaces (right panel) are formed using the identity that eigenvalues of principal components are equivalent to their variances.

For code, see https://github.com/dirkbeck/DM_space_model/blob/main/model_tests/ctx_sSPN_mSPN_coordinated_activity.m.

**Effect of FSI activity on decision-space.** In the analysis plotted in Fig. 4f, we incremented FSI activity $a_{\text{FSI}}$ in eq. 5 and determined the response of daSNC$_i$ in eq. 2 (a single pathway is considered). The activity parameters related to other circuit elements were held constant (see **Common parameters**).

For code, see https://github.com/dirkbeck/DM_space_model/blob/main/disorder_hypotheses/space_dimensionality_vs_FSI.m.

**Effect of cortical SNR on choice.** In the analysis plotted in Supplementary Fig. 6a-e, cortical signal to noise ratio (SNR) is altered and the effect on choice is simulated.

Merton process simulations (see **Defining choice**) are run across ten increments of reward and cost from −1 to 1 arbitrary units for a modeled cost-benefit conflict task. Parameters related to action value are set to their defaults and the T-maze task is used (see **Common parameters**). Here, "turn right" corresponds to receiving the reward and cost combination, while other actions correspond to receiving no reward and no cost. For simplicity, only the direct pathway is used to influence choice. 100 simulations are run for each of 100 reward and cost combinations, and for each combination, choice is averaged.

The above process is replicated with changes to two sets of parameters. First, the effect of changes to decision-space were considered. A different $\mathbf{S}$ in eq. 3 was used depending on specified deci-

sion-space: $\mathbf{S} = \begin{bmatrix} 1 & 0 & 0 \\ 0 & 0 & \vdots \\ 0 & \cdots & 0 \end{bmatrix}$ for 1D decision-spaces, and

$\mathbf{S} = \begin{bmatrix} & & 0 \\ \mathbf{I_2} & & \vdots \\ 0 & \cdots & 0 \end{bmatrix}$ for 2D decision-spaces. Second, changes to cortical

noise were considered by adding i.i.d. Gaussian noise to $\mathbf{x}_{\text{direct}}$ (with mean 0 and standard deviation $\sigma$) at every time step, then recalculating action values in eq. 4 based on the mSPN activities calculated at that time step. Simulations were run for $\sigma = 1,2,\ldots,10$. Default parameters were used for calculating action value (see **Common parameters**).

Examples of single simulations at each level of reward and cost are shown for the $\sigma = 1$ (high cortical SNR) and $\sigma = 5$ (low cortical SNR) cases in Supplementary Fig. 6a–d. In Supplementary Fig. 6e, expected value is averaged across the 100 simulations for each noise level. Expected value here is defined as reward minus $0.75 \cdot$ cost (to add preference for reward compared to cost, coefficient is arbitrary) achieved across reward and cost levels. In the plot, SNR is set to the inverse of $\sigma$.

For code, see https://github.com/dirkbeck/DM_space_model/blob/main/dynamic_model_and_neural_net/cortical_snr.m.

**Effect of dopamine on action/inaction values.** In the analyses plotted in Supplementary Fig. 7c–f, we measured the effect of high versus low dopamine on action and inaction values across a range of cortical inputs to the system.

We modeled a cost-benefit conflict task with increasing reward (scale of 0 to 1 arbitrary units, 100 increments) and constant cost (set to 0.25 arbitrary units). Experimental work has shown that dopamine increases direct pathway activity while decreasing indirect pathway activity and vice versa[53]. Therefore, we set coefficients relating to overall activity of the pathways oppositely: in the low dopamine case, the direct pathway coefficient was 0.1 arbitrary unit and the indirect pathway coefficient 5 arbitrary units; and in the high dopamine case, the indirect pathway coefficient was 5 arbitrary units and the direct pathway coefficient 0.1 arbitrary unit. These coefficients were multiplied by $\boldsymbol{\beta}_{\text{direct}}$ or $\boldsymbol{\beta}_{\text{indirect}}$ in eq. 4, increasing or decreasing the overall sensitivity of action value on data along cortical principal components. In the model, changes to dopamine also involved a change in decision-space: due to their opposite effects on mSPN activity, dopamine biases the direct pathway towards forming higher-dimensional decision-spaces and the indirect pathway towards forming lower-dimensional decision-spaces. For the purpose of this analysis, eq. 4 is reframed to incorporate the effects of dopamine in scaling action value score ($A$, set to either 5 or 0.1 arbitrary units in our analysis) and changing decision-space ($B$, set to 1 arbitrary unit when dopamine is high and 0 when dopamine is low). Here, individual elements are referenced through subscripts based on their $j$th row and column corresponding to the reward or cost dimension.

$$v_{j,\,\text{direct}} = \frac{1}{1 + \exp\left(-A \cdot (\beta_{j,\,\text{reward, direct}} + B \cdot \beta_{j,\,\text{cost, direct}}) - \alpha_{j,\,\text{direct}}\right)} \quad (16)$$

$$v_{j,\,\text{indirect}} = \frac{1}{1 + \exp\left(-A \cdot ((1-B) \cdot \beta_{j,\,\text{reward, indirect}} + \beta_{j,\,\text{cost, indirect}}) - \alpha_{j,\,\text{indirect}}\right)} \quad (17)$$

where:
- $A$ is the multiplicative effect of dopamine released to mSPNs (dimensionless)
- $B$ is the effect of dopamine on decision-space (dimensionless)
- $\beta_j$ is the connection weight from mSPN to an action value neuron. Each $\beta_{j,\text{reward}}$ or $\beta_{j,\text{cost}}$ corresponds to an element of the connection weight matrix $\boldsymbol{\beta}_P$.
- $\text{prior}_j$ is an additive shift corresponding to the neuron encoding action $j$ (activity arb. u.)

The plot in Supplementary Fig. 7c compares the high-dopamine and low-dopamine cases. The plot in Supplementary Fig. 7d shows a similar analysis but for changes in parameters: cost is fixed at 0.5 arbitrary units, and $A$ is set to either 2 arbitrary units (corresponding to the pathway not disconnected) or 0 (corresponding to the pathway disconnected). The plots in Supplementary Fig. 7e, f show progress to action in the case where reward = 1 arbitrary unit and cost = 0.25 arbitrary unit for low versus high dopamine. Deliberation time distributions are formed by aggregating the deliberation times across the 100 simulations. For parameters used for subjective valuation and deliberation time simulation, see **Common parameters**.

For code, see https://github.com/dirkbeck/DM_space_model/blob/main/dynamic_model_and_neural_net/direct_vs_indirect_pathway_SV.m.

**Effect of decision-dimensions on choice.** In the analysis plotted in Supplementary Fig. 7i, we altered the connections between mSPN and action/inaction encoding neurons on choice.

A modeled approach/avoid experiment is conducted by offering an option with reward = 1 arbitrary unit, cost = 1 arbitrary unit, and varying (10 values incremented from [0, 2] arbitrary units) physical proximity to another reward. An additional column is added to $\boldsymbol{\beta}_{\text{direct}}$ and $\boldsymbol{\beta}_{\text{indirect}}$ in eq. 4 to reflect the fact that additional proximity to the other reward increases approach rate: $\boldsymbol{\beta}_{\text{direct}} = \boldsymbol{\beta}_{\text{indirect}} = \begin{bmatrix} 1 & -1 & 1 \\ -1 & 1 & -1 \end{bmatrix}$, where the upper row corresponds to approaching, the bottom row corresponds to not approaching, and the columns correspond to, from left to right, a reward-predominant decision-dimension, a cost-predominant decision-dimension, and a location-predominant decision-dimension. Reward is assigned a greater relative importance than cost or location (score of 3 arbitrary units versus 1 versus 1) in sSPNs, while cost is assigned a greater relative importance than reward or location (score of 3 arbitrary units versus 1 versus 1). daSNC activity is incremented by changing $z_{\text{daSNC},\,i,\,P}$ in eq. 2 for all $i$ and both pathways. Action and inaction values are calculated, and then choice is formed by averaging the results of 1000 Merton process simulations.

For code, see https://github.com/dirkbeck/DM_space_model/blob/main/dynamic_model_and_neural_net/direct_vs_indirect_pathway_proximity_theory.m.

**Effects of sSPN, LHb, and daSNC activity on decision-space.** In Fig. 6a, $b_{\text{sSPN}}$ (eq. 1), $z_{\text{LHb}}$ (eq. 7), and $z_{\text{daSNC},\,i,\,P}$ (for all $i$ and a single pathway, eq. 2) are incremented from 0 to 5 arbitrary units with 20 evenly spaced increments along each axis. Each of the $20 \times 20 \times 20$ points are used to derive $\text{daSNC}_i$ in eq. 2 and converted to decision-spaces via eq. 10.

For code, see https://github.com/dirkbeck/DM_space_model/blob/main/day_to_day_space_sampling/decision_space_sampling.py.

**Effect of decision-space on choice profiles.** In Fig. 6b,e, Supplementary Fig. 8d,f, we form decision-spaces using various decision-dimensions across incremented cortical reward and cost inputs, then classified the action values formed across those reward/cost inputs using a scoring system.

The scoring system, visualized in Supplementary Fig. 8c, is as follows:

Scores for "explore", "riskiness", "high action", "exploit", "safety", "low action" are calculated by incrementing reward and cost on $[-1\ 1]$ (arbitrary units) scales (9 increments are used for each of reward and cost in Fig. 6b, Supplementary Fig. 8d, 6 increments in Fig. 5e, Supplementary Fig. 8f). The notation used here treats $v_{r,c,j}$ as the action value of the $j$th action at a certain reward and cost increment and $v_{r,c}$ as the set of those action values. In the plotted analysis, $k = 4$ actions are assigned action values.

To keep with the indexing conventions of the other equations (for instance, eq. 4), we use the index $j$ here to refer to one of the $k$ actions. In the case of eq. 19, where the actions are iterated over twice, we use $j_1$ and $j_2$.

- Explore. The tendency to pursue multiple actions simultaneously. Scored as the area of the region of reward and cost combinations with a Gini coefficient less than 0.25.

$$\text{explore} = \sum_{r=-1}^{1} \sum_{c=-1}^{1} [\text{gini}(v_{r,c}) < 0.25] \quad (18)$$

$$\text{gini}(v_{r,c}) = \frac{\sum_{j_1=1}^{k} \sum_{j_2=1}^{k} |v_{r,c,j_1} - v_{r,c,j_2}|}{2k \sum_{j=1}^{k} v_{r,c,j}} \quad (19)$$

- Exploit. The tendency to pursue only one action. Scored at the area of the region of reward and cost combinations with a Gini coefficient greater than 0.5.

$$\text{exploit} = \sum_{r=-1}^{1} \sum_{c=-1}^{1} [\text{gini}(v_{r,c}) > 0.5] \qquad (20)$$

- Riskiness. The combined value of actions when reward and cost are high. Scored by examining the combinations where both reward and cost are greater than 0.

$$\text{riskiness} = \sum_{r=0}^{1} \sum_{c=0}^{1} \sum_{j=1}^{k} v_{r,c,j} \qquad (21)$$

- Safety. The combined value of actions when reward and cost are low. Scored by examining the combinations where both reward and cost are less than 0.

$$\text{safety} = \sum_{r=-1}^{0} \sum_{c=-1}^{0} \sum_{j=1}^{k} v_{r,c,j} \qquad (22)$$

- High action. How often actions will have high action values. Scored as the area of the region of reward and cost combinations that have combined action value greater than 0.5.

$$\text{high action} = \sum_{r=-1}^{1} \sum_{c=-1}^{1} \left[ \sum_{j=1}^{k} v_{r,c,j} > 0.5 \right] \qquad (23)$$

- Low action. How often actions will have low action values. Scored as the area of the region of reward and cost combinations that have combined action value less than 0.2.

$$\text{high action} = \sum_{r=-1}^{1} \sum_{c=-1}^{1} \left[ \sum_{j=1}^{k} v_{r,c,j} < 0.2 \right] \qquad (24)$$

In the analyses plotted in Fig. 6b and Supplementary Fig. 8d and the examples in Fig. 6c, action value scores, as measured by the scoring definitions above, are compared when different decision-spaces are constructed but cortical input and system parameters are unchanged. The underlying action values across reward and cost levels resembles those from other analyses (see **Common parameters**) except for an addition of normal random noise (mean = 0, standard deviation = 1) to every element of $\boldsymbol{\beta}_{\text{direct}}$ (see eq. 4).

The analysis in Supplementary Fig. 8d is similar, except for here, a weighted average is taken of action value scores, as measured by the scoring definitions above, between scenarios where different decision-spaces are constructed. A different **S** in eq. 3 is used depending on the specified dimensionality of direct pathway decision-space:

$$\mathbf{S} = \begin{bmatrix} 1 & 0 & 0 \\ 0 & 0 & \vdots \\ 0 & \cdots & 0 \end{bmatrix} \text{ for 1D, } \mathbf{S} = \begin{bmatrix} & & 0 \\ & \mathbf{I_2} & \vdots \\ 0 & \cdots & 0 \end{bmatrix} \text{ for 2D, } \mathbf{S} = \begin{bmatrix} & & 0 \\ & \mathbf{I_3} & \vdots \\ 0 & \cdots & 0 \end{bmatrix}$$

for 3D, and $\mathbf{S} = \mathbf{I_4}$ for 4D. A weighted average of the five decision-spaces is calculated for three levels of sSPN activity (−1, 0, and 1).

For code, see https://github.com/dirkbeck/DM_space_model/blob/main/day_to_day_space_sampling/subjective_value_scores_by_space.m.

In the analyses plotted in Fig. 6e, Supplementary Fig. 8f, for each of 1000 simulations, uncorrelated Gaussian white noise (mean = 0,

standard deviation = 1) is added to every element of $\boldsymbol{\beta}_{\text{direct}}$ (see eq. 4) and 6 by 6 grids of action values across reward and cost combinations are scored by the "explore", "riskiness", "high action", "exploit", "safety", and "low action" metrics. Scores for each metric are compared across simulations and between decision-space groups. Observations that score in the top 10% by a metric are considered outliers. Outlier proportion is plotted in Fig. 6e. The means across the simulations of each score are plotted in Supplementary Fig. 8f.

For code, see https://github.com/dirkbeck/DM_space_model/blob/main/day_to_day_space_sampling/subjective_value_score_extremes.m.

**Instance 2: sparse connectivity and feedforward**
In this section, we describe the instance of the model where cortex input to the system does not change over time and the activities of other circuit elements do not decay over time.

This instance of the model leads to a convenient formation of the model as a circuit with no time component. In this section, we frame this instance mathematically and then describe our related analysis. To focus on the portions of the circuit we analyze using this instance, we define here the subset of the circuit involving cortex, FSI, sSPN, daSNC, and mSPN.

For code, see https://github.com/dirkbeck/DM_space_model/blob/main/dynamic_model_and_neural_net/neural_network_model.m.

**Cortical input**. A set of 4 cortical neurons, notated as $C$, is sampled at random from a population of 50 cortical neurons. Each neuron in $C$ projects to one FSI and each of $q$ SPNs, which each correspond to a decision-dimension. In our analysis, $q$ is set to 4. This process is repeated 10,000 times per each pathway, forming 10,000 groups of 4 cortical neurons, 1 FSI, 4 dsSPNs (or isSPNs), and 4 dmSPNs (or imSPNs) for each pathway.

**Defining FSI activity**. FSI activity is defined as a weighted sum of the activities of connected cortical neurons:

$$\text{FSI}_C = \sum_{q \in C} w_{\text{cortex} \to \text{FSI}} \, \text{cortex}_q + b_{\text{FSI}} \qquad (25)$$

where:
- $C$ is a randomly sampled subset of cortical neurons.
- $\text{FSI}_C$ is the activity of the FSI which receives projection from the cortical neurons in $C$ (activity arb. u.)
- $w_{\text{cortex} \to \text{FSI}}$ is the connection weight between cortical neurons and FSIs (dimensionless)
- $\text{cortex}_q$ is the activity of cortical neuron $q$ (activity arb. u.)
- $b_{\text{FSI}}$ affects the relative activity of all sSPN neurons (activity arb. u.)

In the current instance of the model, there are 10,000 FSIs for the direct pathway (projecting to dSPNs) and 10,000 FSIs for the indirect pathway (projecting to iSPNs). Each cortical neuron in $C$ projects to $\text{FSI}_C$.

**Defining sSPN activity**. sSPN activity is defined as a weighted sum of the activities of connected cortical neurons, divided by a weighted sum of the activities of connected FSIs, plus an additive shift $b_{\text{sSPN}}$ applied to all sSPNs:

$$\text{sSPN}_{s, C} = \frac{1}{|C|} \sum_{q \in C} \frac{w_{q \to s} \, \text{cortex}_q}{\text{FSI}_C} + b_{\text{sSPN}} \qquad (26)$$

where:
- $C$ is a randomly sampled subset of cortical neurons.
- $\text{sSPN}_{s,C}$ is the activity of an sSPN $s$ that receives projection from cortical neurons in $C$. (activity arb. u.)

- $w_{q \rightarrow s}$ is the connection weight between cortical neuron $q$ and sSPN $s$. The weight is equivalent to one of the first four principal components of the cortical activity of the four connected cortical neurons. sSPNs are separated into equal populations that correspond to the first, second, third, or fourth principal component. (dimensionless)
- $\mathrm{FSI}_C$ is the activity of the FSI which receives projection from the cortical neurons in $C$ (activity arb. u.)
- $\mathrm{cortex}_q$ is the activity of cortical neuron $q$ (activity arb. u.)
- $b_{\mathrm{sSPN}}$ represents the relative activity of all sSPN neurons (activity arb. u.)

In the current instance of the model, there are 40,000 neurons for each of dsSPNs and isSPNs. Activities are defined based on a feedforward network, so the simplification is made that sSPN activities are not affected by daSNC activities. All cortical neurons in $C$ project to sSPN$_{s,C}$ for all $s$, and similarly, FSI$_C$ projects to sSPN$_{s,C}$ for all $s$.

**Defining daSNC activity.** The activity of the daSNC element corresponding to decision-dimension $i$ and pathway $P$ is defined by weighted inputs from sSPNs corresponding to decision-dimension $i$ and pathway $P$:

$$\mathrm{daSNC}_{i,P} = \frac{1}{1 + \exp\left(\frac{1}{n_{\mathrm{sSPN},i,P}} \sum_{s \in i, P} w_{s \rightarrow \mathrm{daSNC},i,P} \mathrm{sSPN}_s + \mathrm{RMTg} - z_{\mathrm{daSNC},i,P}\right)} \quad (27)$$

where:

- $\mathrm{daSNC}_{i,P}$ is the activity the daSNC neuron corresponding to decision-dimension $i$ and pathway $P$. (activity arb. u.)
- $n_{\mathrm{sSPN},i,P}$ is the count of sSPNs in each of the direct/indirect pathways
- $w_{s \rightarrow \mathrm{daSNC},i,P}$ is the connection weight from sSPN $s$ to the daSNC neuron corresponding to decision-dimension $i$ and pathway $P$. The weight is fixed in this instance of the model. (dimensionless)
- $\mathrm{sSPN}_s$ is the activity of SPN $s$ (activity arb. u.)
- $\mathrm{RMTg}$ is RMTg activity (activity arb. u.)
- $z_{\mathrm{daSNC},i,P}$ is the bias in the activity of a daSNC neuron corresponding to decision-dimension $i$ and pathway $P$. (activity arb. u.)

In the current instance of the model, there are $q$ daSNC neurons that receive projection from the 10,000 dsSPNs corresponding to each decision-dimension, and likewise $q$ daSNC neurons that receive projection from the 10,000 isSPNs corresponding to each decision-dimension. RMTg also projects to all daSNC neurons.

In our analysis using this instance of the model, we arbitrarily set $w_{\mathrm{sSPN} \rightarrow \mathrm{daSNC},i} = 1$ arbitrary unit for both pathways and $z_{\mathrm{daSNC},i} = -5$ arbitrary units for all $i$ for the direct pathway, and $z_{\mathrm{daSNC},i} = 5$ arbitrary units for all $i$ for the indirect pathway. For simplicity, RMTg is set to 0.

**Defining mSPN activity and the decision-space.** mSPN activity is defined as a weighted sum of the activities of connected cortical neurons, divided by a weighted sum of the activities of connected FSIs. Here, unlike in the definition of sSPN activity in eq. 26, a $d_{i,P}$ term is multiplied to incorporate the weighting of mSPNs by dopamine:

$$\mathrm{mSPN}_{m,C} = \frac{d_{i,P}}{|C|} \sum_{q \in C} \frac{w_{q \rightarrow m} \mathrm{cortex}_q}{\mathrm{FSI}_C} \quad (28)$$

where:

- $C$ is a randomly sampled subset of cortical neurons
- $\mathrm{mSPN}_{m,C}$ is the activity of mSPN $m$ that receives projection from cortical neurons in $C$ (activity arb. u.)
- $d_{i,P}$, which takes the value 0 or 1, is dopamine signaling to mSPNs corresponding to decision-dimension $i$ and pathway $P$. $d_{i,P}$ is the

realization of probabilistic weighting of decision-dimensions based on daSNC activity (see **Instance 1**). (dimensionless)
- $w_{q \rightarrow m}$ is the connection weight between cortical neuron $q$ and mSPN $m$. The weight is equivalent to one of the first four principal components of the cortical activity of the four connected cortical neurons. As described in **Instance 1**, sSPNs are separated into equal populations that correspond to the first, second, third, or fourth principal component. (dimensionless)
- $\mathrm{FSI}_C$ is the activity of the FSI which receives projection from the cortical neurons in $C$ (activity arb. u.)
- $\mathrm{cortex}_q$ is the activity of cortical neuron $q$ (activity arb. u.)

In the current instance of the model, there are 40,000 neurons for each of dmSPNs and imSPNs All cortical neurons in $C$ project to mSPN$_{m,C}$ for all $m$, and similarly, FSI$_C$ projects to mSPN$_{m,C}$ for all $m$.

**Modeling SPN encoding of data, using Instance 2.** To explore the ability of SPNs to successfully encode data along decision-dimensions, even when cortex and SPNs are sparsely connected, we constructed networks with different degrees of dimensionality reduction (Supplementary Fig. 5g). A single pathway is considered. One type of network had 2 times dimensionality reduction (20 cortical neurons, 10 SPNs), another had 10 times dimensionality reduction (100 cortical neurons, 10 SPNs), and another had 100 times dimensionality reduction (1000 cortical neurons, 10 SPNs), similar to what is found in the human brain[67,68].

For each type, we constructed modeled networks with cortex→SPN connections equal to principal components of cortical activity by simulating, for each analyzed pathway, a random symmetric positive definite matrix that is used as the cortical covariance matrix $\Sigma_P$ (see **Defining sSPN activity**) via MATLAB's sprandsym() with density=1. Eigenvalues are arbitrarily specified as $\lambda_1 = 2$, $\lambda_2 = 2$, $\lambda_3 = 0.5$, $\lambda_4 = 0.2$, and $\lambda_5 = \lambda_6 = \ldots = \lambda_P = 0$, i.e. the first and second principal components are very important, the third somewhat important, the fourth slightly important, and the others unimportant. The weights $\mathbf{W}_P$ from cortical neurons to the $q$ SPN circuit elements per pathway are then derived as the first $q$ eigenvectors of $\Sigma_P$.

We incremented the number of cortical neurons that connected to each SPN from 2 to 10. The network was connected sparsely based on the specified number of connections from randomly selected cortical neurons to each SPN. For simplicity in this analysis, we created a cortical signal that resembled the first principal component of cortical activity as a whole (regardless of connectivity to SPNs) and let the first cortical principal component to have large eigenvalue compared to the others (i.e. $\lambda_1 = 1$, $\lambda_2 = 0.1$, $\lambda_3 = \lambda_4 = \ldots = \lambda_P = 0$).

For each of the modeled networks (3 network types by 9 increments from 2 to 10), we simulated the process 1000 times. During each simulation, we calculated the ability of the network to discriminate between the large signal along the first cortical principal component and the absence of signal along the second cortical principal component, given its access to only a subset (2 to 10 cortical neurons) of the complete signal.

Then the SPNs encoding data along the first versus second decision-dimension were assessed in their ability to distinguish between signals along the first cortical principal component versus the second. This was quantified using the Bhattacharyya distance of the activity among SPNs encoding data along the first decision-dimension versus the second, assuming the subpopulations have mean activities $\mu_1$ and $\mu_2$ and standard deviations $\sigma_1$ and $\sigma_2$, respectively.

$$D_B = \frac{1}{4} \frac{(\mu_1 - \mu_2)^2}{\sigma_1^2 + \sigma_2^2} + \frac{1}{2} \ln\left(\frac{\sigma_1^2 + \sigma_2^2}{2\sigma_1\sigma_2}\right) \quad (29)$$

where:
- $\mu_1$ is the mean activity of activities in the first subpopulation (activity arb. u.)
- $\mu_2$ is the mean activity of activities in the second subpopulation (activity arb. u.)
- $\sigma_1$ is the standard deviation of activities in the first subpopulation (activity arb. u.)
- $\sigma_2$ is the standard deviation of activities in the second subpopulation (activity arb. u.)

For the current analysis, $a_{FSI}$ is arbitrarily set to 1 arbitrary unit and $b_{FSI}$ is arbitrarily set to 0.

For code, see https://github.com/dirkbeck/DM_space_model/blob/main/dynamic_model_and_neural_net/dimension_discrimination_vs_sparsity.m.

### Instance 3: full connectivity and dynamics

In this section, we describe the instance of the model where each cortical neuron projects to each FSI, each FSI projects to each SPN (for dsSPN, isSPN, dmSPN, imSPN), and each cortical neuron projects to each SPN.

This instance of the model leads to a convenient formation of the model as a circuit of fewer elements (one FSI, one dsSPN, isSPN, dmSPN, and imSPN per the four decision-dimensions), with a time component. In this section, we frame this instance mathematically and then describe our related analysis.

To focus on the portions of the circuit we analyze using this instance, we define here the subset of the circuit involving cortex, sSPN, daSNC, and mSPN.

**Defining SPN and daSNC activity.** Here, the activities of SPN and daSNC elements and the weights from sSPN to daSNC are represented as a system of differential equations. Because FSI activity is not measured in the related analyses, cortical activity to pathway $P$ after FSI normalization $x_{i,P}(t)$ is used as input to the system in the equations below.

For a model diagram, see Fig. 5a. Note that daSNC activity of 0 (i.e. average activity) leads to 0 change in SPN activity (due to the 1/2 terms in eqs. 30 and 31). Changes to GPe activity are not modeled here, so the equations are simplified to exclude GPe.

$$\tau \cdot \frac{d s_{sSPN,i,P}(t)}{dt} = -s_{sSPN,i,P}(t) + x_{i,P}(t) - w_{daSNC \to sSPN,i,P} \cdot \left(y_{sSPN,i,P}(t) - \frac{1}{2}\right) \tag{30}$$

$$\tau \cdot \frac{d s_{mSPN,i,P}(t)}{dt} = -s_{mSPN,i,P}(t) + x_{i,P}(t) + w_{daSNC \to mSPN,i,P} \cdot \left(y_{sSPN,i,P}(t) - \frac{1}{2}\right) \tag{31}$$

$$\frac{d}{dt} w_{sSPN \to daSNC,i,P}(t) = \kappa \cdot s_{sSPN,i,P}(t) \tag{32}$$

where:

$$y_{sSPN,i,P}(t) = \frac{1}{1 + \exp(w_{sSPN \to daSNC,i,P} \cdot s_{sSPN,i,P}(t) + RMTg - z_{daSNC,i,P})} \tag{33}$$

- $\tau$ is the time constant related to the decay rate of activity (dimensionless)
- $S_{i,P}(t)$ is the SPN activity (either dsSPN, isSPN, dmSPN, or imSPN) corresponding to decision-dimension $i$ and pathway $P$, as a function of time. (activity arb. u.)
- $t$ is time (seconds)
- $x_{i,P}(t)$ is the cortical activity input, after FSI normalization, to an SPN corresponding to decision-dimension $i$ and pathway $P$ (activity arb. u.)

- $w_{daSNC \to sSPN,i,P}$ is the connection weight from a daSNC neuron corresponding to decision-dimension $i$ and pathway $P$ to an sSPN (either dsSPN or isSPN) corresponding to decision-dimension $i$ and pathway $P$ (dimensionless)
- $y_{i,P}(t)$ is the activity of the daSNC neuron corresponding to decision-dimension $i$ and pathway $P$, as a function of time (activity arb. u.)
- $w_{sSPN \to daSNC,i,P}(t)$ is the connection weight from an sSPN (either dsSPN or isSPN) to a daSNC neuron corresponding to decision-dimension $i$ and pathway $P$, as a function of time (dimensionless)
- $\kappa$ is a coefficient that determines the rate at which the connection from sSPNs to daSNC neurons changes depending on sSPN (dsSPN or isSPN) activity (dimensionless)
- $z_{daSNC,i,P}$ is the bias in the activity of a daSNC neuron corresponding to decision-dimension $i$ and pathway $P$ (activity arb. u.)
- RMTg is the output of RMTg, per eq. 7 (activity arb. u.)

In the current instance of the model, there are $q$ dsSPNs, $q$ isSPNs, $q$ dmSPNs, $q$ imSPNs, $q$ daSNC neurons that each receive projection from a dsSPN, and $q$ daSNC neurons that each receive projection from an isSPN. daSNC neurons corresponding to decision-dimension $i$ and pathway $P$ project to sSPNs and mSPNs of the same $i$ and $P$.

**Defining the decision-space.** A decision-dimension is used to form the decision-space at times when daSNC activity corresponding to the decision-dimension exceeds a threshold:

$$\mathbf{S}_{i,P}(t) = \begin{cases} 0 & y_{i,P}(t) < \text{threshold} \\ 1 & y_{i,P}(t) \geq \text{threshold} \end{cases} \tag{34}$$

where:
- $y_{i,P}(t)$ is the activity of the daSNC neuron corresponding to decision-dimension $i$ and pathway $P$, as a function of time. (activity arb. u.)
- $\mathbf{S}_{i,P}(t)$ is the application of dopamine release (via daSNC activity) to mSPN activity corresponding to decision-dimension $i$ and pathway $P$. The notation here is used to match the notation in **Instance 1**; $S_{i,P}(t)$ is the $(i,i)$ element of the diagonal matrix $\mathbf{S}_P(t)$ whose elements correspond to the weights assigned to the decision-dimensions, similar to in eq. 3. (dimensionless)

**Defining action value.** Action value is defined here like in **Instance 1**, except mSPN activity (for dmSPN and imSPN) is defined as a function of time:

$$v_{j,P} = \frac{1}{1 + \exp(-\boldsymbol{\beta}_{j,P} \, \mathbf{s}_{mSPN,P}(t) - \alpha_{j,P})} \tag{35}$$

where:
- $\boldsymbol{\beta}_{j,P}$ is a matrix of weights from dmSPNs to downstream action value encoding neurons for the direct pathway, or imSPNs to downstream inaction value encoding neurons for the indirect pathway (dimensionless)
- $s_{sSPN,P}$ is the activity of sSPNs corresponding to decision-dimension $i$ and pathway $P$ (activity arb. u.)
- $t$ is time (seconds)
- $\alpha_{j,P}$ is an additive shift corresponding to the neuron encoding action $j$ for the direct pathway or inaction $j$ for the indirect pathway (activity arb. u.)

As in the other instances of the model, there is one neuron encoding each $v_{j,P}$. So, there are $k$ neurons encoding action values and $k$ neurons encoding inaction values. Each of these neurons receives projections from all mSPNs of the corresponding pathway.

**Modeling time-variant input, using Instance 3**. In Fig. 5d-i, simulated responses of dsSPN, isSPN, dmSPN, and imSPN are derived using the forward Euler method with step size 0.001 s.

In Fig. 5d,e, the cortical input to sSPN elements corresponding to each of four example direct pathway decision-dimensions is represented by a vector of length 5001 (corresponding to 0 s to 5 s with increments 0.001 s). Four arbitrary input vectors are used, one corresponding to each plotted decision-dimension:

$$2 + \sin(t),\ 1 + \cos(t),\ \sin(2t),\ \cos(2t) \tag{36}$$

The first vector corresponds to a reward-predominant dimension (shown in green in Fig. 5i). A relatively large positive average value (2 arbitrary units) is assigned to it as an example of an important decision-dimension to a decision. A cost-predominant decision-dimension (second vector) is specified to be important, but less so, and novelty-predominant and location-predominant decision-dimensions (third and fourth vectors) are assigned to be relatively unimportant. The number of decision-dimensions used to form decision-space is averaged across time steps in the 5 s simulation. In **d**, simulations are run across 100 evenly spaced increments of an addition to each vector at all timesteps. In **e**, simulations are run across 100 evenly spaced increments of $z_{\mathrm{daSNC},i}$ (for each pathway, depending on the simulation) in eq. 7 from −1 to 1 arbitrary units.

Figure 5f, g show examples of the response of dsSPN, isSPN, dmSPN, and imSPN elements to different cortical inputs. In Fig. 5f, a cortical input of 10 arbitrary units for 2.5 s is followed by an input of 20 arbitrary units for 2.5 s. In Fig. 5g, a cortical input of 10 arbitrary units for 2.5 s is followed by an input of 0 for 2.5 s.

In the analysis shown in Fig. 5h, cortical inputs of 10 arbitrary units for 2.5 s are followed by cortical inputs with prediction errors incremented by 0.1 from −1 to arbitrary units. These prediction errors are relative to the original cortical input of 10 arbitrary units. For example, for the prediction error of −1, there is a signal of 0 for 2.5 s, and for the prediction error of 1, there is a signal of 20 for 2.5 s. To find the change in the activities of circuit elements, their activities at 2.5 s are subtracted from their activities at 5 s.

Figure 5i shows simulations for an example input with $\kappa = 0$ in eq. 32 versus $\kappa = 0.1$ arbitrary units. The cortical input is as follows: in decision-dimension 1 (e.g. reward-predominant), a cortical signal of 10 arbitrary units for the 0-1.23 s timeframe and elsewhere a signal of 0; in decision-dimension 2 (e.g. cost-predominant), a cortical signal of 10 arbitrary units for the 1.25-2.43 s timeframe and elsewhere a signal of 0; in decision-dimension 3, a cortical signal of 10 arbitrary units for the 2.5-3.73 s timeframe and elsewhere a signal of 0; and in decision-dimension 4, a cortical signal of 10 arbitrary units for the 3.75-5 s timeframe and elsewhere a signal of 0.

For code, see https://github.com/dirkbeck/DM_space_model/blob/main/dynamic_model_and_neural_net/sSPN_DA_mSPN_dynamic_interaction.m.

Parameters are set to common values, chosen arbitrarily, with physiologically accurate signs:
- in eqs. 30, 31: $\tau = 1$
- in eq. 30: $w_{\mathrm{daSNC}\to\mathrm{sSPN,\ direct}} = -1$
- in eq. 30: $w_{\mathrm{daSNC}\to\mathrm{sSPN,\ indirect}} = 1$
- in eq. 31: $w_{\mathrm{daSNC}\to\mathrm{mSPN,\ direct}} = 1$
- in eq. 31: $w_{\mathrm{daSNC}\to\mathrm{mSPN,\ indirect}} = -1$
- in eq. 32: $\kappa = -0.01$
- in eq. 33: RMTg $= 0$
- in eq. 33: $z_{\mathrm{daSNC},i,P} = 0$ for all $i$ and $P$
- in eq. 34: threshold $= 0.5$

Additionally, the following initial conditions, also chosen arbitrarily, are used across analyses:

- $s_{\mathrm{sSPN},i,\,\mathrm{direct}}(0) = s_{\mathrm{sSPN},i,\mathrm{indirect}}(0) = s_{\mathrm{mSPN},i,\,\mathrm{direct}}(0) = s_{\mathrm{mSPN},i,\,\mathrm{indirect}}(0) = 0$
- $w_{\mathrm{sSPN}\to\mathrm{daSNc},i,\,\mathrm{direct}}(0) = w_{\mathrm{sSPN}\to\mathrm{daSNc},i,\,\mathrm{indirect}}(0) = 1$

## Shifts in the decision-space in a reinforcement learning task

Here, we run simulations in a Reinforcement Learning (RL) environment to compare the decision-space model with a traditional model of reinforcement learning in the basal ganglia. In a traditional model, dopamine signals RPEs, whereas in the decision-space model, it conveys the decision-space (Fig. 6b). To compare the models, we record RPEs during a very simple RL task and compare it with times at which the decision-space is likely to shift.

**Constructing the RL task**. In the task, an agent moves from a starting position (referred to here as a "state" to align with standard RL terminology) to an intermediate state where it can move left or right to receive one of the two cost/reward outcomes, and then moves to receive one of the outcomes. The agent learns over the course of 20 episodes via a classical Q-learning algorithm (for reinforcement learning foundation, see Supplementary Note 9). Q-values are updated based on the Bellman equation, simplified to exclude the values of future states (because we examine values at outcome state):

$$V(s_t) = V(s_t) + \alpha \cdot [r_t - V(s_t)] \tag{37}$$

where:
- $V(S_t)$ is the value of the current state $S_t$.
- $\alpha$ is the learning rate (set arbitrarily to 0.5).
- $r_t$ is the reward or cost received in $S_t$.

The exploration rate is arbitrarily set to 0.5, and otherwise the strategy is epsilon-greedy. That is, the agent approaches left 25% of the time, right 25% of the time, and 50% of the time chooses the direction it has assigned to the highest Q-value to.

Using the same notation, the RPE at timestep $t$, $\delta_t$, is the difference between the reward observed and the Q-value (note that similar to the formula above, the values of future states have been excluded because we measure RPEs at terminal states):

$$\delta_t = r_t - V(s_t) \tag{38}$$

**Recording choice over a T-maze RL simulation**. In the simulation plotted in Supplementary Fig. 7k, the agent moves through the simple RL-environment. When the agent reaches one of the two terminal states (for either reward/cost outcome), the Q-value of the outcome is updated per eq. 37. In the simulation, for the first 10 episodes, the agent receives +1 reward for approaching right and −1 reward for approaching left. Then for the final 10 episodes, the agent receives −1 reward for approaching right and +1 reward for approaching left.

For code, see https://github.com/dirkbeck/DM_space_model/blob/main/importance_and_RPEs/Tmaze_RL.py.

**Comparing RPEs with episodes when a shift in the decision-space might occur**. In Supplementary Fig. 7l, RPEs are compared with shifts in the decision-space that might occur were a rodent to perform the task. The analysis is performed using the same simulation as in Supplementary Fig. 7k.

RPEs are calculated per eq. 38. The episodes corresponding to likely sudden "shifts" in the decision-space, as plotted, follow the rules:
1) A reward decision-dimension is used, and only used, when a reward appears.
2) A cost decision-dimension is used, and only used, when a cost appears.

3) If a reward (or cost) is expected, then there is no sudden "shift" into using that decision-dimension because such a shift will have occurred earlier than the outcome state.

For code, see https://github.com/dirkbeck/DM_space_model/blob/main/importance_and_RPEs/Tmaze_RL.py.

### Movement of circuit activity across multiple trials
Here, we model changes in circuit activity between trials. We begin by forming advantage and cost functions that guide the realignment of the circuit. Using these, we explore how vulnerability versus resilience in disorder formation could be interpreted through the lens of the model.

**Defining advantage and cost of circuit activity.** 'Advantage' is defined here as the ability of the circuit to produce beneficial decision-spaces at a certain activity. The goal of the sSPN→GPi→LHb→RMTg→daSNC circuit in the model is to produce preferred decision-spaces for action valuation (Fig. 6a–d). For instance, in a laboratory environment when an animal routinely makes a choice to approach depending on reward level, a one-dimensional direct pathway decision-space with a reward dimension may be helpful. During a decision, it may make sense for this animal to reach a circuit activity where forming a one-dimensional direct pathway decision-space is probable.

We represent this logic mathematically as a function of an $n$-element circuit $\{X_1, X_2, \ldots, X_n\}$. In our analysis, we focus on either FSI and sSPN, holding the rest of the circuit elements fixed at default values (see **Common parameters**); or sSPN, LHb, and daSNC, holding the rest of the circuit elements fixed at default values. The advantage of a certain circuit activity is defined as a weighted sum of probabilities the direct pathway decision-space occurs and the benefit of forming each decision-space:

$$\text{advantage}(X_1 = x_1, X_2 = x_2, \ldots, X_n = x_n) = \sum_{l=1}^{2^q} \text{score}_l \cdot \quad (39)$$
$$\text{P}(\text{space}_l | (X_1 = x_1, X_2 = x_2, \ldots, X_n = x_n))$$

where:
- $\{X_1, X_2, \ldots, X_n\}$ are elements of the circuit with activities $x_1, x_2, \ldots, x_n$ (activity arb. u.)
- $q$ is the count of decision-dimensions
- $\text{score}_l$ is a coefficient corresponding to the preference for a given direct pathway decision-space (dimensionless)

The probability each decision-space forms is derived from probability its decision-dimensions individually are used during the decision (daSNC$_i$ in eq. 2, here notated as $d_i$):

$$\text{P}(\text{formation of a decision-space } l)$$
$$= d_1(1 - d_1) d_2(1 - d_2) \cdots d_m(1 - d_m) \quad (40)$$

The rationale for this formation of advantage is theoretical. Elsewhere, we show that direct pathway decision-spaces of different dimensionality are beneficial (and may be used by rodents) for tasks of different difficulties (Fig. 3). We also show that certain decision-spaces are beneficial with certain levels of cortical noise (Supplementary Fig. 6) and for obtaining different types of action values (Fig. 6c–f). We represent this as an assignment of greater value to certain decision-spaces, given external and internal contexts and the task at hand.

'Cost', here, is defined as the difference between the circuit activity and a baseline activity. For most scenarios, the circuit might be best served searching for the circuit activity with the highest advantage. However, there is an obvious counterexample: it could be that it is easiest to form preferred decision-spaces at extremely unusual circuit activity (e.g. very high sSPN, very high LHb, very high daSNC), and only slightly more difficult to form that decision-space at closer to average circuit activity (average sSPN, low LHb, average daSNC). It may be more advantageous for the circuit to shift to the latter activity.

Therefore, we introduce a cost function to form 'net advantage'. We then use net advantage to define the circuit activities that are the most beneficial. The concept of baseline circuit activity is introduced here in order to define cost. This can be interpreted as the circuit activity outside of decision-making.

$$\text{cost}(X_1 = x_1, X_2 = x_2, \ldots, X_n = x_n) = \| [x_1 \, x_2 \, \ldots \, x_n]^\mathsf{T}$$
$$- [x_{1,\,\text{baseline}} \, x_{2,\,\text{baseline}} \, \cdots \, x_{n,\,\text{baseline}}]^\mathsf{T} \| \quad (41)$$

where:
- $\{X_1, X_2, \ldots, X_n\}$ are elements of the circuit with activities $x_1, x_2, \ldots, x_n$ (activity arb. u.)
- Outside of decision-making, $\{X_1, X_2, \ldots, X_n\}$ have baseline activities $x_{1,\,\text{baseline}}, x_{2,\,\text{baseline}}, \ldots, x_{n,\,\text{baseline}}$ (activity arb. u.)

Net advantage is defined as advantage minus cost multiplied by a constant:

$$\text{net advantage}(X_1 = x_1, \ldots, X_n = x_n) = \text{advantage}(X_1 = x_1, \ldots, X_n = x_n)$$
$$- \text{constant} \cdot \text{cost}(X_1 = x_1, \ldots, X_n = x_n) \quad (42)$$

where:
- Functions for reward and cost are taken from eqs. 39 and 41, respectively
- constant alters the weight given to cost compared to reward (dimensionless)

**Visualizing advantage, cost, and net advantage.** Figure 7b shows an example of how a circuit forms advantage per eq. 39. The advantage scores score$_i$ are randomly generated (normal distribution, mean 0, standard deviation 1) for each of the 16 possible decision-spaces formed from four decision-dimensions. sSPN and daSNC activity are incremented across a 10×10 grid of sSPN and daSNC activities which each range from 0 to 10 arbitrary units. The activities of other circuit elements are set to default values in **Instance 1** of the model (see **Common parameters**).

Cost in Fig. 7c is formed using eq. 41. Distance from the circuit baseline point is measured as Euclidean distance in the two plotted decision-dimensions.

Net advantage in Fig. 7d is calculated per eq. 42. For this example, the cost coefficient constant = 1 (dimensionless coefficient).

Figure 7f similarly shows net advantage but after incrementing sSPN, LHb, and daSNC activities. Similar to in the example in Fig. 7d, a set of scores are independently randomly generated and constant in eq. 42 is set arbitrarily to 1 (dimensionless coefficient).

For code, see https://github.com/dirkbeck/DM_space_model/blob/main/circuit_trajectories/advantage_cost_net_advantage_example.m.

**Defining direction of circuit movement.** The direction of movement is defined as a search for the optimal circuit activity to produce advantageous direct pathway decision-spaces. To understand how a circuit governed by eq. 42 might adjust over the course of multiple trials, we relate the adjustment of circuit activity to net advantage. We specify that the circuit adjusts so that it can more easily reach

advantageous decision-spaces. This constitutes a movement in the baseline activity of eq. 42.

Thus, the circuit adapts in the direction of the gradient of net advantage.

$$\frac{\Delta [x_{1,\text{baseline}} x_{2,\text{baseline}} \ldots x_{n,\text{baseline}}]^{\mathsf{T}}}{\text{trial}} = \text{rate} \cdot \nabla \text{net advantage}(X_1 = x_1, \ldots, X_n = x_n)$$

$$(43)$$

where:

- $\{X_1, X_2, \ldots, X_n\}$ are elements of the circuit with activities $x_1, x_2, \ldots, x_n$ (activity arb. u.)
- Outside of decision-making, $\{X_1, X_2, \ldots, X_n\}$ have baseline activities $x_{1,\text{baseline}}, x_{2,\text{baseline}}, \ldots, x_{n,\text{baseline}}$ (activity arb. u.)
- rate is a coefficient that affects the speed of movement of the circuit per trial (trial$^{-1}$)

**Effect of initial circuit activity on future trials.** In Fig. 7e, g, Supplementary Fig. 9a, several examples illustrate trajectories of circuit movement, as defined in eq. 43.

In these analyses, we assume that decision-dimensions are assigned equal importance by sSPN (i.e. in eq. 1, $b_{\text{sSPN}} = 0$). The probability that an individual decision-dimension is used to form decision-space is calculated using eq. 2. Advantage scores score$_i$ are randomly generated (normal distribution, mean = 0, standard deviation = 1 arbitrary unit) for each of the 16 possible decision-spaces created from four dimensions. The value of constant in eq. 42 is set to 1 arbitrary unit. For each increment of a 15 × 15 grid of sSPN and FSI (Fig. 7e) or 15 × 15 × 15 grid of sSPN, LHb, and daSNC (Fig. 7g, Supplementary Fig. 9a), net advantage is calculated for the scenario where that activity is the circuit baseline, i.e. $X_1 = x_{1,\text{baseline}}, X_2 = x_{2,\text{baseline}}, \ldots,$ $X_n = x_{n,\text{baseline}}$. The gradient of this grid is approximated via MATLAB's gradient() routine. Trajectories are integrated from the gradient via MATLAB's streamline(), which uses the forward Euler method.

In the upper panel of Fig. 7e, circuit baseline points are shown for each increment of Euler's method. The bottom panel shows the trajectories of 20 randomly selected points on the [0, 5] arbitrary units range for sSPN and FSI.

In Fig. 7g, starting points are selected to form a 5 × 5 × 5 grid incremented along sSPN, LHb, and daSNC axes. At each point in a trajectory line, the probability a dimension is used to form decision-space is calculated via eq. 2. Then, during plotting, these calculated probabilities are interpolated using MATLAB's patch() routine.

In Supplementary Fig. 9a, the gradient is plotted using MATLAB's streamslice(). Then the trajectories of five randomly selected are plotted.

For code, see https://github.com/dirkbeck/DM_space_model/ blob/main/circuit_trajectories/sSPN_DA_LH_trajectories.m and https://github.com/dirkbeck/DM_space_model/blob/main/circuit_ trajectories/sSNC_DA_LH_trajectories_examples2.m.

**Effect of altered advantage score on future trials.** In Supplementary Fig. 9b, c, we examine the effect of a change to advantage score$_l$ in eq. 39 as a modeled task is being performed.

The circuit adjusts over 100 trials. At the beginning, advantage score is set to 1 (dimensionless coefficient) for the non-D decision-space and 0 for every other decision-space. Over the first 20 and the last 10 time-steps (between the dashed lines on the plots), advantage score is incremented for the 4D decision-space but not others. Over time steps 21-90, advantage score is incremented for the non-D decision-space.

In the plot, the value of each decision-space is divided by the total value of all decision-spaces to form a ratio.

For code, see https://github.com/dirkbeck/DM_space_model/blob/ main/circuit_trajectories/changing_space_value_between_trials.m.

## Reasoning behind the FSI model

In eq. 1, scaling of cortical data serves the important functional role of normalizing $\mathbf{x}_P$ so that the circuit can make sense of data coded across the wide range of firing rates that might be observed in the cortex (Supplementary Fig. 2a; for a discussion of the experimental literature, see Supplementary Note 10). Per the experimental literature reviewed in Supplementary Note 10, many cortical neurons synapse to one FSI and the cortical to FSI connection is excitatory. Physiological analysis of connected FSI and cortical neurons in the current work shows that FSI activity scales linearly with cortical activity (Supplementary Fig. 4k, l).

In the model, the parameter $a_{\text{FSI}}$ is related to the strength of connection between cortex and FSI. The parameter $b_{\text{FSI}}$ is related to firing rate of FSI when $\mathbf{x}_P = 0$.

The algebraic operator that best represents this inhibition is less clear: it could be thought of as a subtraction or as a division depending on the experimental evidence used to model (for a discussion of the experimental literature, see Supplementary Note 10).

A subtraction possibility:

$$\mathbf{s}_{\text{sSPN}}(a_{\text{FSI}}) = \mathbf{x}_P - a_{\text{FSI}} \qquad (44)$$

A division possibility:

$$\mathbf{s}_{\text{sSPN}}(a_{\text{FSI}}) = \frac{\mathbf{x}_P}{a_{\text{FSI}}} \qquad (45)$$

where:

- $a_{\text{FSI}}$ is the weight of cortex→FSI connection (dimensionless)
- $\mathbf{x}_P$ is the activities of the cortical neurons in a given pathway $P$ (activity arb. u.)

The analytic relationship used here leads to a convenient interpretation from the perspective of data processing. It can also be thought of as approximating subtraction or division depending on the scale of $a_{\text{FSI}}$. Below, we show a series approximations of eq. 1 as a function of connection strength from cortex, with input from cortex and other parameters fixed.

Substitution in terms of $a_{\text{FSI}}$:

$$\mathbf{s}_{\text{sSPN}}(a_{\text{FSI}}) = \frac{1}{a_{\text{FSI}} \cdot \|\mathbf{x}_P\|_2 + b_{\text{FSI}}} \mathbf{x}_P \mathbf{W} + b_{\text{sSPN}} \qquad (46)$$

Truncated Taylor series at $a_{\text{FSI}} = 0$:

$$\mathbf{s}_{\text{sSPN}}(a_{\text{FSI}}) = \mathbf{b}_{\text{sSPN}} + \frac{\mathbf{x}_P \mathbf{W}}{b_{\text{FSI}}} - \frac{\|\mathbf{x}_P\|_2 \mathbf{x}_P \mathbf{W}}{b_{\text{FSI}}^2} \cdot a_{\text{FSI}} + \mathcal{O}(a_{\text{FSI}}^2) \qquad (47)$$

Truncated Laurent series at $a_{\text{FSI}} = \infty$:

$$\mathbf{s}_{\text{sSPN}}(a_{\text{FSI}}) = b_{\text{sSPN}} + \frac{\mathbf{x}_P \mathbf{W}}{\|\mathbf{x}_P\|_2} \cdot \frac{1}{a_{\text{FSI}}} - \mathcal{O}(\frac{1}{a_{\text{FSI}}^2}) \qquad (48)$$

Thus, when FSI is weakly connected to cortex, the formula somewhat resembles a subtractive operation, and when FSI is strongly connected to cortex, the formula somewhat resembles a division operation. The ratio between the $a_{\text{FSI}}$ and $b_{\text{FSI}}$ parameters is important here (the first series, before truncation, converges when $\frac{|b_{\text{FSI}}|}{a_{\text{FSI}}} > \|\mathbf{x}_P\|_2$ and the second, before truncation, when $\frac{|b_{\text{FSI}}|}{a_{\text{FSI}}} < \|\mathbf{x}_P\|_2$).

## Inferring decision-space from SPN activity and choice

The decision-space can be inferred from environmental or experimental inputs in conjunction with decision-making data (Supplementary Fig. 2q, r). The method here requires that many decision-making experiments have been run with a similar apparatus but different parameters (for instance, light level), and that average sSPN activity has been measured during the decisions. The parameter data is stored

in a matrix $\mathbf{X} \in \mathbb{R}^{n \times p}$, where $n$ here is a separate trial with separate inputs (rather than different inputs across time steps, as in the description of **Instance 1**) and $P$ is the number of features of the experiment that may be encoded by cortex, for instance temperature, music volume, or light level (similar to the description in **Instance 1**).

The process involves three steps:

**Labeling sessions.** Recorded decisions are labeled using attributes of the process by which the decision was made. This might be achieved in a rodent task that measures response to music volume, for instance, by clustering sessions based on heart rate and distance traveled.

**Constraints based on SPN activity.** The sessions are split by label. $\mathbf{X}$ is standardized such that each column has a mean of 0 and standard deviation 1. Then a linear regression is run on each labeled subset $\mathbf{X}_l$ to find coefficients $\mathbf{b}_l$ that map those observations to predicted SPN activities (averaged across SPN decision-dimensions) $\hat{\mathbf{y}}_l$ for each session in the subset (here, the subscript $l$ is used to indicate all elements of the subset):

$$\hat{\mathbf{y}}_l = \mathbf{X}_l \mathbf{b}_l + b_0 \qquad (49)$$

where:
- $\hat{\mathbf{y}}_l$ is predicted SPN activities
- $\mathbf{X}_l$ is a subset of experimental parameter data corresponding to one label
- $\mathbf{b}_l$ and $b_0$ are the linear regression coefficients that map the experimental parameters to sSPN activity

From the calculated $\mathbf{b}_l$, we can guess the principal axis dimensions that make up decision-space $\mathbf{w}_1, \mathbf{w}_2, \ldots, \mathbf{w}_q$ and the presence of each decision-dimension $e_{li} \in \{0, 1\}$ in each labeled subset. During this process, we consider $\mathbf{b}_l$ as the sum of the dimensions in decision-space in the subset:

$$\mathbf{b}_l = \sum_{i=1}^{q} e_{l,i} \, \mathbf{w}_i \qquad (50)$$

Using this framework, we can form a rule which we can use to assign labels to decision-dimensions and hypothesize $\mathbf{w}_1, \mathbf{w}_2, \ldots, \mathbf{w}_q$:

If $\mathbf{b}_A$ has high correlation with $\mathbf{b}_B$, then one of $\mathbf{b}_A$ or $\mathbf{b}_B$ must correspond to a higher-dimensional decision-space that also includes the dimensions of the other.

For example, we might have data with labels A, B, C, and D and corresponding $\mathbf{b}_A, \mathbf{b}_B, \mathbf{b}_C, \mathbf{b}_D$, where $\mathbf{b}_A$ is correlated with $\mathbf{b}_D$, $\mathbf{b}_B$ is correlated with $\mathbf{b}_D$, and the other possible pairs uncorrelated. It follows from the rule above that $\mathbf{b}_D$ is at least a two-dimensional decision-space including dimensions from $\mathbf{b}_A$ and $\mathbf{b}_B$, and that $\mathbf{b}_A$ and $\mathbf{b}_B$ are at least one-dimensional decision-spaces. So, we would hypothesize that there are two SPN-encoded decision-dimensions used during the decision, $\mathbf{w}_1$ and $\mathbf{w}_2$, and that subsets A and D use $\mathbf{w}_1$, subsets B and D use $\mathbf{w}_2$, and subset C does not use either.

The logic to construct these constraints are as follows. When $e_{l1} = e_{l2} = \ldots = e_{lq} = 0$ (i.e. a zero-dimensional decision-space), $\hat{\mathbf{y}}_l = b_0$. The coefficients assigned to the zero-dimensional decision-space $\mathbf{b}_{l: \text{0D space}}$ should have low correlation with any of the axes $\mathbf{w}_1, \mathbf{w}_2, \ldots, \mathbf{w}_q$. Therefore, it is expected that we find one $\mathbf{b}_l$ that has low correlation with the other $\mathbf{b}_l$, and we can label this $\mathbf{b}_{l: \text{0D space}}$. Further, when $e_{li}$ for one $i$ is equal to 1 and for all other $i$ is equal to 0 (i.e. a 1D decision-space), $\hat{\mathbf{y}}_l = \mathbf{w}_{i \text{ in space}} + b_0$. This decision-space will have high correlation with $\mathbf{w}_{i \text{ in space}}$ but low correlation with other $\mathbf{w}_i$. Therefore, it is expected that we find several $\mathbf{b}_{l: \dim i \text{space}}$ that are not

correlated with one another, or $\mathbf{b}_{l: \text{0D space}}$, but may be related through the multi-dimensional decision-spaces in subset C. Finally, when $e_{li}$ is equal to 1 for multiple $i$, $\mathbf{b}_l$ is calculated as a sum per eq. 50. Therefore, it is expected that some $\mathbf{b}_{l: \dim i \cap \dim j \text{space}}$ are linear combinations of $\mathbf{b}_{l: \dim i \text{space}}$ and $\mathbf{b}_{l: \dim j \text{space}}$.

The example shown in Supplementary Fig. 2r uses simulated data with 2 experimentally observed features (for instance, temperature and music volume) and 100 observations. $\mathbf{X}$ was constructed as a matrix of i.i.d. Gaussian variables with mean 0 and standard deviation 1. The ground-truth principal component matrix $\mathbf{W}$ was constructed as a $2 \times 2$ matrix of i.i.d. Gaussian variables with mean 0 and standard deviation 1. Then, for each of the 100 observations, a decision-space was randomly assigned in a reference dataset. Each decision-dimension for each observation was treated as an i.i.d. uniform variable, and if the variable corresponding to the observation and the decision-dimension exceeded a value (here, 0.5), then the dimension was considered as incorporated in decision-space. The randomly generated decision-spaces were each assigned their own label. Then a simulated SPN activity was created for each of the 100 observations was created by multiplying the ground-truth $\mathbf{W}$ by the decision-space used in each observation, similar to in eq. 1. i.i.d. Gaussian noise (mean 0, standard deviation 0.5) was then added to create simulated SPN activity observations. Using this simulated data, we ran a linear regression for each labeled subset via MATLAB's fitlm() routine. The linear regression fits are plotted as surfaces in Supplementary Fig. 2r. The slopes, with respect to temperature and music volume, were, for label A, −0.15 and 0.48, respectively; for label B, −437.90 and −9.72, respectively; for label C, −9.01 and −274.51, respectively; and for label D, −155.45 and −199.83. Slopes for label A are relatively small, so it is assigned to "decision-space not formed" (matching the reference dataset). Of the remaining labels, label D is closer to an additive combination of labels B and C than other permutations, so label D is assigned the "2D decision-space" (matching the reference dataset), while labels B and C are each considered 1D decision-spaces (matching the reference dataset).

For code, see https://github.com/dirkbeck/DM_space_model/blob/main/model_overview/dimensionality_from_SPN_activity.m.

**Tests using choice.** We can then combine the constraints in Step 2, developed using SPN activity, with an analysis of choice given the hypothesized SPN-encoded decision-dimensions. In Step 2, a set of $\mathbf{w}_1, \mathbf{w}_2, \ldots, \mathbf{w}_q$ are hypothesized. Here, choice at different levels of those $\mathbf{w}_i$ are compared across the labeled subsets. The hypothesis in Step 2 is supported if choice, for each labeled subset, is correlated with the decision-dimensions hypothesized to be used to form decision-space but not correlated with the decision-dimensions hypothesized not to be used to form decision-space.

An example is shown in Supplementary Fig. 2r. For the case where decision-space is not formed, choices do not correlate with any hypothesized SPN-encoded decision-dimension. For a 1D decision-space, choices correlate with one hypothesized SPN-encoded decision-dimension. For a 2D decision-space, choices correlate with two hypothesized SPN-encoded decision-dimensions.

In the plotted examples, we plot simulated choices in each of the four decision-spaces used in the analysis. Choices are simulated from reference dataset (i.e. absent noise added during simulation) average sSPN activities by session $y$ by treating $Z = \frac{1}{1 + \exp(-y + \text{i.i.d. Gaussian noise})}$ as a random variable, where $Z < 0.5$ corresponds to a "turn left" action and $Z \geq 0.5$ to a "turn right" action. The threshold of 0.5 is chosen because it represents the expected value of $Z$ at average sSPN activity ($y = 0$). In the plots, we interpolate possible subject values from choice at different combinations of the two decision-dimensions $\mathbf{w}_1$ and $\mathbf{w}_2$, as derived in Step 2. For each action for each labeled subset, a logistic regression is used to convert actions to the value of the action across the grid. As expected, observations of label A, assigned to "decision-

space not formed" have little correlation with either of the putative decision-dimensions derived in Step 2; observations of labels B and C, assigned to 1D decision-spaces, are correlated with one putative decision-dimension but not the other; and observations of label D, assigned to the "2D decision-space," are correlated with both.

For code, see https://github.com/dirkbeck/DM_space_model/blob/main/model_overview/dimensionality_from_SPN_activity.m

### Testing the model through analysis of neural data

As a further test of the decision-space model, we examined the relationships between behavior and neural activity in tasks that required different reward versus cost dimensions. The analysis, new to the current work, was performed using the Corticostriosomal Circuit Stress Experiment database (published with Friedman et al., 2017). We found more functionally connected sSPN and mSPN in tasks that were difficult. We then analyzed cortex, FSI, sSPN, and mSPN during these tasks and found evidence of dimensionality reduction from cortical neurons to SPNs.

**Defining decision difficulty by task.** We defined decision difficulty through deliberation time, calculated as the time between when the door opened during the T-maze task and when the animal made a movement between one end of the maze or the other. Deliberation time distributions were analyzed for each trial group (for control and stress: cost-benefit cost, benefit-benefit, cost-cost, non-conflict cost-benefit). Skewness was calculated using MATLAB's skewness() routine. Example distributions are shown in Supplementary Figs. 3b, 4a, b (6 animals, 35 sessions) and summaries across groups in Supplementary Figs. 3d, 4c, d (14 rats, 249 sessions).

For instructions to run code, see https://github.com/dirkbeck/DM_space_model/blob/main/Cross%20Correlation%20Pattern%20Counts/createPaperFigures.m.

**Fitting the modeled deliberation times.** To replicate the deliberation time distributions with a computational model, we leveraged a diffusion model where two functions would both start at zero and would continue towards either a positive or negative pre-defined threshold that represented either performing an action (positive threshold) or not performing an action (negative threshold). The first threshold that was reached by either function was which decision we considered as selected. The x-axis was the progression of time as the functions ran. The deliberation time for this model was the amount of time (in seconds) it took for the first function to reach its threshold. To match the experimental distribution times analyzed in Supplementary Fig. 3b, we ran the drift diffusion process adjusting drift rate, the thresholds, and noise until the modeled deliberation time distribution had a similar median and skew to experimental data from Supplementary Fig. 3c (https://doi.org/10.7910/DVN/SMKW0I). An additional 1.5 seconds was uniformly added to modeled distribution times.

For instructions to run code, see https://github.com/dirkbeck/DM_space_model/blob/main/Cross%20Correlation%20Pattern%20Counts/createPaperFigures.m.

**Connected SPNs through cross-correlation.** We used the Corticostriosomal Circuit Stress Experiment database to identify sSPN and mSPN among recorded neurons in the striatum. For details, see the Supplemental Materials & Methods of Friedman et al. (2017). 14785 cells across 14 control animals were analyzed. The total number of identified striosomal and matrix neurons per task: NCB = non-conflict cost-benefit (14 sSPNs, 260 mSPNs), CC = cost-cost (46 sSPNs, 400 mSPNs), CBC = cost-benefit conflict (84 sSPNs, 717 mSPNs), BB = benefit-benefit easy (50 sSPNs, 515 mSPNs, chocolate milk concentration <50), BB = benefit-benefit difficult (33 sSPNs, 731 mSPNs, chocolate milk concentration >=50).

We binned the firing rates of sSPN and mSPN during the −3s to 3 s window across all tasks into 1·5 bins. We used these firing rates to determine cross-correlation (MATLAB's xcorr()) between sSPN and mSPN recorded in the same session. Correlated pairs were defined as paired sSPN and mSPN that had a linear regression fit with correlation squared (MATLAB's corrcoef()) > 0.5 and significance p < 0.04. To obtain the percentage of significantly correlated neurons, we counted the number of pairs that met the threshold for correlation and divided by the total number of identified pairs. Examples are plotted in Supplementary Fig. 3e,f and counts in Supplementary Fig. 3g.

To determine the significance threshold for the counts of correlated pairs, we shuffled the sSPN and mSPN pairs across the database. We then performed the process above on the shuffled data. Using these shuffled pairs, we formed a distribution of correlations that might happen by chance. The threshold of significance was set to the 3 standard deviation mark among this distribution of shuffled pairs.

For instructions to run code, see https://github.com/dirkbeck/DM_space_model/blob/main/Cross%20Correlation%20Pattern%20Counts/runMe.m.

**Connected SPNs through Granger causality.** sSPN and mSPN pairs were identified similarly to the previous section. 14785 cells across 14 control animals were analyzed, and 25758 cells across 9 stress animals. The total identified number of striosomal and matrix cells for each of the tasks is the following: Control CC = cost-cost (46 sSPNs, 400 mSPNs), Control BB = benefit-benefit (83 sSPNs, 1246 mSPNs), Control CBC = cost-benefit conflict (84 sSPNs, 717 mSPNs), Stress CBC = cost-benefit conflict (41 sSPNs, 898 mSPNs, Stress BB = benefit-benefit (156 sSPNs, 2813 mSPNs).

Then firing rates were determined over the span of the session by binning each trial into bins of 100 ms. Granger causality (MATLAB's gctest()) was run on these sSPN and mSPN firing rate pairs to determine whether each pair was functionally connected. In control animals, 92 connected pairs were identified for the cost-benefit conflict task, 77 connected pairs for the cost-cost task, 1 connected pair for the non-conflict cost-benefit task, and 116 pairs for the benefit-benefit task. In stress animals, 110 connected pairs were identified for the cost-benefit conflict task and 292 connected pairs for the benefit-benefit task.

Next, for each of the trials with a functionally connected sSPN and mSPN pair, the class of motif was determined as either sSPN excited / mSPN excited, sSPN excited / mSPN inhibited, sSPN inhibited / mSPN excited, or sSPN inhibited / mSPN inhibited. These classes were formed by identifying whether, separately, the neurons were excited, inhibited, or neither in each of 5 blocks per each trial ([−15s, −3s], [−3s 0 s], [0 s 2.5 s], [2.5 s, 4.5 s], [4.5 s 20 s]). A motif was counted if, in a certain block, each neuron was either excited or inhibited.

Excited or inhibited classifications were determined through inter-spike interval analysis, plotted in Supplementary Fig. 3h-j. The median inter-spike interval for each neuron across the full trial was calculated. A neuron was considered inhibited during the periods of the trial when the inter-spike interval exceeded the median. Conversely, a neuron was considered excited during periods when the inter-spike interval fell below median. Excited blocks were defined as blocks where excitation time exceeded inhibition time by 10%. Inhibited blocks were defined, oppositely, as blocks where inhibition time exceeded excitation time by 10%. Blocks that reached neither threshold were not classified.

For instructions to run code, see https://github.com/dirkbeck/DM_space_model/blob/main/Cross%20Correlation%20Pattern%20Counts/createPaperFigures.m.

**Analyzing neural dimensionality reduction.** In the analysis plotted in Fig. 3f, using data from the Corticostriosomal Circuit Stress Experiment database, we analyzed firing rates of 1) PL neurons that project to

striatum (i.e. sSPN-projecting cortex, 221 sessions, 2-47 neurons per session), 2) FSIs (96 sessions, 2-12 neurons per session), 3) sSPNs (27 sessions, 2-9 neurons per session), and 4) mSPN (13 sessions, 2-6 neurons per session). For methods of classification, see Friedman et al. (2017). Spike data was converted to firing rates by separating the spikes into 5-10 bins.

Covariance between the neurons was determined from the firing rates of simultaneously recorded neurons over time. For each trial, the effective correlation[49] was calculated from the activities of the $p$ neurons over time:

$$\text{effective correlation} = 1 - \|\text{Corr}(X)\|^{1/p} \tag{51}$$

Effective correlations were then averaged across trials. Confidence intervals were determined from standard error between the sessions. p-values were calculated using a two-sample t-test (MATLAB's ttest2 routine).

For instructions to run code, see https://github.com/dirkbeck/DM_space_model/blob/main/neuron_pair_analysis/run_me.m.

**Changes to choice after adding cost to a reward offer.** In the analysis plotted in Supplementary Fig. 4j, we sought to define choice patterns before and after stress when a small cost was added to a reward offer. To do this, we analyzed choice data from the CBC task before and after stress (i.e. reward and a small cost) and in the BB task before and after stress (i.e. only reward). We recorded the approach percentage across sessions and then averaged these between groups. Data from 17 rodents across 38 sessions is used for Control CBC, data from 13 rodents across 24 tasks for Stress CBC, data from 23 rodents across 114 tasks for Control BB, and data from 14 rodents across 116 tasks for Stress BB.

For data, see https://github.com/dirkbeck/DM_space_model/blob/main/disorder_hypotheses/experimental_data_analysis_choice_before_after_stress.xlsx.

**Analyzed cortex-FSI connectivity.** Using data from the Corticostriosomal Circuit Stress Experiment database (published with Friedman et al. (2017)), firing rates of striatum-projecting prelimbic cortex neurons (i.e. sSPN-projecting cortex) and FSIs were analyzed. For details regarding neuron classification, see Friedman et al. (2017). Spike data was converted to firing rates by separating data into 5-10 bins. A linear regression of the form $a\,x + b$ was fit through the firing rates. Examples are plotted in Supplementary Fig. 4k, l and averages of the square of Pearson correlation coefficient across neuron pairs and the $a$ slope parameter from the regression fit are plotted in Supplementary Fig. 4o, p. 78 neuron pairs across 7 rodents were analyzed before stress and 37 neuron pairs were analyzed across 4 rodents after stress.

For instructions to run code, see https://github.com/dirkbeck/DM_space_model/blob/main/neuron_pair_analysis/run_me.m.

**Modeled cortex-FSI connectivity after stress.** Connected cortical neurons and FSI show increased correlation and reduced slopes between their firing rates after stress (Supplementary Fig. 4k, l, o, p). To understand how this impacts SPNs, we modeled two changes to the cortex→FSI connectivity that might produce the experimental results (Supplementary Fig. 4q, r). A first factor, connection weight $w_q$, measured the strength of connection between the $q$th connected cortical neuron and the FSI, that is, how strongly FSI would respond to an increase in activity in the connected cortical neuron. A second factor, number of connections $c$, measured the number of connected cortical neurons to the FSI.

We solved for correlation between the neurons and slope using identities that link covariance, correlation, and slope. This process assumes that FSI receives input only from the cortex.

Note that, to align with the indexing used in **Instance 2** (e.g. eq. 25), we use the index $q$ for cortical neurons. In eq. 55, we use $q_1$ and $q_2$ to refer to different cortical neurons.

$$\text{correlation(cortex 1, FSI)} = \frac{\text{cov(cortex 1, FSI)}}{\sqrt{\text{var(cortex 1)} \cdot \text{var(FSI)}}} \tag{52}$$

$$\text{slope(cortex 1, FSI)} = \frac{\text{cov(cortex 1, FSI)}}{\text{var(FSI)}} \tag{53}$$

$$\text{cov(cortex 1, FSI)} = w_1 \text{var(cortex 1)} + \sum_{q=2}^{c} w_q \cdot \text{cov(cortex 1, cortex } q) \tag{54}$$

$$\text{var(FSI)} = \sum_{q=1}^{c} w_q{}^2 \text{var(cortex } q) + \sum_{q_1=1}^{c} \sum_{\substack{q_2=1 \\ q_1 \neq q_2}}^{c} w_{q_1} w_{q_2} \text{cov(cortex } q_1, \text{ cortex } q_2) \tag{55}$$

For simplicity, we assumed in our analysis that variance is equal across cortical neurons projecting to the FSI (i.e. var(cortex 1) = var(cortex 2) = ... = var(cortex $c$)) and that the covariance between those neurons is equal (i.e. cov(cortex 1, cortex $q$) = cov(cortex 1, cortex 2) = cov(cortex 1, cortex 3) = ... = cov(cortex 1, cortex $c$)). So, in the analysis plotted in Supplementary Fig. 4q, r, we substitute uniform variance and covariance to obtain:

$$\text{correlation(cortex 1, FSI)} = \sqrt{\frac{\text{var(cortex 1)} + (c-1) \cdot \text{cov(cortex 1, cortex } q)}{c \cdot \text{var(cortex 1)}}} \tag{56}$$

$$\text{slope(cortex 1, FSI)} = \frac{1}{cw} \tag{57}$$

where:
- $c$ is the number of cortex neurons connected to each FSI
- $w$ is the weight of each cortex→FSI connection

Thus, changes in strength of connection may be more likely to affect slope, while changes in the quantity of connections, the covariance between cortical neurons, and the variance of cortical neurons may also affect the cortex to FSI correlation. In stress, correlation is larger while slope is reduced, suggesting reduction in quantity of cortex to FSI connections.

For code, see https://github.com/dirkbeck/DM_space_model/blob/main/disorder_hypotheses/ctx_to_FSI_synchrony_analysis.m.

**Simulation illustrating the theoretical change.** We then conducted a simulation to examine the case where cortical data and cortex to FSI connectivities were less uniform. To do this, we constructed a network with random unit normal connection weights from cortex to FSI. Then we randomly lesioned all but several of the connections to reflect the sparsity of the brain. The connection from the first cortical neuron to the FSI was always preserved.

In each of ten simulations, a cortical input to sSPN was generated as a random unit normal $100 \times 1$ vector. Activity of the first cortical neuron and the FSI was recorded. These are plotted in Supplementary Fig. 4m, n along with a linear regression fit.

For code, see https://github.com/dirkbeck/DM_space_model/blob/main/disorder_hypotheses/ctx_to_FSI_synchrony_analysis.m.

**Reporting summary**
Further information on research design is available in the Nature Portfolio Reporting Summary linked to this article.

## Data availability

• Source data are provided with this paper. • This paper primarily analyzes existing, publicly available data in the **Corticostriosomal Circuit Stress Experiment** database[69] (https://data.mendeley.com/datasets/z9jd8xhj84/1). • This paper additionally analyzes data published alongside the current paper in the **Computational model of striosomal circuit in normal and disordered decision-making** database[70] (https://doi.org/10.7910/DVN/SMKW0I). • All original code has been deposited in the DM_space_model repository[71] https://zenodo.org/records/15433189 and is publicly available as of the date of publication. • Any additional information required to reanalyze the data reported in this paper is available from the lead contact upon request. • See **Instructions to run code, Supplementary Information** for instructions to run the code that analyzes the data. Source data are provided with this paper.

## Code availability

• All original code has been deposited in the DM_space_model repository[71] (https://zenodo.org/records/15433189). • See the **Instructions to run code, Supplementary Information** for instructions to run the code.

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

## Acknowledgements

We greatly appreciate G. Schoenbaum, and Y. Shaham for their constructive criticism and discussions during model development. This project was supported by the NSF/CAREER (#2235858), NIH/NIDA (#RO1DA058653), U-RISE T34GM145529, G-RISE T32GM144919, 1R25GM132959-05.

## Author contributions

conceptualization: A.F., K.A.G.; data curation: D.W.B., L.D.D., L.I.R., S.M.D., A.G., Q.Z.; formal analysis: D.W.B., L.I.R., L.D.D., C.N.H., A.G., Q.Z., D.T., M.P.; funding acquisition: A.F., K.A.G.; investigation: A.F., D.W.B., L.I.R., L.D.D., C.N.H., Q.Z., S.B.H., A.Y.M., A.G., K.N., K.A.G.; methodology: A.F., D.W.B., S.M.D., L.I.R., A.A.S., N.F.R., R.J.I.; project administration: A.F., K.A.G.; software: A.F., D.W.B., L.I.R., L.D.D., C.N.H., Q.Z., A.G.; supervision: A.F. K.A.G.; validation: A.F., D.W.B., L.D.D., A.G., Q.Z., S.M.D., L.I.R.; visualization: D.W.B., S.B.H., P.V., S.A.B., A.Y.M., C.N.H., R.J.I., A.A.S., N.F.R.; writing – original draft: A.F., D.W.B., C.N.H., S.U.B., K.A.G., L.E.O., G.L.W.; writing – review & editing: A.F., D.W.B., C.N.H., K.A.G., T.M.M., M.P. These authors contributed equally: L.D.D., L.I.R., S.M.D., and D.T.

## Competing interests

The authors declare no competing interests.
