## [Transparent Peer Review file · Nature Communications]

A decision-space model explains context-specific decision-making

Corresponding Author: Dr Alexander Friedman

Version 0:

Reviewer comments:

Reviewer #1

(Remarks to the Author)

The authors present a computational model of distinct subpopulations involved in decision-making, with a particular emphasis on the striatum. They differentiate between striosomes and matrix neurons and examine their interactions with dopaminergic neurons (daSNC). Unlike existing models, this work introduces the novel aspect of separating the striatal subpopulations, a distinction shown to play a critical role in various neuropsychiatric disorders. The authors demonstrate that patterns of SPN activity, modeled as a decision-space, can predict context-dependent decision-making, highlighting the distinct roles of striosomes and matrix neurons.

The article is both original and novel, introducing an innovative separation of the striatal population into striosomes and matrix neurons. The authors provide a detailed study supported by a robust methodology, and their results effectively reproduce experimental data. However, some improvements can be performed in the manuscript to help the flow of reading and comprehension.

To enhance the clarity and accessibility of the manuscript, I propose the following changes, most of which are minor:

Comments on the Main section:

(1) In the main section, the authors emphasize that many modeling studies exploring the competition between direct and indirect pathways omit the division of the striatum into striosome and matrix subpopulations (line 57). Specifically, the authors reference [25], a review paper on computational models of the basal ganglia from 2013. However, more recent models have been developed that also consider the cortico-basal ganglia-thalamic loop, incorporating plasticity rules based on dopamine delivery at corticostriatal synapses (treating SPNs as a single unit).

In line 58, the authors state: "The omission of striosomes from models of decision-making or basal ganglia function hinders the interpretation of important features of the striatum because it prevents an accurate depiction of striatal-daSNC interplay, which is primarily striosomal." Could the authors elaborate on the specific features missed in recent studies describing decision-making and reinforcement learning that include dopaminergic plasticity rules? Additionally, what are the benefits of considering sSPN and mSPN populations separately in such models? While the separation of striosomes and matrix neurons is emphasized in the results section, explicitly stating the implications and broader impact of this distinction in the introduction would strengthen the narrative.

Beyond [25], the authors should also reference more recent studies that address related topics. For example, recent articles by Bogacz, Humphries, Verstynen, and others could provide relevant context. Some of these studies are already cited in the discussion (references 58-60) and could be included here to strengthen the introduction and provide a broader perspective.

(2) The authors introduce the concept of "decision-dimension" for the first time in the main section, referring to Table 1. On an initial read, the explanation of this concept in Table 1 is somewhat unclear. However, I later realized that there is a detailed description of this concept in the Methods section. To improve clarity, the authors should include a cross-reference in Table 1 pointing to the relevant section in the Methods, allowing readers to obtain a more thorough understanding of the concept.

(3) Regarding the terminology used in Table 1, sSPN and mSPN are often referred to as striosome and matrix components, respectively. To avoid potential confusion, the authors could consider including this second notation (striosome and matrix) in Table 1 or, alternatively, consistently using sSPN and mSPN throughout the manuscript. This would ensure greater clarity and consistency for the readers.

Comments on the Results section:

- (4) In line 87, the reader is referred to Supplementary Note 1, where the authors discuss “Instance 1.” However, the term “Instance 1” is not defined or linked to any earlier explanation, leaving the reader uncertain about its meaning. Adding a reference to where “Instance 1” is first introduced or defined would resolve this ambiguity.
- (5) The model is described in multiple sections (Results section, Methods section, and Supplementary Notes). To enhance the flow and provide readers with a comprehensive overview, the authors should include a reference to the Methods section at the beginning of the Results section, where the model is first mentioned. This cross-reference would prevent confusion, such as the issue with “Instance 1” mentioned above.
- (6) A graphical representation of the network would be helpful for illustrating the connectivity and providing context. At the beginning of the Results section and within the Methods section, it would be beneficial to include a reference to the network sketch. I have seen a representation of it in Figure 1e,f, which I am not sure is the complete network implemented (see comments below). A clear visualization of all the populations and connections considered in the model would help readers to follow the description more easily.
- (7) In line 332, it is unclear if the GPe population is included in the network. If it is part of the model, it should be represented in Figure 1e (as it is shown in Figure 4a) or added to a new network-sketch figure for completeness (see comment above). This will ensure all populations considered in the model are visually accounted for.
- (8) In Figures 1e-g and Extended Data Figure 1 (referenced in lines 167-181), the authors illustrate how sSPN activity encodes the decision-space. Did the authors attempt a similar analysis for mSPN activity? If so, it would be valuable to include this information to highlight any differences or similarities between sSPN and mSPN activity in encoding decision-space. If not, such an analysis could strengthen the study by providing additional insights into the roles of these populations.
- (9) Regarding what is discussed in lines 331-350, a more complex pathway structure is involved in avoiding actions, beyond the usual direct/indirect competition, where the Arkypallidal neurons at GPe plays a critical role (e.g., as described in Mallet et al.'s Pause-and-Cancel model or the review by Giossi et al., 2024). How might these unconsidered pathways influence the model, particularly regarding the imSPN population, given its lack of connection to GPe neurons? Adding a discussion on the potential impact of these pathways could strengthen the interpretation and generalizability of the model.
- (10) Lines 361-376: Reward prediction errors (RPE) have been extensively used to modulate dopaminergic effects on corticostriatal connections, often by altering synaptic strengths (e.g., Humphries et al., Bogacz et al., Vich et al.). The authors indicate that their model adopts a different approach to this modulation. Could the authors elaborate on why their interpretation differs from these prior studies? These studies consider also the concept of an eligibility trace, which evaluates how “eligible” a neural population is to receive a reward. Models incorporating eligibility traces often demonstrate the ability to learn new actions when rewards are switched. Could the authors provide a comparison between their model and these models? This would help clarify the unique advantages and contributions of the proposed approach.
- (11) Principal Components of Cortical Activity: Could the authors provide a description of the means associated with the four principal components of cortical activity they consider? This could be included when discussing “Decision-dimensions and decision-space” in the Methods section.

Comments on the Discussion section:

- (12) Figure 5: Figure 5b is somewhat unclear. The upper legend describes the colors blue, red, yellow, green, and purple, while the figure legend states that there are three different isosurfaces per color (representing decision-space dimensionality), ranging from lightest to darkest. However, in Figure 5b, it is difficult to discern 15 distinct isosurfaces—I can only identify 9. Additionally, the colors may cause confusion; for instance, I see one purple, two “reds,” two “blues,” three “greens,” and one that appears to be a mix between green and yellow. An additional explanation in the legend or the text might help clarify the visualization, as I understand the challenge of effectively presenting such complex data. Additionally, have you considered applying a clustering technique to Figure 5a? This could help quantify and better represent the information associated with Figure 5b in the caption.

Other minor comments:

- (1) Line 85-87: The punctuation marks in sentence “Our model describes how physiological interactions between elements of a striosome-centered circuit inform decision-making. (For extended reasoning behind our choice of circuit elements, see Supplementary Note 1.) Striatum-projecting...” feels slightly awkward. I suggest moving the dot written before the parenthesis after the closing parenthesis and removing the dot after “Note 1.” That is, “Our model describes how physiological interactions between elements of a striosome-centered circuit inform decision-making (for extended reasoning behind our choice of circuit elements, see Supplementary Note 1). Striatum-projecting...”
- (2) Line 103: When describing the three ways signals are passed from sPNSs to daSNC, it would be clearer to use brackets to introduce the notation, such as “(dsSPN→daSNC), (isSPN→GPe→daSNC), ...”. The sentence structure could be improved by separating the list with commas and semicolons instead of using multiple dots.

- (3) Methods Section: Please review the description of the subindexes. The use of indices i , j , and k is sometimes unclear, and there may be some confusion in their explanation.
- (4) Line 146: Consider adding the reference to the “Merton process model” here. This model closely resembles the widely used Drift-Diffusion model (Ratcliff, 2008) in decision-making. Is there a significant difference between them?
- (5) Figure 1d: Consider adding more tick marks on the y-axis for better visualization.
- (6) Line 192: The phrase “or (higher-)” should be corrected to “(or higher-)”.
- (7) Line 196-197: Could you provide a more specific definition of “simple” and “difficult” tasks? It might be helpful to also consider terms like conflict and volatility, in addition to the number of possible choices (as mentioned in line 222). Does the model account for factors such as motivation or urgency?
- (8) Line 225: The “Granger causality-based tool” would benefit from a citation. Could you please provide a reference for this?
- (9) Figure 2e: Could you clarify the notation “(sSPNs = XX, mSPNs = YY)” in Figure 2e? Also, what are the units on the x-axis?
- (10) Extended Data Fig. 3: It would improve clarity if the bins in the histogram were ordered consistently with the other panels (Control CBC, Stress CBC, Control BB, Stress CC).
- (11) Line 433: It would be helpful to include a description of the PTSD nomenclature here.
- (12) Line 444: The citation should refer to “Extended Data Fig. 6” instead of “Extended Fig. 6.”
- (13) Line 452: Could you clarify what is meant by “a traditional model that learns only the discounted sum of future rewards”? The cited models typically learn from past rewards.

Overall, the manuscript presents a novel and interesting approach to decision-making. The methodology is solid, and the results provide valuable insights. However, I believe the suggestions above could enhance clarity, improve the flow, and help strengthen the manuscript.

(Remarks on code availability)

I have not carefully reviewed/executed the code provided by the authors. However, I have reviewed the different folders it contains and confirmed that all the code and data files are publicly visible. It is worth noting that the README file contains very limited information. A description of the different folders, code, etc., would be appreciated to help locate the information and be able to execute the code.

Reviewer #2

(Remarks to the Author)

The authors conceived a computational model concerning decision-making and action selection based on the known anatomy of the brain circuitry, especially the striosome-matrix compartments. In the striatum, both striosome-matrix structure and direct-indirect pathways are two major anatomical compartments but the clear functional and computational roles remain elusive. The authors attempt to propose an integrative model. Although the results look quite interesting, the reviewer has some fundamental concerns.

Most importantly, it is very difficult to evaluate the internal mechanisms of the model based on the provided results. Fig 1. is related to the model itself, and Figs 2-6 are related to specific situations such as some experiments, stress conditions, aging, etc. So Fig. 1 should do the job of explaining the model. But Fig. 1 a-e, g. are either schematic diagrams or illustrations, and Fig. 1f is an introduction to the equation. For sure, the equation provides a detailed mechanism. But what is critical is how the information flows in terms of the modeled neural activity. Some are shown in Ext. Fig. 1a, but this is too limited. A connection diagram and example time courses of units for each anatomical sub-unit would be helpful for the readers to understand how the model works.

The above comment is about within-trial dynamics. It is also needed to elaborate learning dynamics of the decision-making model. For example, when animals make a choice in 2AFC task and when the reward is given, how the model updates behavior based on the given reward. The authors provide mathematical equations, but it would be very difficult for the readers to understand how the model changes decisions over time. Direct comparison of trial-by-trial choice with a simple reinforcement learning model would be very helpful.

Finally, the model is complicated and there are many parameters. How the parameters are chosen needs to be explained further. It would be great if authors could provide any basis for how they chose such parameters shown in Common parameters, whether and how much such parameters are biologically plausible.

Without a more detailed and straightforward elaboration of the mechanisms of the model, the reviewer is concerned that the

demonstration explaining several normal and abnormal decision-making processes would have quite limited implications and inspiration to the readers.

Below is a minor point:

line 51: (isSPNs and imSPNs)

The reviewer tried to uncompress the .zip file from Github but an error occurred while uncompressing the .zip file. The Git clone was also not completely successful (errors related to long file path).

(Remarks on code availability)

I downloaded the DM_space_model-main.zip file from Github but had an error while unzipping it. I tried to get the source but the git generated errors - Filename too long.

Version 1:

Reviewer comments:

Reviewer #1

(Remarks to the Author)

I would like to thank the authors for their careful and thoughtful revisions. They have thoroughly and satisfactorily addressed all the comments and suggestions raised. In my opinion, the modifications introduced in the manuscript, along with the clarifications provided, have significantly improved both its clarity and scientific rigor. I therefore have no further comments or objections, and consider the manuscript suitable for publication in its current form.

(Remarks on code availability)

I have not personally executed the provided code, but I have reviewed the contents available in the repository. The folder includes the necessary scripts and usage instructions, which have been improved in both clarity and completeness from the previous version. The documentation now offers sufficient information to understand how to run the code and reproduce the main results, making it more accessible to readers.

Reviewer #2

(Remarks to the Author)

Authors have made a significant improvement on the manuscript, and carefully addressed most of the reviewer's points.

One minor comment: Fig. 1 title can be changed to " A striosome-centered circuit model regulating dopamine release to the striatum.". The current title sounds like that Fig 1. is an established fact.

(Remarks on code availability)

I could run some codes and confirmed that those codes generate figures on the manuscript. but since there are too many files, reviewing codes line-by-line was not feasible.

Reviewer Comments:

Reviewer 1

The authors present a computational model of distinct subpopulations involved in decision-making, with a particular emphasis on the striatum. They differentiate between striosomes and matrix neurons and examine their interactions with dopaminergic neurons (daSNC). Unlike existing models, this work introduces the novel aspect of separating the striatal subpopulations, a distinction shown to play a critical role in various neuropsychiatric disorders. The authors demonstrate that patterns of SPN activity, modeled as a decision-space, can predict context-dependent decision-making, highlighting the distinct roles of striosomes and matrix neurons. The article is both original and novel, introducing an innovative separation of the striatal population into striosomes and matrix neurons. The authors provide a detailed study supported by a robust methodology, and their results effectively reproduce experimental data. However, some improvements can be performed in the manuscript to help the flow of reading and comprehension. To enhance the clarity and accessibility of the manuscript, I propose the following changes, most of which are minor:

Comments on the Main section:

1. In the main section, the authors emphasize that many modeling studies exploring the competition between direct and indirect pathways omit the division of the striatum into striosome and matrix subpopulations (line 57). Specifically, the authors reference [25], a review paper on computational models of the basal ganglia from 2013. However, more recent models have been developed that also consider the cortico-basal ganglia-thalamic loop, incorporating plasticity rules based on dopamine delivery at corticostriatal synapses (treating SPNs as a single unit). In line 58, the authors state: “The omission of striosomes from models of decision-making or basal ganglia function hinders the interpretation of important features of the striatum because it prevents an accurate depiction of striatal-daSNC interplay, which is primarily striosomal.” Could the authors elaborate on the specific features missed in recent studies describing decision-making and reinforcement learning that include dopaminergic plasticity rules? Additionally, what are the benefits of considering sSPN and mSPN populations separately in such models? While the separation of striosomes and matrix neurons is emphasized in the results section, explicitly stating the implications and broader impact of this distinction in the introduction would strengthen the narrative. Beyond [25], the authors should also reference more recent studies that address related topics. For example, recent articles by Bogacz, Humphries, Verstynen, and others could provide relevant context. Some of these studies are already cited in the discussion (references 58-60) and could be included here to strengthen the introduction and provide a broader perspective.

We thank the reviewer for these questions. To link our work to more recent research on the cortico-basal ganglia-thalamic loop, we added a review of these studies to our Introduction, lines 60-80, and supplementary note 8 (lines 2092-2112). In the revised manuscript, we discuss modern cortico-basal ganglia-thalamic models that explore different aspects of decision making, action selection, and learning. We also now discuss how these models explain cortico-basal ganglia-thalamic circuit physiology, where many of the circuit’s subsequent regions have fewer neurons, and how this anatomical architecture may be related to function. For example, some research has suggested that basal ganglia structures perform dimensionality reduction, while more recent work considers the idea that basal ganglia structures work as “weights”/bias for output signals used downstream (Humphries, 2024).

To emphasize the important link between the cortico-basal ganglia-thalamic loop and dopaminergic plasticity, we now discuss in the Introduction modeling work that explores Cortico-basal ganglia-thalamic loops and dopamine-related plasticity (lines 60-80). One such model suggests that control ensembles within the cortico-basal ganglia-thalamic network/loop are mediated by dopaminergic plasticity to adjust ensemble use for balancing accuracy and speed when making decisions (Bahuguna et al., 2024). Another idea that has been explored through modeling is using dopaminergic plasticity between cortical and striatal synapses as a mechanism for shifting decision making based on available feedback/information. This circuit's function is also examined through the function of direct and indirect pathway balances/activity; more recent work has posited that whether an action is refrained from or actively stopped is a result of the activity between pathways and the stage of execution the action is before the decision to cancel the action was made (Dunovan et al., 2015).

Relatedly, we feel that the reviewer has a very interesting comment about the importance of the different dopamine plasticity rules in matrix and striosome neurons in the context of reinforcement learning. We have added a **Supplementary Note 9** (lines 2114-2129) documenting our thoughts. Because of the markedly different short-term plasticity rules in matrix versus striosome neurons, we wonder if there are different long-term plasticity rules, also. If so, this would be very relevant to the reinforcement learning models of the circuit where dopaminergic neurons play one role (a "critic") and matrix neurons another (an "actor"). Because striosomes project to dopaminergic neurons, perhaps they receive critic-related updates during learning, and matrix separately receives actor-related updates. Future computational work could explore how these updates might take place during learning.

To more explicitly elaborate on the benefits of considering sSPN and mSPN populations separately in models of the basal ganglia, we now state much more explicitly in lines 54-58 of the Introduction and in a new **Supplementary Note 1** (lines 1905-1942) the reasons why modeling the striosomal and matrix compartments separately is so essential. The importance is threefold. First, dissociable functional roles have been found experimentally for striosomes (resolving conflict, mediating dopamine, habit formation, value encoding for adapting to changing tasks, and action-outcome learning) and matrix (subjective valuation, action selection/initiation, and motor functions). So, the compartments have distinct functional roles. Second, they have important differences in connectivity. These compartments have distinct anatomical connections; the striosomes receive preferential input from the prelimbic, infralimbic, posterior orbitofrontal, and insular cortices while the matrix are preferentially innervated by the primary motor cortex, with both compartments receive projections from other cortical regions. Third, pathway dopamine receptor activity has opposite effects on striosomal and matrix neurons; in striosomes, direct pathway dopamine reduces upstates while in matrix activation increases/lengthens upstates (Prager et al., 2020). Given all these factors, computational models that account for striosomal and matrix striatal subdivisions are critical to understanding decision-making and learning processes.

2. The authors introduce the concept of "decision-dimension" for the first time in the main section, referring to Table 1. On an initial read, the explanation of this concept in Table 1 is somewhat unclear. However, I later realized that there is a detailed description of this concept in the Methods section. To improve clarity, the authors should include a cross-reference in Table 1 pointing to the relevant section in the Methods, allowing readers to obtain a more thorough understanding of the concept.

We thank the Reviewer for this suggestion. We added a cross-reference from Table 1 (line 1528) to the "Decision-dimensions and decision-space" section of the Methods (starts line 2278).

3. Regarding the terminology used in Table 1, sSPN and mSPN are often referred to as striosome and matrix components, respectively. To avoid potential confusion, the authors could consider including this second notation (striosome and matrix) in Table 1 or, alternatively, consistently using sSPN and mSPN throughout the manuscript. This would ensure greater clarity and consistency for the readers.

We thank the Reviewer for this suggestion. For improved clarity, we changed references to “sSPN” in Table 1 to “striosomal striatal projection neuron” and references to “mSPN” to “matrix striatal projection neuron” (line 1528).

Comments on the Results section:

4. In line 87, the reader is referred to Supplementary Note 1, where the authors discuss “Instance 1.” However, the term “Instance 1” is not defined or linked to any earlier explanation, leaving the reader uncertain about its meaning. Adding a reference to where “Instance 1” is first introduced or defined would resolve this ambiguity.

We apologize for this omission and thank the Reviewer for this suggestion. To clarify our use of instances of our model, we added **Fig. 1c** (legend in lines 691-700), which summarizes our use of Instances 1, 2, and 3 of our model, the reason we use each, and the related figures. Additionally, we added an **Extended Data Fig. 1** (legend in lines 940-966) with cartoon representations of the analyses related to each instance.

5. The model is described in multiple sections (Results section, Methods section, and Supplementary Notes). To enhance the flow and provide readers with a comprehensive overview, the authors should include a reference to the Methods section at the beginning of the Results section, where the model is first mentioned. This cross-reference would prevent confusion, such as the issue with “Instance 1” mentioned above.

We thank the Reviewer for this suggestion. We have added a cross-reference to the **Analyzed Instances of the Model** section in Methods from the first sentence of Results (lines 104-105). We feel that this addition, in conjunction with the new **Fig. 1c** and **Extended Data Fig. 1** (see response to comment 4), will clarify our modeling techniques from the beginning of Results.

6. A graphical representation of the network would be helpful for illustrating the connectivity and providing context. At the beginning of the Results section and within the Methods section, it would be beneficial to include a reference to the network sketch. I have seen a representation of it in Figure 1e,f, which I am not sure is the complete network implemented (see comments below). A clear visualization of all the populations and connections considered in the model would help readers to follow the description more easily.

We thank the Reviewer for this suggestion. We included a new **Fig. 1b** (lines 683-689), which depicts a complete representation of the circuit. We now reference this circuit diagram from the beginning of Results and from the related Methods section.

7. In line 332, it is unclear if the GPe population is included in the network. If it is part of the model, it should be represented in Figure 1e (as it is shown in Figure 4a) or added to a new network-sketch figure for completeness (see comment above). This will ensure all populations considered in the model are visually accounted for.

We thank the Reviewer for this suggestion. In lieu of the previous **Fig. 1e**, which showed a partial representation of the circuit, we have added a new **Fig. 1b** as discussed in Comment 6. With this change, we have now visually accounted for all circuit elements. To a similar end, we added a

new z_{GPe} variable to eq. 2 (lines 137-139) to show how GPe computationally biases the formation of the indirect pathway decision-space. Additionally, we created a new **Fig. 2f** (legend in line 729) to demonstrate the functional role of GPe. This way, the connectivity, computational, and functional roles of GPe are clearly stated.

Finally, we added **Supplementary Note 6: Effect of GPi, LHb, and RMTg, and GPe on the direct and indirect pathway spaces** (lines 2040-2062) to make clear the assumptions that underlie our model of GPe (and GPi, LHb, and RMTg). There is only one study on the connectivity of striosomes through GPe, so our model relies on a hypothesis about the details of connectivity. We also discuss how different connectivity assumptions would change our results. For instance, if only a subset of striosomes project to GPi, then there could be nonlinear effects of striosome activity on the decision-space, but decision-spaces would still form in the direct and indirect pathways. That is, there would still be a direct pathway decision-space and an indirect pathway decision-space.

8. In Figures 1e-g and Extended Data Figure 1 (referenced in lines 167-181), the authors illustrate how sSPN activity encodes the decision-space. Did the authors attempt a similar analysis for mSPN activity? If so, it would be valuable to include this information to highlight any differences or similarities between sSPN and mSPN activity in encoding decision-space. If not, such an analysis could strengthen the study by providing additional insights into the roles of these populations.

We thank the Reviewer for this suggestion. We added a new **Fig. 2c** (legend in line 717) that shows how changes in sSPN and mSPN activity affect the formation of the decision-space. Additionally, we created a new **Fig. 2d** (legend in lines 719,720) that shows the role of each in affecting action values. Through these analyses, we show that sSPNs, and not mSPNs, affect the decision-space. We also show that while mSPNs, but not sSPNs, have a direct effect on action-values, sSPNs also affect action values via their influence on the decision-space.

9. Regarding what is discussed in lines 331-350, a more complex pathway structure is involved in avoiding actions, beyond the usual direct/indirect competition, where the Arkypallidal neurons at GPe plays a critical role (e.g., as described in Mallet et al.'s Pause-and-Cancel model or the review by Giossi et al., 2024). How might these unconsidered pathways influence the model, particularly regarding the imSPN population, given its lack of connection to GPe neurons? Adding a discussion on the potential impact of these pathways could strengthen the interpretation and generalizability of the model.

We thank the reviewer for this suggestion. To discuss how these unconsidered pathways influence the model, we have added a new passage to **Discussion** in lines 487-489 and a new **Supplementary Note 7: Implications of GPe pathways** (lines 2064-2090). Recent research has indicated that the Arkypallidal neurons of the GPe project the most strongly to the striatal neurons associated with movement and signal to both direct and indirect pathway neurons, but with stronger signaling to indirect pathway neurons. So, the Arkypallidal neurons have an important effect on the construction of the indirect pathway decision-space. How exactly this might affect the indirect pathway decision-space will be elucidated by future experimental work. We explore possible future modifications to the model in the Supplementary Note. For our model, the key experimental evidence will be determining whether the arkypallidal neurons preferentially project to the striosomes or matrix. If future work shows arkypallidal-striosome connections, our model suggests they regulate the indirect-pathway decision-space, akin to striosome-daSNC interactions we explore in **Fig. 5** but indirect-pathway specific. If they target matrix or both, they might act as switches that abruptly reduce the decision-space dimension when new information demands adaptation. Or arkypallidal neurons might affect action values directly. We expect to fine-tune our model based on future experimental work to gain more depth on the role of GPe in affecting the indirect pathway decision-space.

10. Lines 361-376: Reward prediction errors (RPE) have been extensively used to modulate dopaminergic effects on corticostriatal connections, often by altering synaptic strengths (e.g., Humphries et al., Bogacz et al., Vich et al.). The authors indicate that their model adopts a different approach to this modulation. Could the authors elaborate on why their interpretation differs from these prior studies? These studies considers also the concept of an eligibility trace, which evaluates how "eligible" a neural population is to receive a reward. Models incorporating eligibility traces often demonstrate the ability to learn new actions when rewards are switched. Could the authors provide a comparison between their model and these models? This would help clarify the unique advantages and contributions of the proposed approach.

We thank the Reviewer for this suggestion. To compare our model with a classical model where dopamine encodes reward prediction error (RPE), we included a new **Extended Data Fig. 7j** (legend in lines 1354-1357) that contrasts how dopamine fits into the two models. Dopamine plays an important role in reinforcement learning (RL) in both interpretations. In the classical view, dopamine is crucial for updating values the agent assigns to states via eligibility traces. In our model, dopamine dictates how the agent interprets the state ("state" in RL is analogous to "cortical input" in terms of our model). While these functional roles are different, interestingly, the models make similar predictions about the episodes at which dopamine is expected to spike during a learning task. We show this through a new **Extended Data Figs. 7k,l**. (legend in lines 1357-1377). In our new analysis, we conduct a simple RL experiment. In the middle of the experiment, the rewards are suddenly switched. We show that during this experiment, a classical model and our decision-space model predict dopamine spikes at similar episodes. Thus, our model offers a new explanation for a commonly observed experimental result. The interpretation of the result, however, is different. In particular, the decision-space model makes the testable prediction that the animal has changed its decision-space in response to the unexpected information. A topic of future research will be determining the relationship between reward prediction errors and shifts in the decision-space over learning (in contrast to during a decision, which our model makes a strong claim about). To emphasize this need for future research, we have added a passage to the end of **Discussion** covering future directions (lines 489-491).

11. Principal Components of Cortical Activity: Could the authors provide a description of the means associated with the four principal components of cortical activity they consider? This could be included when discussing "Decision-dimensions and decision-space" in the Methods section.

We thank the Reviewer for this suggestion. We introduce several possible decision-dimensions as examples throughout the text (specifically, a reward-predominant dimension, a cost-predominant dimension, a novelty-predominant dimension, and a location-predominant dimension). We also suggest other more abstract information that could be mapped to decision-dimensions, such as informational information, such as social dominance, the desire to replicate, exploration, or morality. To better explain our reasoning behind the choice of these examples, we included a passage in "**Decision-dimensions and decision-space**" in the Methods section that explains our choices (lines 2296-2313).

The new passage explains that we largely form our example decision-dimensions based on the literature that demonstrates neurons in the circuit responding to these environmental inputs (reward, cost, novelty). However, we emphasize that any of these decision-dimensions might correspond to a wide range of information. For example, a given individual might map reward level and shades of the color green onto the same dimension.

This new passage compliments our **Extended Data Fig. 5** (legend in lines 1236-1292), which suggests how the brain might computationally derive information along decision-dimensions from a many-dimensional cortical signal. In this figure, we construct a network using Instance 2 of our model, where the connectivity between the cortex and SPNs is sparse. Thus, each SPN only has access to several cortical neurons. In our simulations shown in **Extended Data Fig. 5b-e**, the

SPNs learn the principal components of the activity over time of only the small cortical subset to which they connect. These cortical-SPN weights could be learned, for instance, by a modified Oja's rule. We found that the population of SPNs nonetheless encode information about decision-dimensions in their distributions of firing rates. Thus, through learning principles similar to those found in the brain, a large network of cortical neurons and SPNs could feasibly encode principal components.

Comments on the Discussion section:

12. Figure 5: Figure 5b is somewhat unclear. The upper legend describes the colors blue, red, yellow, green, and purple, while the figure legend states that there are three different isosurfaces per color (representing decision-space dimensionality), ranging from lightest to darkest. However, in Figure 5b, it is difficult to discern 15 distinct isosurfaces—I can only identify 9. Additionally, the colors may cause confusion; for instance, I see one purple, two “reds,” two “blues,” three “greens,” and one that appears to be a mix between green and yellow. An additional explanation in the legend or the text might help clarify the visualization, as I understand the challenge of effectively presenting such complex data. Additionally, have you considered applying a clustering technique to Figure 5a? This could help quantify and better represent the information associated with Figure 5b in the caption. We thank the Reviewer for these comments. To present the concept in the previous **Figs. 5a,b** more clearly, we designed a new **Fig. 6a** (lines 883-885); note that **Fig. 5** of the previous manuscript is now **Fig. 6** in the revised manuscript). Previously, **Fig. 5a** was a scatterplot where points were sampled from a probability distribution, and **Fig. 5b** showed the probability distribution. For clarity, we now include a single improved visual of the probability distribution, separating the various decision-space dimensions (0-dimensional, 1D, 2D, 3D, 4D) onto their own subplots. Using this new representation, we show how different decision-spaces form more likely at different activities of sSPN, daSNC, and LHb, although any decision-space can form at any set of activities with small probability.

Other minor comments:

13. Line 85-87: The punctuation marks in sentence “Our model describes how physiological interactions between elements of a striosome-centered circuit inform decision-making. (For extended reasoning behind our choice of circuit elements, see Supplementary Note 1.) Striatum-projecting...” feels slightly awkward. I suggest moving the dot written before the parenthesis after the closing parenthesis and removing the dot after “Note 1.” That is, “Our model describes how physiological interactions between elements of a striosome-centered circuit inform decision-making (for extended reasoning behind our choice of circuit elements, see Supplementary Note 1). Striatum-projecting...” We thank the Reviewer for this suggestion. We made the change as suggested in lines 102-105 to improve readability.

14. Line 103: When describing the three ways signals are passed from sPNSs to daSNC, it would be clearer to use brackets to introduce the notation, such as “(dsSPN→daSNC), (isSPN→GPe→daSNC), ...”. The sentence structure could be improved by separating the list with commas and semicolons instead of using multiple dots. We thank the Reviewer for this suggestion. To improve readability, we used the (dsSPN→daSNC) notation when introducing the connections between circuit elements in lines 126-131.

15. Methods Section: Please review the description of the subindexes. The use of indices i, j, and k is sometimes unclear, and there may be some confusion in their explanation. We thank the Reviewer for this point. We expect that the lack of consistency in indexing was confusing. The indices “i” and “j” were used to refer to a certain decision-dimension and a certain action, respectively, in most cases, but there were several exceptions. To be consistent in our

notation, we changed the exceptions to other indices in lines 2908-2910, 2916, 3819, 3120, and 3826-3828. Additionally, we added a short paragraph to the beginning of Methods explaining this convention in lines 2314-2317.

16. Line 146: Consider adding the reference to the “Merton process model” here. This model closely resembles the widely used Drift-Diffusion model (Ratcliff, 2008) in decision-making. Is there a significant difference between them?

We thank the Reviewer for this suggestion. We added a reference to the Merton process model to the main text in line 170 (it was previously cited only from Methods). Additionally, we added a sentence in lines 171-173 explaining that our Merton process model is similar to the classical drift-diffusion model but allows for more than three potential actions.

17. Figure 1d: Consider adding more tick marks on the y-axis for better visualization.

We thank the Reviewer for this suggestion. In our new version of the manuscript, we moved an updated version of this analysis to **Fig. 2**, where it is now one of four plots that compares the functional role of circuit elements in biasing the decision-space or affecting action values (see Comment 7). To improve the comparison, we relabeled the y-axis from “average decision-dimensions used” to “probability decision-dimension is used.” The former is a multiple of the latter, so the plotted lines remain unchanged, but the axis limits have changed from [0, 4] to [0, 1]. This change will make the plot consistent with others in the new version of the paper (i.e. **Fig. 2f**, **Fig. 4f**) and, we hope, improve interpretation of the analysis. We want to emphasize in this plot the slopes of the lines rather than the precise activity levels that produce 1, 2, 3, or 4-dimensional decision-space, on average (which keeping the old axis title and adding tick marks might have emphasized).

18. Line 192: The phrase “or (higher-)” should be corrected to “(or higher-).”

We thank the Reviewer for this correction. We made the change as suggested on line 221.

19. Line 196-197: Could you provide a more specific definition of “simple” and “difficult” tasks? It might be helpful to also consider terms like conflict and volatility, in addition to the number of possible choices (as mentioned in line 222). Does the model account for factors such as motivation or urgency?

We thank the Reviewer for these questions. To provide a more detailed summary of scenarios where we could expect a high- versus low-dimensional decision-space, we added a **Supplementary Note 5: Situations in which different types of decision-spaces form** (lines 2014-2038). The note summarizes the scenarios discussed in the paper (conflict and number of options in **Fig. 3**, disorders versus control in **Fig. 4**, motivation in **Fig. 5e**). While we do not offer a single metric for determining simple versus difficult tasks, we hypothesize other factors that may influence the dimension of the decision-space (urgency, phase of learning). We now reference this note from line 252, which was line 222 in the previous version.

20. Line 225: The “Granger causality-based tool” would benefit from a citation. Could you please provide a reference for this?

We thank the Reviewer for this suggestion. We added a citation and edited the wording for clarity in lines 253-256. In our analysis, we use classical Granger causality tests (using a MATLAB package) to measure how neurons interact across a task. Then we compute statistical significance by using the same techniques on shuffled data. We also cross-reference the **Connected SPNs through Granger causality** section in Methods from line 256.

21. Figure 2e: Could you clarify the notation “(sSPNs = XX, mSPNs = YY)” in Figure 2e? Also, what are the units on the x-axis?

We thank the Reviewer for these questions. The (sSPNs = XX) verbiage was meant to indicate the count of neurons. We’ve clarified this by instead writing “XX sSPNs” in all cases, in **Fig. 2e**

and through the text where we used similar notation (lines 772, 774, 1111-1114, 1127, 1128, 1202-1204, 3715-3718, 3739-3741).

The x-axis (skewness of the deliberation time distribution) on **Fig. 3e** (former **Fig. 2e**) is dimensionless, which we've indicated for clarity. We also made a similar change to **Extended Data Fig. 3i**, where skewness is also plotted. In both cases (lines 775, 1096), we also added the units to the figure legend.

22. Extended Data Fig. 3: It would improve clarity if the bins in the histogram were ordered consistently with the other panels (Control CBC, Stress CBC, Control BB, Stress CC).

We thank the Reviewer for this suggestion. In **Fig. 4a**, the bars are ordered: Control BB, Control CBC, stress BB, stress CBC. So, in **Extended Data Fig. 4** in the new version of the manuscript, we moved the bars to be consistently in this order. (Note that **Extended Data Fig. 3** in the previous version is now **Extended Data Fig. 4**).

Additionally, there was similar inconsistency in the order of the tasks in the plots of control animals in **Extended Data Fig. 3** (previously **Extended Data Fig. 2**). So, we updated the order to be consistent across tasks.

23. Line 433: It would be helpful to include a description of the PTSD nomenclature here.

We thank the Reviewer for this suggestion. We added a description of the PTSD nomenclature in line 471.

24. Line 444: The citation should refer to “Extended Data Fig. 6” instead of “Extended Fig. 6.”

We thank the Reviewer for this correction. We made the change as suggested in line 468.

25. Line 452: Could you clarify what is meant by “a traditional model that learns only the discounted sum of future rewards”? The cited models typically learn from past rewards.

We thank the Reviewer for this question. We meant by this comment that algorithms such as Q-learning (Watkins 1989) learn an estimate of future rewards; however, the wording was confusing. Since, in our new submission, we analyze a RL framework directly in **Extended Data Fig. 7j-l** (legend in lines 1353-1377), we shorted this section of the Discussion to be clearer and more concise.

Overall, the manuscript presents a novel and interesting approach to decision-making. The methodology is solid, and the results provide valuable insights. However, I believe the suggestions above could enhance clarity, improve the flow, and help strengthen the manuscript.

26. I have not carefully reviewed/executed the code provided by the authors. However, I have reviewed the different folders it contains and confirmed that all the code and data files are publicly visible. It is worth noting that the README file contains very limited information. A description of the different folders, code, etc., would be appreciated to help locate the information and be able to execute the code.

We thank the Reviewer for this suggestion. We added a detailed description of the organization of the directory and the process by which to run the codes to the README file (previously this description was only in the manuscript). Additionally, we added an **Instructions to run code** section in Methods (lines 2150-2243) with descriptions of the code corresponding to each figure. This includes links to RUNME files where we include step-by-step instructions for running our analysis of experimental data.

Reviewer #2

The authors conceived a computational model concerning decision-making and action selection based on the known anatomy of the brain circuitry, especially the striosome-matrix compartments. In the striatum, both striosome-matrix structure and direct-indirect pathways are two major anatomical compartments but the clear functional and computational roles remain elusive. The authors attempt to propose an integrative model. Although the results look quite interesting, the reviewer has some fundamental concerns.

1. Most importantly, it is very difficult to evaluate the internal mechanisms of the model based on the provided results. Fig 1. is related to the model itself, and Figs 2-6 are related to specific situations such as some experiments, stress conditions, aging, etc. So Fig. 1 should do the job of explaining the model. But Fig. 1a-e, g. are either schematic diagrams or illustrations, and Fig. 1f is an introduction to the equation. For sure, the equation provides a detailed mechanism. But what is critical is how the information flows in terms of the modeled neural activity. Some are shown in Ext. Fig. 1a, but this is too limited. A connection diagram and example time courses of units for each anatomical sub-unit would be helpful for the readers to understand how the model works.

We thank the Reviewer for these suggestions. To 1) better explain the model and 2) describe how information flows in terms of the modeled neural activity, we added a new **Fig. 1b** (legend in lines 683-689) that focuses on the connectivity of the circuit. This new panel shows a detailed information flow in terms of the modeled neural activity. Additionally, to explain where (and why) we examine time courses, we created a new **Fig. 1c** (lines 691-700) and new **Extended Data Fig. 1** (lines 940-966) that describes in detail how we apply instances of the model to various analyses. **Fig. 1** and **Extended Data Fig. 1** now serve as a roadmap for the connectivity of the circuit, the bridge between physiology and higher-level concepts like the decision-space and action values, and the types of analyses we do across the other figures. This supplements our existing **Extended Data Fig. 2** (formerly **Extended Data Fig. 1**) where example time courses are shown.

Note that much of the content that was previously in **Fig. 1** has been moved to a new **Fig. 2**. We felt that the content corresponding to these suggestions deserves important placement in the paper, and splitting the figure will divide two separate ideas: how the physiological model is constructed (**Fig. 1**), and that physiological representation gives rise to concepts like decision-dimensions, the decision-space, and functional roles of circuit elements (**Fig. 2**).

2. The above comment is about within-trial dynamics. It is also needed to elaborate learning dynamics of the decision-making model. For example, when animals make a choice in 2AFC task and when the reward is given, how the model updates behavior based on the given reward. The authors provide mathematical equations, but it would be very difficult for the readers to understand how the model changes decisions over time. Direct comparison of trial-by-trial choice with a simple reinforcement learning model would be very helpful.

We thank the Reviewer for these suggestions. While our decision-space model describes the process of single decisions, not learning dynamics, we introduced an analysis of how the decision-space might shift over the span of learning. In a new **Extended Data Fig. 7j** (legend in lines 1354-1357), we compare dopamine's role during reinforcement learning (RL) in the classical model of the basal ganglia, versus dopamine's role during RL in our model. In **Extended Data Fig. 7k** (lines 1357-1365), we introduce a two-alternative fixed choice (2AFC) which is learned by an RL agent, and plot trial-by-trial choice. Then in **Extended Data Fig. 7l** (lines 1365-1377), we compare across 20 successive 2AFC trials RPEs (which dopamine encodes in the classical model of the basal ganglia) and shifts in the decision-space (which dopamine communicates in our model). The timing of the largest RPEs and decision-space shifts aligns, suggesting that dopamine spikes in such a task could be used as support for either model. During episodes when the animal is

surprised by a pleasant reward, they are most likely to shift from a neutral to a reward-driven decision-space. Conversely, if a reward is suddenly omitted, they might retreat from a reward-predominant decision-space, and if a cost suddenly appears, they may form a cost-predominant decision-space. For reference, we have also added a corresponding section to Methods, **Shifts in the Decision-Space in a Reinforcement Learning Task** (lines 3275-3280), explaining the process behind the simulations in more detail.

Thus, our model offers a new explanation for a commonly observed experimental result. The interpretation of the result, however, is different, with the decision-space model making the testable prediction that the animal has changed its decision-space in response to the unexpected information. A topic of future research will be determining the relationship between reward prediction errors and shifts in the decision-space over learning (in contrast to during a decision, which our model makes a strong claim about). To emphasize this need for future research, we have added a passage to **Discussion** (lines 487-491) covering future directions.

3. Finally, the model is complicated and there are many parameters. How the parameters are chosen needs to be explained further. It would be great if authors could provide any basis for how they chose such parameters shown in Common parameters, whether and how much such parameters are biologically plausible.

We thank the Reviewer for this suggestion. Our model contains many parameters, but they are used to probe the selective effects of brain region individually on the decision-space, action values, or choice, rather than as free parameters that are used to fit data collectively. As an example, two of the parameters in eq. 5, z_{LHb} for overall lateral habenula (LHb) activity, and z_{RMTg} for overall rostromedial tegmental nucleus (RMTg) activity, are added. So, from a strictly computational perspective, these parameters are redundant. However, these parameters serve an important role for probing the biologically plausible connectivity of our modeled circuit to ask what the individual effect of each brain region on the decision-space, if the other regions are assumed not to change (for instance, see the analysis in **Table 6**).

We believe that the reviewer raises important concerns, so in our revised manuscript, we explain in more detail the rationale behind our parameter choices. We approach this in two ways. First, in our new **Extended Data Fig. 1** (legend in lines 940-966), we now explain which parameters are used to measure relative output (for instance, activity in arbitrary units) versus absolute output (for instance, deliberation time in seconds). Our visuals separate the parameters used for the various stages, and instances, of our model. We feel that this description is important to give the reader before diving into the results of our analysis, so we added a cross-reference to it from the second sentence of Results. Second, we added a new Methods section, **Rationale behind parameter choices** (lines 2598-1639), which we also cross-reference from Results. This section details our strategy for defining **Common parameters** on an individual basis.

Without a more detailed and straightforward elaboration of the mechanisms of the model, the Reviewer is concerned that the demonstration explaining several normal and abnormal decision-making processes would have quite limited implications and inspiration to the readers.

We thank the Reviewer for their comments. We understand that our first version of the manuscript offered an insufficiently detailed elaboration of the mechanisms of the model. Our new **Figures 1b,c** and **Extended Data Figure 1** now offer a detailed but straightforward explanation about these mechanisms, which we expect will clarify the analysis on normal and abnormal decision-making processes in the later figures.

4. line 51: (isSPNs and imSPNs)

We thank the reviewer for this correction. We have made the change as suggested on what is now line 53.

The reviewer tried to uncompress the .zip file from Github but an error occurred while

uncompressing the .zip file. The Git clone was also not completely successful (errors related to long file path).

I downloaded the DM_space_model-main.zip file from Github but had an error while unzipping it.

I tried to get the source but the git generated errors - Filename too long.

We thank the Reviewer for raising this point, and we apologize for the confusion. If this reviewer has git installed, we suggest cloning the directory by running "git config --system core.longpaths true" and then "git clone https://github.com/dirkbeck/DM_space_model" from a git command line. If not, we are sending the editor a zipped version of the directory, minus the analysis of the experimental data (which has very large files). The individual files can also be viewed directly on github.

We have also added a detailed description of the organization of the directory and the process by which to run the codes to the README file (previously this description was only in the manuscript). Additionally, we added an **Instructions to run code** section in Methods (lines 2150-2243) with descriptions of the code corresponding to each figure. This includes links to RUNME files where we include step-by-step instructions for running our analysis of experimental data.

REVIEWERS' COMMENTS

Reviewer #1 (Remarks to the Author):

I would like to thank the authors for their careful and thoughtful revisions. They have thoroughly and satisfactorily addressed all the comments and suggestions raised. In my opinion, the modifications introduced in the manuscript, along with the clarifications provided, have significantly improved both its clarity and scientific rigor. I therefore have no further comments or objections, and consider the manuscript suitable for publication in its current form.

Reviewer #1 (Remarks on code availability):

I have not personally executed the provided code, but I have reviewed the contents available in the repository. The folder includes the necessary scripts and usage instructions, which have been improved in both clarity and completeness from the previous version. The documentation now offers sufficient information to understand how to run the code and reproduce the main results, making it more accessible to readers.

Reviewer #2 (Remarks to the Author):

Authors have made a significant improvement on the manuscript, and carefully addressed most of the reviewer's points.

One minor comment: Fig. 1 title can be changed to "A striosome-centered circuit model regulating dopamine release to the striatum.". The current title sounds like that Fig 1. is an established fact.

We thank the reviewer for this point. We changed the Fig. 1 title to "A striosome-centered circuit model regulating dopamine release to the striatum" as suggested.

Reviewer #2 (Remarks on code availability):

I could run some codes and confirmed that those codes generate figures on the manuscript. but since there are too many files, reviewing codes line-by-line was not feasible.